# There Are over 60 Ways to Produce Biocompatible Calcium Orthophosphate (CaPO$_4$) Deposits on Various Substrates

**Sergey V. Dorozhkin** [ID]

Faculty of Physics, M.V. Lomonosov Moscow State University, Leninskie Gory 1-2, Moscow 119991, Russia; sedorozhkin@yandex.ru

**Abstract:** A The present overview describes various production techniques for biocompatible calcium orthophosphate (abbreviated as CaPO$_4$) deposits (coatings, films and layers) on the surfaces of various types of substrates to impart the biocompatible properties for artificial bone grafts. Since, after being implanted, the grafts always interact with the surrounding biological tissues at the interfaces, their surface properties are considered critical to clinical success. Due to the limited number of materials that can be tolerated in vivo, a new specialty of surface engineering has been developed to desirably modify any unacceptable material surface characteristics while maintaining the useful bulk performance. In 1975, the development of this approach led to the emergence of a special class of artificial bone grafts, in which various mechanically stable (and thus suitable for load-bearing applications) implantable biomaterials and artificial devices were coated with CaPO$_4$. Since then, more than 7500 papers have been published on this subject and more than 500 new publications are added annually. In this review, a comprehensive analysis of the available literature has been performed with the main goal of finding as many deposition techniques as possible and more than 60 methods (double that if all known modifications are counted) for producing CaPO$_4$ deposits on various substrates have been systematically described. Thus, besides the introduction, general knowledge and terminology, this review consists of two unequal parts. The first (bigger) part is a comprehensive summary of the known CaPO$_4$ deposition techniques both currently used and discontinued/underdeveloped ones with brief descriptions of their major physical and chemical principles coupled with the key process parameters (when possible) to inform readers of their existence and remind them of the unused ones. The second (smaller) part includes fleeting essays on the most important properties and current biomedical applications of the CaPO$_4$ deposits with an indication of possible future developments.

**Keywords:** calcium orthophosphate; hydroxyapatite; deposits; coating; film; layer; bioceramics; conversion; surface

## 1. Introduction

All known materials have their own specific properties and, depending on the applications, those properties may or may not be desirable. That is, certain ones are aggressive, corrosive or biotoxic; others are sensitive to light, heat and oxidation; some are hydrophilic, transparent, slimy, etc. To eliminate the undesirable properties, the surfaces of improper materials need to be modified. This resulted in the appearance of a specialized sub-discipline of materials science called *surface engineering*, which modifies the surfaces of solid materials in various ways. In a broad sense, surface engineering has applications in chemistry, and mechanical and electrical engineering (especially in relation to semiconductor manufacturing) [1], which is beyond the scope of this review.

Generally, surface modifications can be broadly divided into three categories: (1) depositing a material onto a surface with desired function and properties; (2) transforming an existing surface into a more desirable composition, structure or morphology; (3) partially removing material from an existing surface to create a specific topography [2].

As can be seen from the list of the options, the first two categories involve the application of surface deposits (coatings, films and layers) to solve problems in traditional forms. Regarding biomaterials, their properties are likely to be important when they are implanted in the human body. That is, in the case of artificial bone grafts, synthetic materials used in vivo must have appropriate properties, both surface and bulk, to meet the dual requirements of biocompatibility and application-specific mechanical properties. That is why cytotoxic, genotoxic, allergic, neurotoxic, carcinogenic and mutagenic factors are considered when evaluating the biomedical properties of orthopedic implant materials [2]. To meet all these requirements, the surfaces of bioincompatible materials can be modified with appropriate deposits (coatings, films and layers) to create favorable surface conditions for adsorption of proteins from biological fluids and to promote cell–extracellular matrix interactions and production of growth factors. Otherwise, either fibrous tissue will surround implants made of bioincompatible materials or mechanically weak grafts will not function properly. Both types of defects prolong the healing time. Therefore, diverse surface treatments have been developed to improve the biocompatibility and osteoconductivity of artificial implants [3].

On the other hand, some compounds, such as calcium orthophosphates (abbreviated as $CaPO_4$), are well suited to in vivo applications due to their chemical similarity to the inorganic substances found in mammalian bones and teeth [4–6]. A complete list of known $CaPO_4$ is given in Table 1. However, since all types of $CaPO_4$ are ceramic, they are all mechanically weak (brittle) and cannot be subjected to physiological loads (other than compressive ones) occurring in the human skeleton. For many years, therefore, the clinical applications of $CaPO_4$ alone were largely limited to non-load-bearing areas of the body. However, investigations have continued and researchers have begun to deposit biocompatible $CaPO_4$ on the surface of mechanically strong but biologically inert or biotoxic materials in order to combine the benefits of various materials [7,8]. For example, metal implants are used in artificial joints such as hip joints and artificial tooth roots as sufficient mechanical stability is required. Since no metal alone causes osseointegration, i.e., they do not create a mechanically stable connection between the implant and bone tissue, they are coated with $CaPO_4$ to create osseointegration. However, the problem of osseointegration is not limited to metals. Biodegradable polymers are also generally not bioactive. Therefore, to overcome this disadvantage, the surface of those polymers is also coated with $CaPO_4$ and can be replaced by autogenous bone after implantation, as $CaPO_4$ is involved in the same bone regeneration response as natural bones [7–15].

However, in order to successfully fulfill the important functions (i.e., bioactive adaptation of biologically inert implants), all types of $CaPO_4$ deposits (coatings, films and layers) must meet a number of requirements. The minimum requirements for HA coatings (Table 2) first appeared in the 1992 US guidelines of the Food and Drug Administration (FDA) [16] and sometime later in the International Organization for Standardization (ISO) standards [17]. Subsequently, the FDA guidelines were updated in 1997 [18] and the ISO standards in 2000 [19], 2008 [20] and 2018 [21]. In addition, there is a 2002 ISO standard for the determination of HA coating adhesion strength [22], which was revised in 2018 [23]. In short, important quality characteristics for $CaPO_4$ deposits include thickness, phase composition, crystallinity, Ca/P ratio, microstructure, porosity, surface texture and roughness. All these parameters are likely to affect the mechanical properties of $CaPO_4$ deposits such as cohesion, bond strength, tensile strength, shear strength, Young's modulus, fatigue life and residual stress.

**Table 1.** Existing calcium orthophosphates and their major properties [5,6].

| Ca/P Molar Ratio | Compound | Formula | Solubility at 25 °C, $-\log(K_s)$ | Solubility at 25 °C, g/L | pH Stability Range in Aqueous Solutions at 25 °C |
|---|---|---|---|---|---|
| 0.5 | Monocalcium phosphate monohydrate (MCPM) | $Ca(H_2PO_4)_2 \cdot H_2O$ | 1.14 | ~18 | 0.0–2.0 |
| 0.5 | Monocalcium phosphate anhydrous (MCPA or MCP) | $Ca(H_2PO_4)_2$ | 1.14 | ~17 | c |
| 1.0 | Dicalcium phosphate dihydrate (DCPD), mineral brushite | $CaHPO_4 \cdot 2H_2O$ | 6.59 | ~0.088 | 2.0–6.0 |
| 1.0 | Dicalcium phosphate anhydrous (DCPA or DCP), mineral monetite | $CaHPO_4$ | 6.90 | ~0.048 | c |
| 1.33 | Octacalcium phosphate (OCP) | $Ca_8(HPO_4)_2(PO_4)_4 \cdot 5H_2O$ | 96.6 | ~0.0081 | 5.5–7.0 |
| 1.5 | α-Tricalcium phosphate (α-TCP) | $\alpha\text{-}Ca_3(PO_4)_2$ | 25.5 | ~0.0025 | a |
| 1.5 | β-Tricalcium phosphate (β-TCP) | $\beta\text{-}Ca_3(PO_4)_2$ | 28.9 | ~0.0005 | a |
| 1.2–2.2 | Amorphous calcium phosphates (ACP) | $Ca_xH_y(PO_4)_z \cdot nH_2O$, $n$ = 3–4.5; 15–20% $H_2O$ | b | b | ~5–12 d |
| 1.5–1.67 | Calcium-deficient hydroxyapatite (CDHA or Ca-def HA) e | $Ca_{10-x}(HPO_4)_x(PO_4)_{6-x}(OH)_{2-x}$ (0< $x$ <1) | ~85 | ~0.0094 | 6.5–9.5 |
| 1.67 | Hydroxyapatite (HA, HAp or OHAp) | $Ca_{10}(PO_4)_6(OH)_2$ | 116.8 | ~0.0003 | 9.5–12 |
| 1.67 | Fluorapatite (FA or FAp) | $Ca_{10}(PO_4)_6F_2$ | 120.0 | ~0.0002 | 7–12 |
| 1.67 | Oxyapatite (OA, OAp or OXA) f, mineral voelckerite | $Ca_{10}(PO_4)_6O$ | ~69 | ~0.087 | a |
| 2.0 | Tetracalcium phosphate (TTCP or TetCP), mineral hilgenstockite | $Ca_4(PO_4)_2O$ | 38–44 | ~0.0007 | a |

a These compounds cannot be precipitated from aqueous solutions. b Cannot be measured precisely. However, the following values were found: 25.7 ± 0.1 (pH = 7.40), 29.9 ± 0.1 (pH = 6.00), 32.7 ± 0.1 (pH = 5.28). The comparative extent of dissolution in acidic buffer is: ACP >> α-TCP >> β-TCP > CDHA >> HA > FA. c Stable at temperatures above 100 °C. d Always metastable. e Occasionally, it is called "precipitated HA (PHA)". f Existence of OA remains questionable.

**Table 2.** FDA requirements for HA coatings [16].

| Properties | Specification |
|---|---|
| Thickness | Not specific |
| Crystallinity | 62% minimum |
| Phase purity | 95% minimum |
| Ca/P atomic ratio | 1.67–1.76 |
| Density | 2.98 g/cm$^3$ |
| Heavy metals | <50 ppm |
| Tensile strength | >50.8 MPa |
| Shear strength | >22 MPa |
| Abrasion | Not specific |

## 2. General Knowledge, Terminology and Definitions

Wikipedia describes the term coating as a covering applied to the surface of a substrate with the purpose of improving surface properties, such as wettability, adhesion and appearance, as well as corrosion, wear and scratch resistance [24]. Of course, this also applies to films, which are generally considered to be thinner coatings than coatings. Layer is one more significant definition, which means a single thickness of another material that covers

a surface and/or forms a part or section on it. Lastly, deposition process is described as applying a coating, film or layer onto a surface.

Historically, the use of various coating materials dates back to the metal age. Animal fats, beeswax, gelatin, vegetable oils and various clay minerals are known to have protected the first metal tools and artifacts (iron, brass and silver). Water repellency, shine, corrosion protection, wear resistance and lubrication are examples of properties sought by early humans. Later, Egyptians painted on papyrus, tomb walls, coffin lids and many other places. The oldest surviving tomb murals at Hierakonpolis date from around 3400 BC. Consider also the gilding techniques that have been practiced continuously for at least four thousand years. These techniques are also thought to have been practiced earliest by the Egyptians. Many excellent surviving pieces, including statues, crowns and coffins, attest to this high level of skill. For example, 18th Dynasty (1567–1320 BC) leaf specimens from Luxor are thought to be about 0.3 μm thick. Such leaves were carefully glued and adhered to the smoothened wax or resin-coated wooden surface by mechanical (cold) gilding to produce the earliest veneers [25]. Returning to the subject of this review, the earliest research paper on $CaPO_4$ coatings that I can find was published in 1965 [26]. Since then, more than 7500 papers have been published on various aspects of the subject, which is a good representation of the practicality of the topic.

The creation of coatings, films and layers is one of the oldest arts and at the same time one of the newest sciences, but the distinction between these terms is not yet well established and, moreover, can vary from one field of science or technology to another. Namely, in the food industry a coating is a thin layer of a material formed as a coating, while a film is a thin layer, prepared from a material, which can be placed on or between other components [27]. The main difference between them is that coatings are applied in liquid form, while films are molded as solid sheets and afterwards are applied as wrappings [28]. Thus, the term "layer" defines the terms "coating" and "film".

To further clarify this theme, extensive searches of scientific databases (Scopus, ISI Web of Knowledge) revealed a large number of fixed collocations. For example, according to Scopus (as of April 2023), the combination of the terms "wear protection + coating" is used more frequently in the title of a publication than the terms "wear protection + film" (284 and 29 occurrences, respectively). Conversely, the combination "ferroelectric + film" is used much more frequently than the combination "ferroelectric + coating" (9524 and 127 entries respectively). The "apatite + coating" combination, the subject of this study, was found in 5290 publication titles, while the "apatite + film" and "apatite + layer" combinations were found in 848 and 765 publication titles, respectively. Similar correlations were found for "calcium + phosphate + coating", "calcium + phosphate + film" and "calcium + phosphate + layer" combinations, which were found in 1595, 265 and 301 publication titles, respectively. As can be seen from these figures, $CaPO_4$ deposits of all types are most commonly associated with coatings. However, to balance the impact of terminology, the term "$CaPO_4$ deposits" will be used in this review instead of "$CaPO_4$ coatings", "$CaPO_4$ film" or "$CaPO_4$ layers". Perhaps the facts mentioned above are just a matter of terminology or a matter of habit in each field of science and technology.

Since almost any type of material can be deposited and/or fabricated on both similar and dissimilar substrates, there are many possibilities to classify the known types of deposits (coatings, films and layers). For example, metallic, polymeric, ceramic and composite deposits can be classified according to their structural material. Similarly, they can be classified according to the state of the precursor material (gas, liquid or solid) and the temperature used for deposition. They can also be classified according to properties such as biodegradability, edibility, transparency, reflectivity, conductivity, hardness, porosity, solubility and permeability, as well as adhesion to the substrate. Furthermore, using a formation approach, all coating types can be divided into two broad categories: (i) transformation type (e.g., oxide film formation by surface oxidation) and (ii) deposition type (Figure 1) [29]. Furthermore, deposition can also be classified according to the deposition technique (Table 3) [30–32], which results in either line-of-sight (deposited on only one surface in front of the coater) or non-line-of-sight (deposited on all surfaces, resulting in

coatings with complex structures). Importantly, coatings and films can consist of one or many individually deposited layers, all of which can be divided into single-layer deposition and multi-layer deposition. The former is done by a single process, while the latter is done by layer-by-layer deposition. Furthermore, the individual layers of a multilayer film or coating are either indistinguishable (in which case they behave like a thick single-layer one) or they can be distinguished from each other. In the latter case, there may be an opportunity (sometimes only hypothetically) to remove one or several individual layers from the surface, making the deposit thinner. Furthermore, in layer-by-layer deposition techniques, it is easily possible to change the composition of the deposited material and thus deposit multilayered structures with gradual changes in composition and/or properties. This allows for classification into graded and ungraded deposits. Furthermore, the deposit itself can completely cover the substrate or only partially cover the substrate, which is still another classification. Finally, any type of deposit can be thin or thick. These terms are relative and the distinction between them is not quite clear. Generally, researchers consider thin deposits to be those that are nanometers to a few micrometers thick. Therefore, all types of thick deposits have a thickness exceeding several micrometers. Interestingly, according to scientific databases, deposits of all kinds are more frequently "thin" than "thick". That is, according to Scopus (as of April 2023), the combination of the terms "thin + coating" is used more often than "thick + coating" in the titles of research papers (with 10,097 and 4337 publications, respectively). Similarly, the combination "thin + film" in the title of research papers is more commonly used than the combination "thick + film" (265,246 and 23,571 publications, respectively), while a combination of terms "thin + layer" is used more repeatedly, when compared with that of "thick + layer" in the title of research papers (56,696 and 16,274 publications, respectively). As can be seen from the numbers, the combination "thin film" appears to be the most common and "thick coating" is the least common one.

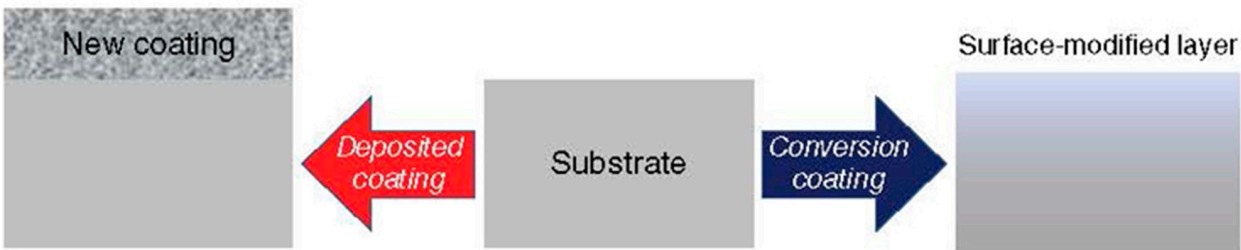

**Figure 1.** A schematic diagram showing the difference between formation of the conversion coatings and the deposited ones. Reproduced with permission from [29].

**Table 3.** A mutual comparison of several deposition techniques of calcium orthophosphate coatings, films and layers on various substrates [28–30].

| Technique | Thickness | Advantages | Disadvantages |
|---|---|---|---|
| Plasma spraying | 30–200 μm | A simple and flexible technique; uniform and smooth coatings are produced; high deposition rates; low cost | Line-of-sight technique; high temperatures induce partial decomposition and formation of non-stoichiometric and amorphous compounds; expensive equipment; simultaneous incorporation of biological agents is impossible; rapid cooling produces cracks |
| Flame spraying | 100–250 μm | Most economical among all thermal spraying techniques; easily adaptable; porous deposits | Line-of-sight technique; high temperatures induce partial decomposition and formation of non-stoichiometric and amorphous compounds; crack development at lower temperatures, simultaneous incorporation of biological agents is impossible; rapid cooling produces cracks |

**Table 3.** *Cont.*

| Technique | Thickness | Advantages | Disadvantages |
|---|---|---|---|
| High velocity oxy-fuel spraying | 30–200 μm | High deposition rates; uniform deposition; improved wear and corrosion resistance and biocompatibility; no post treatment required | Line-of-sight technique; high temperatures induce partial decomposition and formation of non-stoichiometric and amorphous compounds; simultaneous incorporation of biological agents is impossible; rapid cooling produces cracks |
| RF magnetron sputtering | 0.5–3 μm | Uniform coating thickness on flat substrates; high purity and adhesion; dense pore-free deposits; excellent coverage of steps and small features; ability to coat heat-sensitive substrates | Line-of-sight technique; expensive; low deposition rates; produces amorphous coatings; high temperatures prevent from simultaneous incorporation of biological agents |
| Pulsed laser deposition (laser ablation) | 0.05–5 μm | Coatings with crystalline and amorphous phases; dense and porous coatings; high adhesive strength; ability to produce wide range of multilayer coatings from different materials | Line-of-sight technique; expensive; high temperatures prevent simultaneous incorporation of biological agents; lack of uniformity |
| Ion beam assisted deposition | 0.05–1 μm | Uniform coating thickness; high reproducibility and reliability; dense; high adhesion; wide atomic intermix zone at the coating/substrate interface | Line-of-sight technique; expensive; produces amorphous coatings |
| Sputtering | 0.5–3 μm | Uniform coating thickness on flat substrates; dense; high adhesion | Line-of-sight technique; expensive equipment; time-consuming; produces amorphous coatings |
| Electrostatic spray deposition | 10 nm–30 μm | Low cost; easy set-up; ambient conditions; a wide choice of both precursors (dissolved salts, suspensions, sols) and substrates | Line-of-sight technique; problems coating large surfaces; low flow rates; requires high temperatures to decompose the precursor solvents and salts |
| Dip coating | 2 μm –5 mm | Easy set-up; low cost; coatings applied quickly; can coat complex substrates | Requires high sintering temperatures; possible thermal expansion mismatch; crack appearance |
| Spin coating | 2 μm–0.5 mm | Easy set-up; low cost; coatings applied quickly | Requires high sintering temperatures; possible thermal expansion mismatch; crack appearance; cannot coat complex substrates |
| Sol–gel technique | <1 μm | Can coat complex shapes; low processing temperatures; thin coatings; inexpensive process; can incorporate biological molecules | Some processes require controlled atmosphere processing; expensive raw materials; high permeability; low wear resistance; hard to control the porosity |
| Electrophoretic deposition | 0.1–2.0 mm | Uniform coating thickness; rapid deposition rates; simple setup; low cost; can coat complex substrates; can incorporate biological molecules | Difficult to produce crack-free coatings; requires post treatment at high temperatures |
| Electrochemical (cathodic) deposition | 0.05–0.5 mm | Good shape conformity; room temperature process; uniform coating thickness; short processing times; can incorporate biological molecules | Sometimes stressed coatings are produced, leading to their poor adhesion with substrate; requires good control of electrolyte parameters |

**Table 3.** *Cont.*

| Technique | Thickness | Advantages | Disadvantages |
|---|---|---|---|
| Biomimetic process | <30 μm | Low processing temperatures; can form bonelike apatite; can coat complex shapes; can incorporate biological molecules | Very low deposition rates; requires replenishment and a pH constancy of the simulating solutions (HBSS, SBF, etc.) |
| Hydrothermal deposition | 0.2–2.0 μm | Coatings are crystalline; can coat complex shapes | High pressure and temperatures are required |
| Thermal substrate deposition | 0.2–2.0 μm | Deposition is enhanced by heat and current; different $CaPO_4$ phases can be formed | Less common technique; coatings of diverse crystallinities are produced |
| Hot isostatic pressing | 0.2–2.0 μm | Produces dense coatings; homogeneous structure; high uniformity; high precision; no dimensional or shape limitations | Cannot coat complex substrates; high temperature required; thermal expansion mismatch; elastic property differences; expensive; removal/interaction of encapsulation material; high temperatures prevent simultaneous incorporation of biological agents |
| Micro-arc oxidation | 3–30 μm | Simple, economical and environmentally friendly technique, suitable for coating of complex geometries | Unless the proper electrolytes are used, the procedure rather should be considered as a pre-deposition technique onto which $CaPO_4$ are deposited by other methods |
| Dynamic mixing method | 0.05–1.3 μm | High adhesive strength | Line-of-sight technique; expensive; produces amorphous coatings |

It is also worth mentioning the reasons why surfaces of different materials are coated. There are many of them. These are:

1. The kernel contains substances that are toxic, cause adverse or allergic reactions, or have a bitter or unpleasant odor;
2. The coating protects the core material from its environment and increases its stability and shelf life;
3. The coating improves mechanical integrity, which means that coated products are more resistant to misuse (e.g., wear and tear);
4. Modification of the surface properties of the core, such as biocompatibility, light reflection, electrical conductivity, color, etc.;
5. Decoration (in cases where the core alone is tasteless);
6. The core contains material that can easily migrate and stain hands, clothes, etc.;
7. Changing the release profile of active ingredients, such as pharmaceuticals, from the core.

Reasons 1, 2, 3, 4 and 7 are generally applicable to the biomedical sector, while reasons 1, 2 and 4 are relevant to the subject of this review.

At the end of this section, it is worth noting that in a sense all types of cladding materials are functionally similar to graded ones, but both composition and properties at the core/coating interface have very high gradients.

## 3. Brief Knowledge on the Important Pre- and Post-Deposition Treatments

Since $CaPO_4$ bioceramics have excellent biomedical properties but poor mechanical ones [4–6], extensive research has been focused on their spreading over the surfaces of materials possessing a better mechanical behavior. As a result, various deposition techniques of $CaPO_4$ have been developed. Some of them are mentioned in Table 3. In most cases, however, the surface of substrates must be prepared before deposition. This preparation usually includes cleaning and degreasing to remove any surface contamination that may have occurred during production or storage. Depending on the nature and properties of the substrate, it can be cleaned with solvents such as acetone, alcohols, trichloroethylene and mixtures thereof, or distilled water [33–43]. In addition,

various types of mechanical modifications of the surface are widely used to increase the mechanical fixation between $CaPO_4$ deposits and substrates. Examples include sandblasting [41,44,45] or grit-blasting [33,35,36,46,47], abrading [37,48], polishing [39,40,43,49] and grinding [38]. In addition, the substrate surface can be chemically treated, modified and/or functionalized to promote chemical bonding between the deposit and the substrate [50,51]. Examples include chemical activation [36,48,52], acid [33,36,37,39,41,45,48,52,53] and/or alkali [34–37,39,41,48,54–60] treatments, anodizing [38,40,42,61], etching [35,36,52,53,60,62], chemical polishing [39,42], oxidation [36,39,42,58–60,63], pickling [48], passivation and/or discoloration, phosphorylation [64–66], grafting [67,68] and silanization [51]. For example, in the case of polymers, etching can be carried out with solvents [62]. In another study, a mixture of nitric, hydrochloric and sulfuric acid combined with hydrogen peroxide was used to functionalize the surface of carbon fibers [69]. All these processes seem to be particularly important when $CaPO_4$ needs to be deposited on chemically inert surfaces such as carbon fibers [69], carbon nanotubes [70] and graphene nanosheets [71]. In the case of chemically active metals such as Mg and its alloys, boiling water treatment is sometimes applied to form Mg hydroxide layers on their surfaces [72]. Most such treatments are carried out by dipping, spraying, rinsing or immersion, depending on both the quality requirements and the limitations of the substrate to be coated.

In addition, there are physical treatments that result in various types of surface modifications. Examples include thermal treatments [36,37], glow discharge [73–75], ultraviolet [49,75–77], and high-energy low-current DC electron beam [78,79] irradiations, as well as electron cyclotron resonance plasma oxidation [80] and laser [81,82] treatments, which together produce very thin surface layers of the substrate with oxidation and/or partial degradation [49,73–81] or surface texturization [82]. More specifically, the surface of substrates can be coated with an interlayer of another compound [48,67,83–98] containing special binding agents [99–101] before the $CaPO_4$ are deposited. That is, zinc orthophosphate [48], bioglass [88,89], titania [90–92], SiC [93], calcium carbonate [94], titanium [95,96] and diamond [97] coatings have been pre-deposited on the surface of various substrates onto which $CaPO_4$ are subsequently deposited. Similarly, organophosphate polymers have been chemically bonded to chemically inert polyethylene via surface graft polymerization of phosphate-containing monomers [67]. For the same purpose, collagen was also immobilized by pre-deposition [83]. It was also found that pre-depositing Ti powder on the surface of the Ti-6Al-4V alloy substrate reduces residual stresses in the deposited HA and increases the adhesion strength of the interface (Figure 2). Thus, Ti appears to be a bonding layer between Ti-6Al-4V alloy and HA [95]. In other studies, Ti substrates were wet blasted with HA/Ti mixed powder and a bond layer was formed by embedding HA/Ti powder [102–104]. In addition, some studies used commercial inkjet printers to print a defined pattern on the titanium underlying oxide layer, which was then coated with a superhydrophobic coating. Furthermore, a commercial inkjet printer was used to print patterns on the titanium underlying oxide layer, which was then coated with a superhydrophobic coating, and the superhydrophobic molecules in some regions were decomposed by UV light to form superhydrophobic and superhydrophilic micropatterns. $CaPO_4$ were then deposited on both superhydrophobic and superhydrophilic areas [98]. In addition, inert compounds can be pre-deposited to eliminate chemical interactions between the substrate surface and $CaPO_4$ deposits. In other words, an intermediate FA layer was added to prevent chemical reactions between $ZrO_2$ and HA during sintering [105,106]. In addition, ion implantation of calcium and phosphorus can alter the substrate surface, leading to partial $CaPO_4$ formation [107–110]. Calcium or orthophosphate ions can also be pre-adsorbed or incorporated [53,60,111]. Furthermore, pre-immobilization of some biologically active compounds (biomolecules) such as alkaline phosphatase enzyme [112–114] to the substrate surface can be used to facilitate $CaPO_4$ deposition. As can be seen from the references cited above, several sequential treatments of the substrate surface are widely used.

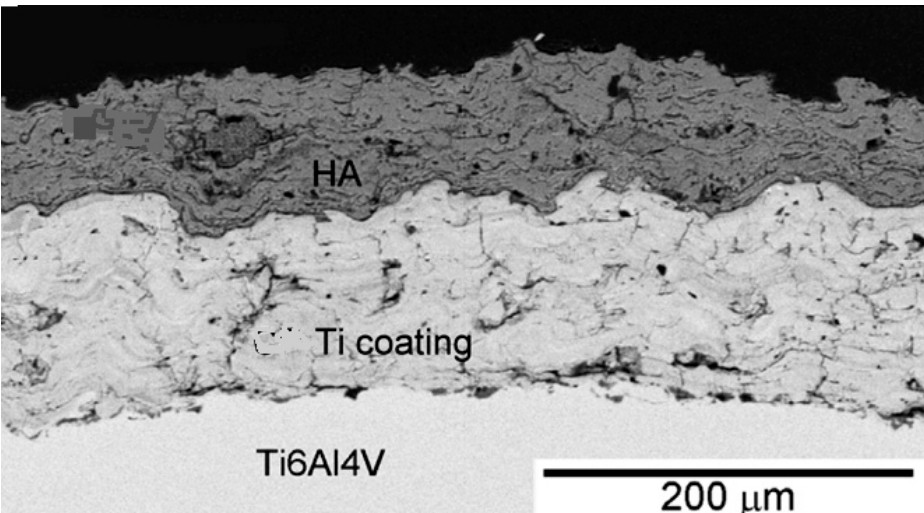

**Figure 2.** A cross-sectional SEM image of a plasma-sputtered HA deposited onto a Ti bond coating over the surface of a Ti-6Al-4V alloy substrate. Reproduced with permission from [95].

Furthermore, once $CaPO_4$ deposits are produced, various sorts of post-deposition actions can be used to induce crystallization/recrystallization of various compounds, improve their anchorage and evaporate the trapped solvent traces. Namely, post-deposition heat treatments (annealing) transform the deposited amorphous (ACP) and non-apatite phases into more stable compounds such as HA, simultaneously increasing their crystallinity, improving corrosion resistance and further reducing residual stresses [44,47,115–125]. All these annealing processes are carried out at high temperatures as $CaPO_4$ were deposited on thermally stable substrates. However, $CaPO_4$ were also deposited onto various polymeric substrates that cannot withstand high temperatures. However, it has been found that $CaPO_4$ deposits on polymers can be annealed by prolonged (~10 h) annealing at temperatures just below the melting point of the polymer [126,127]. Laser remelting [128] and laser-induced crystallization [129] can also be used.

The presence of water in the post-coating heat treatment also plays an important role in the conversion of ACP to HA [130–134]. Namely, the presence of water vapor caused a significant increase in the crystallinity of the coating compared to heat treatment at 450 °C in the dry state [132,133]. Self-healing of microstructural defects was also found [135,136]. It is also possible to chemically treat $CaPO_4$ coated samples with alkaline aqueous solutions [43,124,137,138] or F-containing aqueous solutions [139,140] for the same purpose. Similarly, coated samples can be stored in boiling water or treated hydrothermally [10,63,141–146]. Both saturated steam treatment [147] and immersion in water at 37 °C [44] can be used. Sometimes this is performed to convert one type of $CaPO_4$ deposit into another, such as converting DCPD deposits into HA [148–152]. In addition, various sorts of physical treatments can be used, as well. The examples comprise corona electrical charging [153], laser microstructuring [154], electron beam irradiation [36,155–157], oxygen plasma [62] and pulsed photon [158] treatments. Some of the results of such treatments are shown in Figures 3 and 4. Besides changes in phase and composition (Figure 3), the post-deposition treatments cause changes in both the surface morphology and roughness of deposited $CaPO_4$ (Figure 4). Finally, other compounds such as stearic acid [159], sodium bisphosphonate [160] and bioactive glasses [161] can be adsorbed or deposited onto the $CaPO_4$ coatings to produce additional properties such as water repellency [159]. It should be emphasized that the use or not of post-deposition techniques has a significant impact on the structure and properties of $CaPO_4$ deposits.

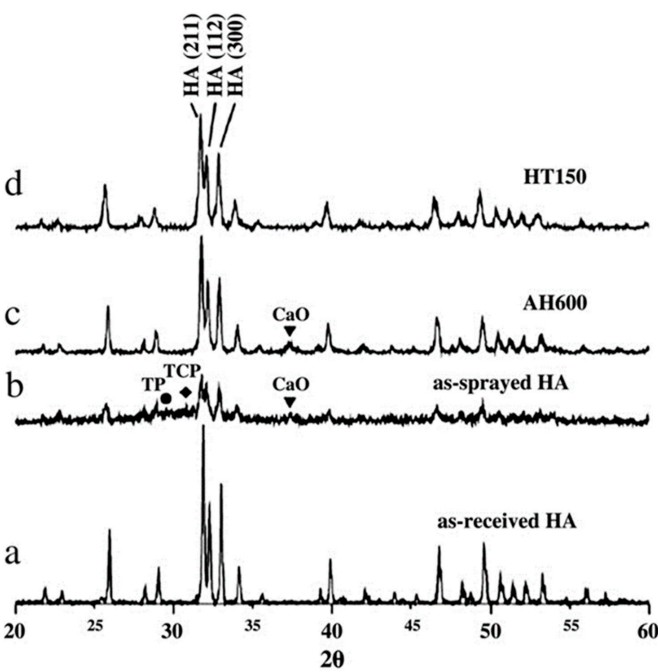

**Figure 3.** XRD patterns of (**a**)—initial HA powder, (**b**)—sprayed HA coating, (**c**)—HA coating heat treated at 600 °C (AH600) and (**d**)—HA coating hydrothermally treated at 150 °C (HT150). It can be easily seen that the crystallinity and phase purity of the poorly crystalline HA coatings are increased by both thermal and hydrothermal treatments. Reproduced with permission from [142].

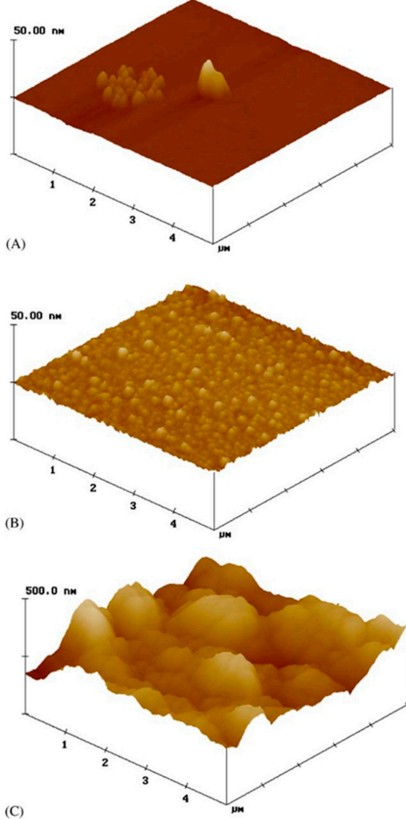

**Figure 4.** Typical surface morphologies of CaPO4 coatings sputtered onto glass substrates: (**A**) sputtered surface; (**B**) after 1 h annealing at 350 °C in the absence of water vapor; (**C**) after 1 h annealing at 350 °C in the presence of water vapor. Reproduced with permission from [28].

**Part 1. Methods to Produce Biocompatible CaPO$_4$ Deposits**

**4. CaPO$_4$ Deposited on Various Substrates**

Since the majority of deposition processes are non-equilibrium, commonly the materials deposited will be different from those used for deposition. This means that the composition of the coatings, films and layers after deposition is not necessarily constrained by the phase diagram. Furthermore, the major parameters such as amorphization, crystallinity, microstructure and surface morphology depend on the deposition technique, as well as using or non-using of pre- and post-deposition actions. Moreover, different techniques produce deposits with dissimilar properties, morphologies and surface structures. This is due to specific formation mechanisms, most of the details of which are unknown. Furthermore, almost all deposition processes of CaPO$_4$ can be realized under the simultaneous application of additional physical forces such as magnetic fields [162,163], UV light [164] or ultrasonic treatment [139,165,166]. In non-wetting techniques, the CaPO$_4$ deposition process can be carried out in the presence of water to optimize moisture [167]. All these additional processing variables lead to further variations in the composition, structure, crystallinity, orientation, porosity, hardness, adhesion and other properties of CaPO$_4$ deposits.

*4.1. Thermal Spraying Deposition Techniques*

Thermal spraying is the process of depositing a thermally melted or thermally softened material by spraying it onto a cold surface. The material, including the coating material or its precursors, can be heated by various methods, such as high-temperature flames or plasma jets, and thermal spraying is classified as flame spraying and plasma spraying, and the maximum achievable temperature is the main difference between them. In each case, the deposited material is fed into a jet stream, where it is melted or thermally softened, and the resulting droplets are flattened and pushed towards a substrate. As the temperature of the jet drops rapidly as a function of distance, the droplets quickly solidify to form a deposit. Typically, the deposit consists of a series of overlapping crepe-shaped lamellae, called "splats", formed by the flattening of liquid or softened droplets [168]. These splats remain attached to the substrate and are a component of the deposits. Due to the very high process temperatures, thermal sputtering techniques are characterized by the high kinetic energy of the flux components, the monatomic to droplet uncertainty of the flux molecular composition even when the elemental composition is well defined (droplets may contain remnants of the crystalline phase) and the presence of ionization components. Moreover, very rapid (~10$^6$ K/s) cooling often leads to the formation of metastable and amorphous phases, morphologically and structurally heterogeneous phases, as well as deviations from the elemental composition of the initial powder due to the loss of volatile components. A high degree of instability is therefore predetermined in thermal spraying technology. In this context, the growth mechanism of the sputtered deposits has not yet been determined.

In general, the quality of thermal spray coatings improves with increasing particle velocity. Since the raw materials are usually powders of a few μm to 1 mm, the lamellae have μm thickness and transverse dimensions of a few μm to several hundred μm. Thermal spraying therefore always produces thicker deposits (from ~20 μm to several mm depending on the raw materials and process) over a larger area at a higher deposition rate than the other processes listed in Table 3 [169].

As far as I have discovered, the idea of deposition of CaPO$_4$ by thermal spraying was introduced in Japan. So, on 3 April 1979, a patent was issued for "an improved bone, joint and root implant for bones, joints and teeth comprising a metal base material, an HA coating layer formed by thermal spraying of HA powder or a mixture of HA powder and ceramic powder around the periphery of the metal base material, and optionally a binding agent between the metal base material and the HA layer, and further ceramic layers" [170]. This patent application was filed on 29 December 1976, so that date can be considered as the earliest reference to thermal spraying of CaPO$_4$.

Since thermal spraying is carried out at very high temperatures, the adhering hot droplets continuously heat the substrate. As a result, phase transformation and recrystallization can occur near the surface. For example, plasma-sputtered HA coatings on low modulus Ti-24Nb-4Zr-7.9Sn alloy substrates were found to undergo martensitic transformation and recrystallization near the surface. These two phenomena were attributed to a combination of temperature and cooling processes [171]. Indeed, such phenomena introduce additional uncertainties in both the mechanical properties and adhesion of the deposits. Moreover, the substrate temperature was found to play an important role in the splat morphology (Figure 5), which also affects the properties of the deposited $CaPO_4$. Preheating of the substrate can be applied to influence its properties. Namely, deposition without preheating the substrate was found to result in HA coatings with a crystallinity gradient in the thickness direction, but this gradient practically disappeared when the substrate was preheated to 200 °C [172]. Furthermore, the substrate can be maintained at an optimum temperature with appropriate cooling techniques [173,174].

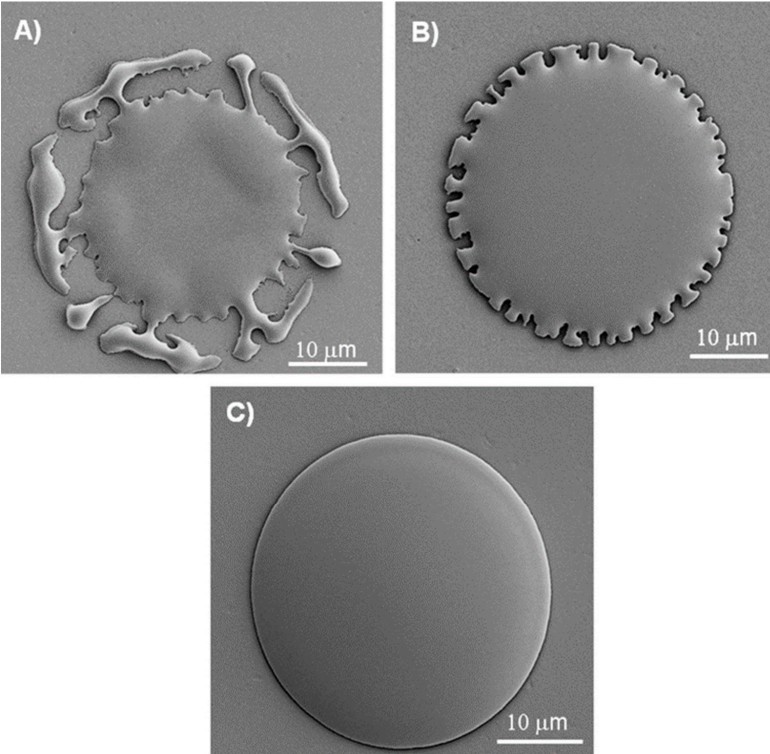

**Figure 5.** Typical topographic morphologies of single flame-sprayed $CaPO_4$ splats deposited onto Ti substrates preheated to various temperatures: (**A**)—no preheating (ambient conditions), (**B**)—preheated to 100 °C, (**C**)—preheated to 300 °C. Reproduced with permission from [174].

As a result, $CaPO_4$ thermal spray deposits on implants are produced commercially. For example, APS Materials Inc. (Dayton, OH, USA) and Osstem Implant Co. (Seoul, Korea) manufacture Ti implants coated with HA.

### 4.1.1. Plasma Spraying

Commonly, plasma is referred to as the fourth state of matter because, unlike the gaseous, liquid and solid states, it does not obey the classical physical or thermodynamic laws. In industry, plasma is used for various processing techniques, including modification and activation of various surfaces [175]. Plasma spraying is one of the rapid solidification techniques and requires equipment that produces a hot jet of ionized gas in the form of a plasma. Such a device is called a plasmatron (plasma generator) and is powered by an electric arc or high-frequency discharge.

According to reference [8], plasma spray technology was accidentally discovered by a student in 1970, who used the equipment to study the melting and rapid solidification of aluminum oxide coatings on metal surfaces [176]. Plasma thermal spraying is a technique in which a direct current (DC) arc (synonyms: plasma torch, plasma gun) is created between two electrodes, through which a stream of gas (usually Ar, but He, $H_2$ and $N_2$ can also be used) passes. A high-voltage discharge between the cathode and anode in the arc causes these gases to become an ionized mixture (plasma) at very high (~20,000 K) temperatures and speeds as high as ~400 m/s. The material to be deposited (a raw material) is usually a powder, but it can also be added to the plasma jet as a liquid, suspension or wire. The temperature of the plasma jet drops rapidly as a function of distance; typical temperatures remain around 2000–3000 K only 6 cm outside the electrodes but they drop further with increasing distance. The injected particles are only exposed to high temperatures for a very short time (~$10^{-3}$ to $10^{-4}$ s), but most become sufficiently heated to reach a molten, semi-melted or at least softened state, which is important for adhesion. Therefore, every sprayed surface (metal, ceramic or organic material such as paper) is coated. A single pass of the plasma gun produces a layer about 5–15 lamellae thick. Once the entire substrate has been coated with one layer, the gun returns to its initial position and another layer is applied [177]. Therefore, plasma-sprayed deposits frequently consist of several layers. By choosing the optimum relations between the size of the particles (molten deposits of a given thickness on bigger particles occupy a smaller relative volume than on small particles), the type of gas (the heat capacity of the plasma, and hence its ability to raise the temperature of the particles, is highly dependent on the gas used) and its pressure, the plasma velocity (the longer the particles stay within the plasma, the higher their temperature) and cooling process of the substrates, one prepares $CaPO_4$ deposits with the desired crystallinity and composition. Typical current values used for thermal deposition of HA range from 350 A [131] to 1000 A [178]. An excellent schematic setup of the plasma sputtering process is available in the literature [29,32,177–181]. A typical image of plasma sputtered HA deposits is shown in Figure 6 [180].

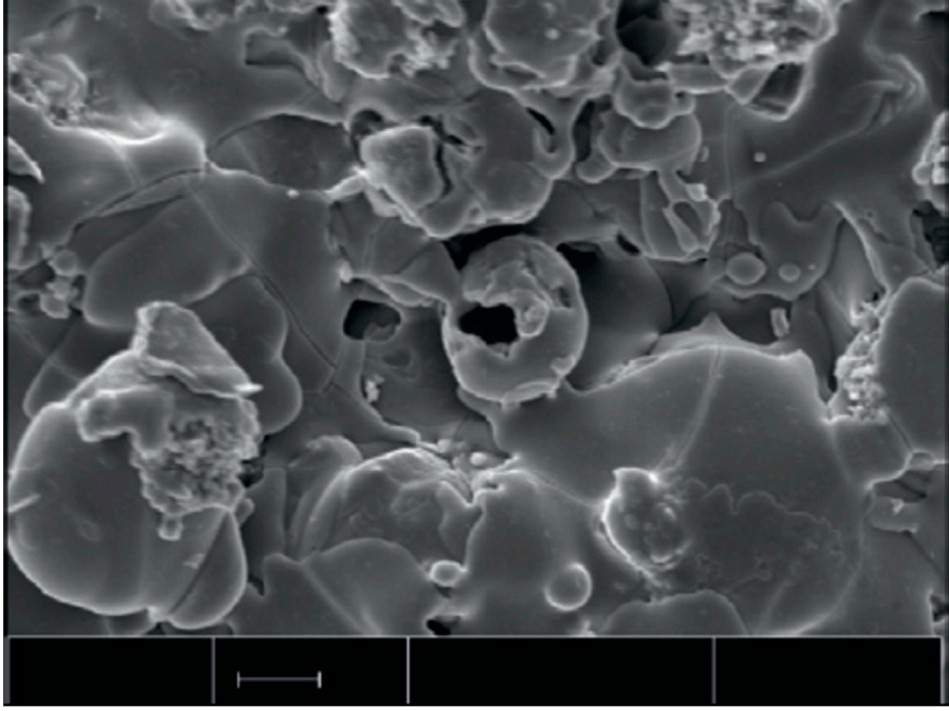

**Figure 6.** A scanning electron micrograph of a typical plasma-sprayed HA coating on a titanium implant. As seen in the image, the morphology of the coating has a rough relief resulting from the presence of drops of different sizes in the flux. The bar is 10 μm. Reproduced with permission from [180].

Numerous variations of the plasma spray technique are available: atmospheric pressure [182,183], supersonic atmospheric pressure [184], vacuum and/or low pressure [185,186], axial [187], powder [10,141], suspension [188–194], liquid and/or solution [189], liquid and/or solution precursor [195–197], gases [198,199], gas tunneling [200–202], high frequency [203,204], low energy [205], high power [172,206], and pulse [207,208] plasma sputtering techniques can be used to deposit $CaPO_4$. For example, compared with conventional atmospheric pressure plasma spraying, the low heat input to the plasma torch can avoid overheating and subsequent pyrolysis of powder particles and excessive local overheating of the coating/substrate [205]. Biphasic $HA/\beta$-TCP coatings were produced from a mixture of high purity DCPD and $CaCO_3$ precursor powders with a Ca/P molar ratio of 1.67. The powders were formed into free-flowing aggregates of 105–300 μm and 300–500 μm in size and plasma sprayed onto sand-blasted Ti6Al4V coupons [209]. A similar approach uses a mixture of DCPD and $CaCO_3$ powders to deposit HA with minor admixtures of TTCP and CaO [210]. In both cases, the reactions between the precursors were performed inflight. All these modifications have some specific advantages, for example coatings of 5–50 μm can be obtained, which are several times thinner than those obtained by dry powder processing [181]. In addition, there is microplasma spray technology [211–215], characterized by small dimensions, low noise (25–50 dB), almost no dust and low power consumption (e.g., 1–4 kW, c.f. 10–40 kW or more). These allow operation under normal working room conditions. $CaPO_4$ deposition is possible on small parts and components containing thin sections, which is not possible with other methods. The low heat input of the microplasma jet reduces overheating of both the deposition material and the substrate. In addition, microplasma jetting generally reduces the formation of additives and/or amorphous phases, resulting in a higher degree of phase purity and crystallinity (e.g., >80%, c.f. ~70%), and a high degree of porosity (e.g., ~20%, c.f. ~2–10%) which facilitates the growth of bone tissue. The mechanical properties of the deposits are generally good [211–215]. This expands the field of application of plasma spraying to produce various functional coatings.

It is important to emphasize that small voids such as cracks and pores [216], as well as incomplete bonding areas, are present in the deposited lamellae. Due to this heterogeneous structure, plasma-sputtered deposits could possess properties significantly different from those of the initial bulk material [177]. Furthermore, due to very high processing temperatures (coupled with partial loss of volatile $P_2O_5$ at high temperatures, leading to incompatible melting of $CaPO_4$ with complex dehydration, dehydrogenation and decomposition reactions) and subsequent rapid solidification, various inclusions and metastable phases are always formed in the deposits. For example, in the case of plasma sputtering of $CaPO_4$, complex mixtures and/or solid solutions of various $CaPO_4$ phases, such as $\alpha$-TCP, $\beta$-TCP, HA, high-temperature ACP, OA, TTCP combined with other compounds such as calcium pyrophosphate, calcium metaphosphate and CaO are obtained [44,47,217–226]. Of these, the presence of CaO causes the most serious problems because the hydration reaction that occurs during storage and after implantation in vivo converts CaO to $Ca(OH)_2$, increasing the volume by about 50% and leading to significant internal deterioration and cracking, especially when CaO is present in large quantities [218]. Therefore, the chemical and phase compositions of the final deposit also depend on the thermal history of the powder particles. Thus, the solubility of the plasma sprayed deposits varies depending on the amounts of soluble phases such as CaO and ACP. Moreover, distribution of amorphous, metastable phases as well as by-products is always heterogeneous. For example, the crystallinity of the coating is reported to be lower at the interface with the Ti substrate than at the surface of the coating. This occurred due to the higher cooling rate of the first layer as the metal has a higher thermal diffusivity than $CaPO_4$ [220]. Moreover, the influence of the presence of amorphous and metastable phases on the overall properties of the deposits depends on their location. That is, such phases present in the surface layer promote bone tissue growth due to their high bioabsorbability, whereas, near the coating/substrate interface, the rapid dissolution of amorphous and metastable phases reduces adhesion and the coating delaminates before bone tissue is formed. In addition, residual stresses (two types of stresses:

compressive and tensile; see below) in plasma-sputtered $CaPO_4$ are also a factor in the thermal expansion coefficient of $CaPO_4$. Since the thermal expansion coefficient of metals exceeds that of $CaPO_4$, compressive residual stresses were observed in metal substrates coated with $CaPO_4$. The values range from −140 to −10 MPa, depending on the deposition conditions, as well as on an application or non-application of pre- and post-deposition techniques [95,213,227–231]. Furthermore, immersion in a simulated body fluid (SBF) was found to lead to zero compressive residual stresses and compression-to-stress redistribution [95]. Therefore, while the general regularities are the same, quantitative estimates of physical properties may vary between different studies.

There are many technical parameters that affect the interaction of the deposited $CaPO_4$ particles with plasma jet and substrate; hence, the properties of the deposition. Those parameters include feedstock type, plasma gas composition, current, flow rate, energy input, torch offset distance, substrate cooling, etc. [232–234]. Ideally, only the thin outer layer of each powder particle should be heated to a molten plastic state where both chemical changes and phase transitions inevitably occur. This plastic state is necessary to obtain a dense and adhesive coating, but should be negligible as a volume fraction of the particles. By choosing the optimal relationship between the processing variables, $CaPO_4$ can be deposited with the desired thickness and crystallinity [217,221,222,235].

It was also found that the size of deposited $CaPO_4$ particles affects their melting properties in the plasma flame. That is, larger particles melt in the plasma flame to a lower degree than smaller particles [236–238]. Namely, during sputtering of HA particles with sizes exceeding ~55 μm, they were found to remain crystalline and showed little melting during plasma sputtering; HA particles between 30 and 55 μm were partially melted and consisted of a mixture of crystalline and amorphous (ACP) phases, while HA particles below ~30 μm were completely melted and contained large amounts of ACP and trace amounts of CaO [236]. Similar results were obtained in another study [238]. Schematically, both partially and completely melted splashes are shown in Figure 7 [239]. In yet another study, plasma spraying of 20–45 μm sized HA particles was found to produce more dense lamellar deposits than those obtained by plasma spraying of 45–75 and 75–125 μm sized HA particles. The deposits formed from 20–45 μm sized HA particles did not show the presence of voids but contained a smoother surface profile due to uniformly stacked disk-like spatters. In contrast, those composed of 45–75 and 75–125 μm sized HA particles contained a large number of unmelted particles, voids and macropores [237]. Interestingly, the roughness of the sputtered deposits can be used as a measure of the degree of particle melting. That is, as the deposited particles become more fluidized in the plasma flame, they become less viscous and can spread significantly over the substrate. This results in a smoother coating. Partially melted particles cannot be easily leveled on the surface. This leads to large ripples and coarse deposits [240,241].

A diagram of the formation of various phases during plasma sputtering of HA coatings is shown in Figure 8 [239]. According to the authors, if the outer shell of the HA particle is melted and the core remains unmelted, the heat transfer is insufficient to completely melt the particle. This model is modified to a fully melted hydroxyl-rich core, which further changes with heat transfer to the particles. The first condition describes a molten droplet with a hydroxyl-deficient skin. The core containing the hydroxyl-rich melt crystallizes during precipitation to form HA. By diffusion of the droplet, the dehydrogenated regions exposed to the substrate form ACP phases, while regions remote from the substrate crystallize to form OA. OA requires smaller atomic rearrangements to crystallize from the viscous melt and therefore crystallizes preferentially as a TCP and TTCP mixture. The growth starts in the hydroxyl-rich core and eventually transforms into OA in response to the depleted hydroxyl concentration in the upper part of the lamella. As molten particles flatten with increasing cooling rates, the entire particle becomes amorphous; TCP and TTCP are observed more frequently with higher heat conduction to the particles. If heat dissipation through the already solidified amorphous and crystalline layers is slow, TCP and TTCP can nucleate on the upper surface of the lamella; the growth of TCP and

TTCP can slow the growth of OA by latent heat of fusion. As dehydrogenation in molten particles progresses, less HA and OA are produced, which means that a large amount of dehydrogenated material will contain more amounts of TCP and TTCP. If the droplet is fluid, the growth mechanism can start inside the droplet as diffusion is faster. When higher heating conditions are used, CaO is observed. In addition to the hydroxyl group deficiency, the outer shell of the molten particles also becomes phosphate group deficient [239,241]. Furthermore, a numerical simulation model of the behavior of HA powder in a plasma jet has been proposed [242]. In that model, the authors created a temperature field and estimated the phase composition within a single HA particle during plasma jetting (Figure 9). After the hot particles hit the substrate surface, the relatively clear phase separation shown in Figure 9 disappears [222]. Thus, a heterogeneous mixture of various compounds (e.g., ACP, α-TCP, β-TCP, HA, OA, TTCP, calcium pyrophosphate, calcium metaphosphate and CaO) is dispersed on a microcrystalline scale [242]. More details on this can be found in the subject reviews [225,226,243].

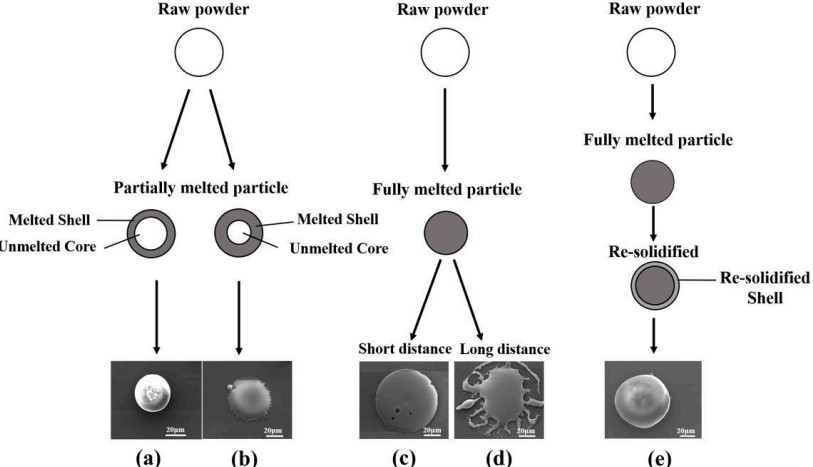

**Figure 7.** A schematic representation of splat type single particles during plasma spraying: (**a**) hemispherical shape splat; (**b**) oblate spheroidal splat; (**c**) disk; (**d**) splashed shape splat; (**e**) hemispherical shape splat. Reproduced with permission from [239].

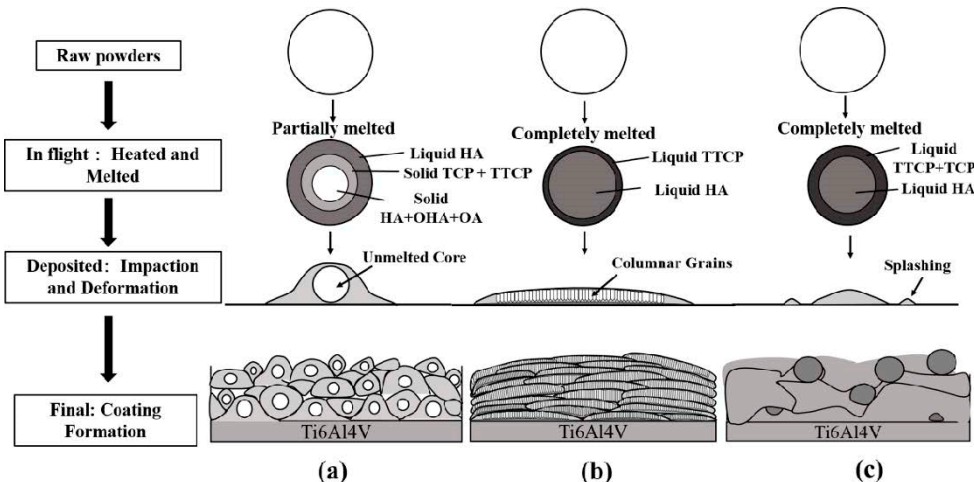

**Figure 8.** Phase formation in plasma sprayed HA deposits. The process steps show different melt chemistries as a function of particle temperature: (**a**) HA deposits contain many partially melted particles, (**b**) HA deposits with columnar structure, (**c**) HA deposits with a high amount of ACP. Reproduced with permission from [239].

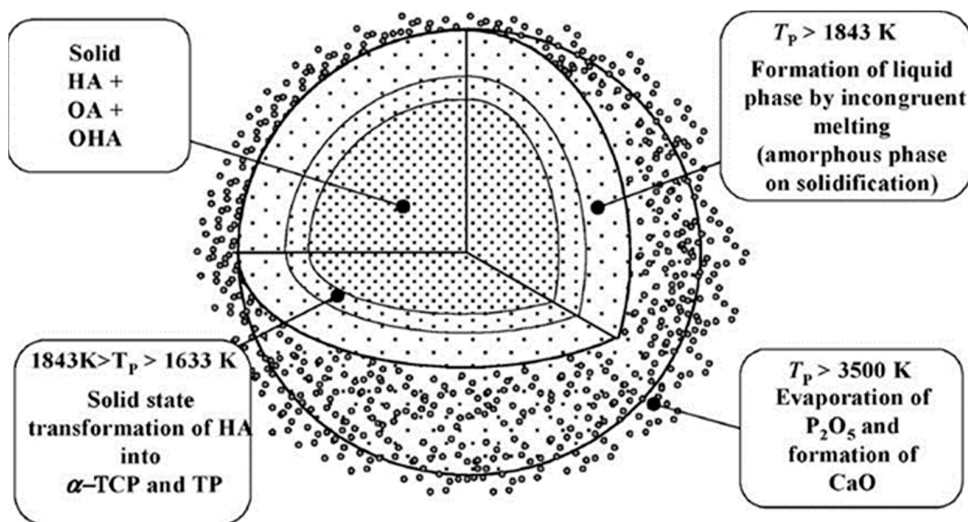

**Figure 9.** The temperature and assumed phase composition within a single plasma sprayed HA powder particle. Reproduced with permission from [242].

To finalize this section, plasma spray technology appears to be the most advanced CaPO$_4$ (mainly HA) deposition process on an industrial scale. Examples include PureFix™ HA (Stryker Howmedica Osteonics, NJ, USA), Duofix® HA (DePuy Orthopedics, NJ, USA), Corail® uncemented stems coated with HA (DePuy Synthes, NJ, USA) [244], BIOTAL® HA (Plasma Biotal, Derbyshire, UK), Osteotite® HA coated bone screws (Orthofix, TX, USA), Osprovit® HA (Eurocoating, Trento, Italy), Landos Corail threaded cup (Landanger, France), Endosteal dental screw implants (Interpore Cross International, CA, USA), Margron femoral stem (Portrald Orthopedics, Australia), AQB Implant System (Advance Co. Ltd., Tokyo, Japan), contract manufacturers Medicoat (Mägenwil, Switzerland) and Orchid (MI, USA), among others. More complex technologies are also known. For example, Zimmer Biomet's MP-1® HA coating (Zimmer Dental, Carlsbad, CA, USA) uses a pressurized hot water post plasma spraying process called MP-1 to convert plasma sprayed CaPO$_4$ deposits into non-crystalline HA and ACP components into crystalline HA [245].

Furthermore, Cam Bioceramics (Leiden, The Netherlands), Plasma Biotal (Buxton, Derbyshire, UK), Bioceramed (Loures, Portugal) and MedicalGroup (Véran, France), in addition to their CaPO$_4$-based business, also offer their customers contract manufacturers using plasma spiling to deposit CaPO$_4$ (mainly HA) on specific implants. The popularity of CaPO$_4$ plasma deposition technology is demonstrated by the fact that many manufacturers have adopted CaPO$_4$ plasma deposition technology.

4.1.2. High Velocity Oxy-Fuel (HVOF) Spraying

In the 1990s, a new thermal spray process named HVOF spraying was created [246,247]. Various types of mixtures of gaseous (e.g., acetylene, propylene, propane, H$_2$, CH$_4$, natural gas) or liquid (e.g., paraffin) fuel and oxygen are fed into a combustion chamber, where it is ignited and burned continuously. The resulting hot (up to 3000 °C) gas with a pressure of 1 MPa is ejected through converging and diverging nozzles, and passes through a straight section. The injection velocity at the barrel exit (>1000 m/s) exceeds the sound speed. A CaPO$_4$ powder is injected into the gas stream and its particles are accelerated to about 800 m/s. The hot gas and the powder stream are directed towards a substrate to be coated. The CaPO$_4$ particles are partially melted and deposited in splashes on the substrate. The resulting CaPO$_4$ have a low porosity and a high adhesion strength [246–251]. This deposition process is sometimes referred to as thermal printing [252]. Due to the high process temperatures, the deposited CaPO$_4$ are also a complex mixtures of various phases (HA, OA, TTCP, high-temperature ACP, α-TCP and β-TCP), and other compounds such as calcium pyrophosphate, calcium metaphosphate and CaO [223].

Like the plasma spray results, in the case of HVOF spraying, larger particles were found to melt to a lower degree than smaller particles [248,249]. In other words, cross-sectional SEM examinations of $50 \pm 10$ μm sputtered HA particles showed that they only partially melted from the surface, while examinations of $30 \pm 10$ μm HA particles showed almost complete melting. The melting rate can therefore vary from 20% to 100% depending on the particle size [249]. The morphology of $CaPO_4$ deposits shown in Figure 10 further demonstrates the influence of melting state on grain size. It clearly shows the interface region between the melted and unmelted portions within the HA splat. It indicates that the size of HA particles in the unmelted part is much larger than those in the molten part, explaining the effect of rapid cooling during coating formation on grain growth [248]. Furthermore, qualitative investigation of HA particles (partially molten) sputtered by HVOF by Raman spectroscopy showed that pyrolysis of HA occurs in the molten part rather than the unmelted part [249]. Therefore, careful selection of the appropriate powder size and proper HVOF spraying parameters is necessary to achieve high crystallinity of the deposit and reduce the amount of miscible phases.

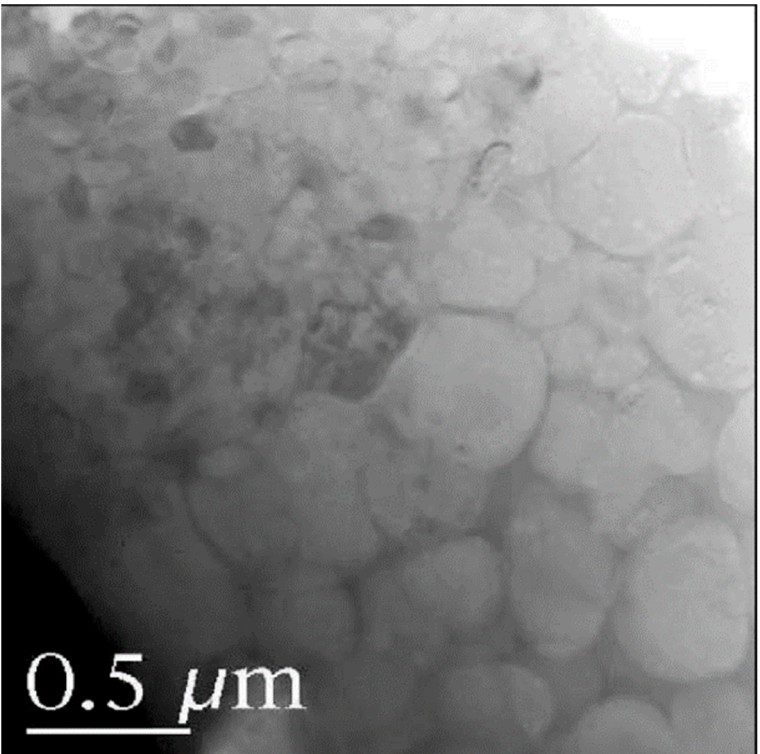

**Figure 10.** A TEM image of a sputtered HA coating, showing the interface between the molten and unmelted portions within the HA splat, and the different grain sizes. Reproduced with permission from [248].

At the end of this section, mention should be made of the existence of the High Velocity Suspension Flame Spraying (HVSFS) process, developed at the Institute for Manufacturing Technology of Ceramic Components and Composites at the University of Stuttgart, Germany, which appears to be an improvement on HVOF spraying. As the name suggests, a $CaPO_4$-based suspension is sprayed [253–255]. In addition, some studies have used lower velocities (~300 m/s). These techniques have been called "low velocity oxyfuel (LVOF)" [256] or "flame spraying" [257]; in the case of LVOF spraying, the specific energy and power consumption required to melt the $CaPO_4$ powder is lower than in HVOF; thus, LVOF is considered more economical.

### 4.2. Vapor Deposition Techniques

All vaporization techniques are broadly classified into two categories: physical vapor deposition (PVD) and chemical vapor deposition (CVD). Furthermore, PVD techniques are classified into (1) thermal evaporation, in which the material is heated until its vapor pressure becomes higher than the ambient pressure, and (2) ion sputtering, in which a high-energy ion or electron beam is applied to a solid target to pluck atoms from its surface [32,179]. PVD is a flexible deposition method, and atomic-scale or nanometer-scale thin film materials can be deposited with structure control. PVD is usually performed in a vacuum but can also be performed in the presence of various gases. The target is the material to be deposited (in this case $CaPO_4$). The substrate is placed inside a chamber and evacuated to a predetermined pressure. The sputtering proceeds when ions and atoms from the target collide and exchange momentum. The displaced molecules and atoms are then deposited on a substrate placed in the vacuum chamber. A key advantage of the sputtering is that materials with very high melting points can be easily deposited. For effective momentum transfer, a sputtering gas must have an atomic weight close to that of the target; Ne or Ar is used to sputter light elements; Kr or Xe is used for heavy elements [258]. However, oxygen can also be used for deposition of $CaPO_4$ bioceramics. Oxygen has several properties, one of which is better stoichiometry with respect to the HA of the deposited coating, film or layer [259].

Various types of techniques are used to sputter materials, including ion beams [260–269], radio frequency (RF) magnetrons [270–282], pulsed lasers [61,164,283–298], electron cyclotron resonance plasma [299–302], diode, direct current, reactive sputtering or deposition [303,304]. However, since all types of $CaPO_4$ belong to electrical insulators, the latter three sputtering techniques cannot be used; besides, the physical and collective state of the $CaPO_4$ source can affect the deposition rate. Namely, deposition rates of HA appeared to be much higher on solid plate targets than on powder pellet targets due to their apparent densities of 75% and 18%, respectively [273].

Depending on the processing parameters and type of sputtering equipment, the chemical composition and structure of the deposited $CaPO_4$ can differ significantly from the initial material used for sputtering. It is suggested that the difference in the Ca/P ratio between the initial $CaPO_4$ and the sputtered ones is likely due to preferential sputtering of calcium as orthophosphate ions can be pumped out before they are deposited on the substrate [305]. It has also been suggested that orthophosphate ions may be sputtered by incoming ions or electrons as they are weakly bound to the growing sediment [271]. However, all spraying techniques have the advantage of being able to place fine and compact coatings with strong adhesion.

#### 4.2.1. Ion- and Electron-Beam Assisted (IBAD and EBAD) Depositions

IBAD is a PVD technique performed in a vacuum, where ions of the deposition material (in this case $CaPO_4$) are generated from a source and deposited by impinging electrons. It is therefore also called EBAD. The sources can be both single-target and dual-target. In the former, a $CaPO_4$ source is vaporized with a single electron gun [306], while in the latter it is an independent method in which Ca-containing (e.g., CaO) and P-containing (e.g., $P_2O_5$) sources are vaporized simultaneously with two electron guns (dual ion beam system) or an electron gun and resistance heater, respectively. CaO has a high melting point of 2572 °C and a relatively low vapor pressure, making it suitable for electron gun evaporation, while $P_2O_5$ has a low melting point of 500 °C and a high vapor pressure, making resistance heater evaporation possible. The latter is known as "simultaneous vapor deposition (SVD)" and allows fine-tuning of the Ca/P ratio of the deposit by selecting optimal parameters [307–310]. In all cases, however, the outgoing ions are accelerated towards the target by an electric field emitted from the grid. After ions leave the ion source, they are neutralized by electrons from a second external filament, forming neutral atoms. The pressure gradient between the ion source and the sample chamber is created by a gas inlet in the ion source, which is injected into the sample chamber through a tube [311]. A

typical deposition system therefore consists of an electron or ion beam that vaporizes a $CaPO_4$ bulk target to produce a cloud of elements directed towards the substrate surface, and a source that simultaneously irradiates the substrate with high-energy inert (e.g., $Ar^+$) or reactive (e.g., $O_2^+$) gas ions to aid deposition. This technique, which consists of two main parts, is called "ion beam dynamic mixing" when $Ca^+$ ions are used for irradiation [312,313]. Excellent illustrations of the IBAD technique can be found in these references [32,179,181].

In IBAD, a thin (several hundred atomic layers thick), amorphous (ACP) deposit is first produced. It is then crystallized by ion implantation with $Ar^+$, $N_2^+$, $O_2^+$ and other ions [260–269]; the high bond strengths associated with IBAD technology appear to be a result of atomically mixed interface layers with thicknesses down to a few microns. Studies have shown that the chemical composition of ion-beam assisted deposits can be modified. For example, β-TCP was evaporated with an electron beam and $CaPO_4$ were deposited on silicon wafers both with and without simultaneous $Ar^+$ ion beam irradiation. The results confirmed that simultaneous $Ar^+$ irradiation has a significant effect on both the morphology (Figure 11) and composition of the deposits. Namely, the ones formed without $Ar^+$ irradiation had a Ca/P ratio of ~0.76 and reacted with moisture in the air immediately after they were removed from the chamber, while deposits formed with $Ar^+$ bombardment had a Ca/P ratio of ~0.80 and a smooth, featureless surface topography [264]. The effect of $Ar^+$ ion beam current on the binding strength of the coating and its dissolution in physiological solutions was investigated in another study. With increasing current, the bond strength between the substrate and the coating increased, but the dissolution rate in physiological solutions significantly decreased [262].

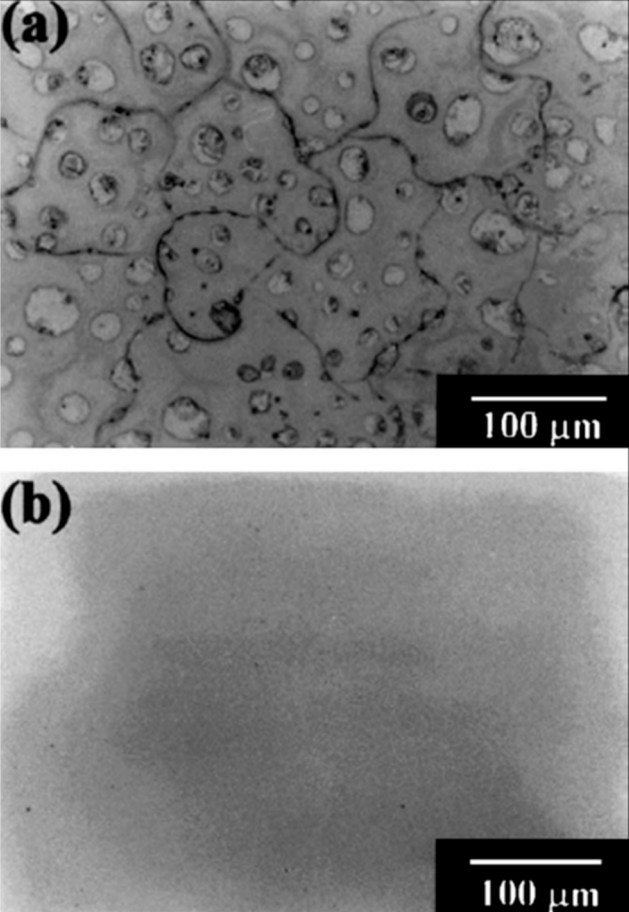

**Figure 11.** Two optical micrographs of $CaPO_4$ deposits on Si wafers: (**a**) without ion beam irradiation, (**b**) with $Ar^+$ ion beam irradiation (120 V, 0.8 A). Reproduced with permission from [264].

In one more study, $CaPO_4$ were deposited onto silicon using simultaneous double ion beam deposition. In this method, electron beam heaters and resistance heaters vaporized $P_2O_5$ and CaO, respectively, while $Ar^+$ ion beams were focused on the substrate to aid deposition. All deposits were found to be amorphous (ACP) regardless of the current density level of the ion beam. Therefore, a heat treatment was applied to crystallize them. The effect of ion beam current density on the phase composition of crystallized $CaPO_4$ is shown in Figure 12. The Ca/P ratio was found to increase with increasing ion beam current density. This is probably due to the faster sputtering of $P_2O_5$ compared to CaO from the layers to be coated; as seen in Figure 12, a biphasic (HA + TCP) mixture was observed when the ion beam was used without or at a current density of 180 mA/cm$^2$, while only the HA peak was observed at an ion beam current density of 260 mA/cm$^2$ [308]. In another study, X-ray photoelectron spectroscopy (XPS) analysis of FA deposited on titanium revealed several distinct regions: (i) The environmentally exposed surfaces showed high carbon concentrations due to air pollution. (ii) The bulk region contained relatively constant concentrations of Ca, O, P and F, indicating chemical reactions of calcium fluoride ($CaF_2$) and FA formation. (iii) The substrate region showed high concentrations of Ti and O photoelectron peaks, indicating the coexistence of $CaPO_4$ in titanium oxide [261]. Other researchers have also found similar region structures [263]. Furthermore, when cross-sections of functionally graded HA deposits on silicon substrates obtained by dual IBAD and simultaneous heat treatment were examined, microstructural analysis of the deposits showed a gradual decrease in grain size and crystallinity towards the surface, leading to nanoscale grains and finally an amorphous layer on the surface [266].

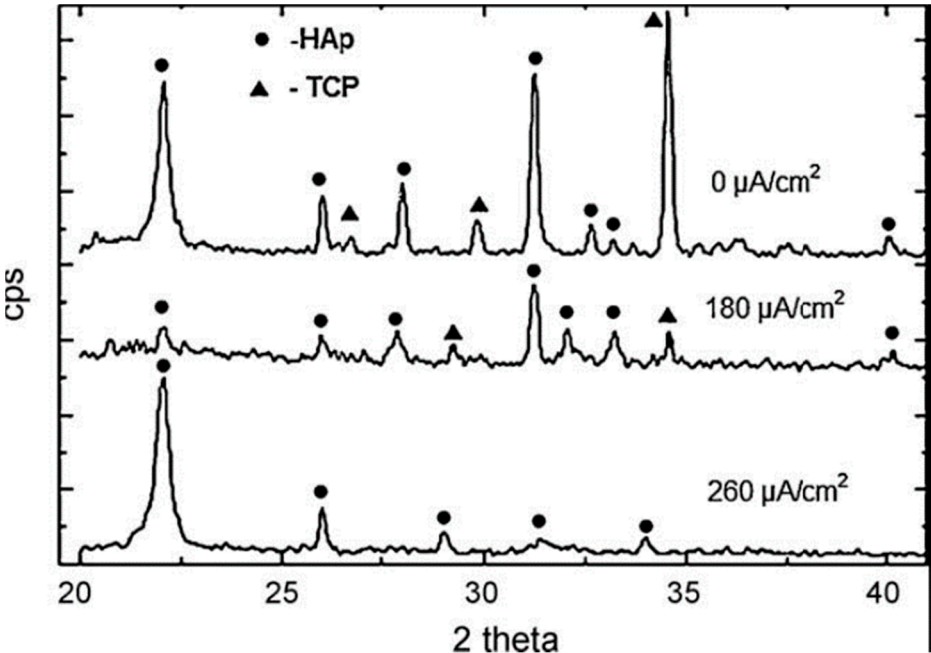

**Figure 12.** XRD patterns of fully crystallized (after a heat treatment at 1200 °C) $CaPO_4$ coatings sputtered at three different ion beam current densities. Reproduced with permission from [308].

Moreover, the substrate temperature was found to be important as well. Namely, a series of multilayer $CaPO_4$ deposits were produced on Ti substrates being heated to various temperatures such as 650 °C, 550 °C and 450 °C. As seen in Figure 13, the $CaPO_4$ deposits consisted of three different layers, distinguished by crystalline features corresponding to the various substrate temperatures during deposition [268]. Further details on the IBAD technique are available in the literature.

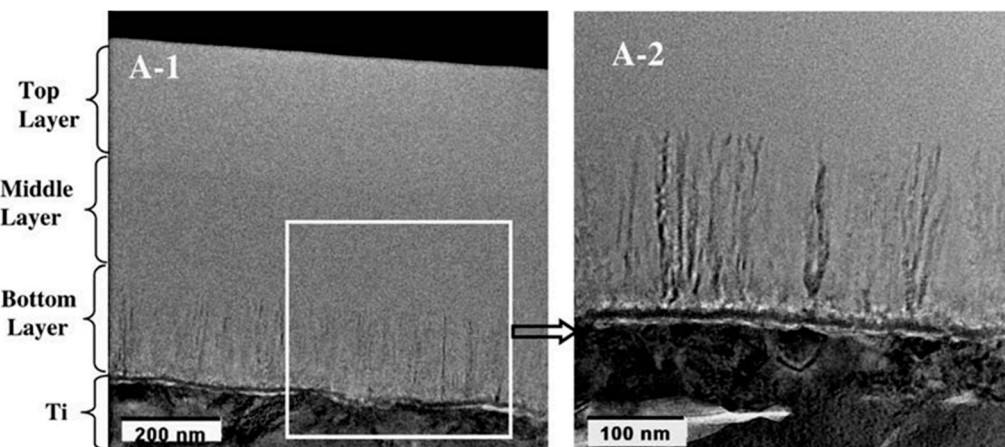

**Figure 13.** TEM images of cross-sections of CaPO4 deposits on Ti substrates heated to various temperatures: 650 °C, 550 °C and 450 °C. (**A-1**) is an image of a cross-section containing both Ti and the coating, while (**A-2**) is a higher magnification image of an area showing only the interface and the crystal layer. The ~190 nm thick layer closest to the substrate was deposited at 650 °C. This layer shows a columnar crystal structure with 4–30 nm wide crystal grains perpendicular to the interface. The middle semi-crystalline layer with a thickness of ~226 nm was deposited at 550 °C and the upper amorphous layer with a thickness of ~278 nm was deposited at 450 °C. Reproduced with permission from [268].

### 4.2.2. Pulsed Laser Deposition (PLD)

With a discovery of lasers in the late 1950s [314], researchers began to focus beams on materials to observe their interactions. Therefore, the PLD (synonym: laser ablation deposition) technique for making thin films became increasingly popular with the advent of lasers delivering nanosecond pulses in the 1970s [315]. However, the first papers on PLD of $CaPO_4$ were published in 1992 using excimer and ruby lasers, respectively [283,284].

PLD is the thin-film version of PVD technology, where a high-power laser is used as an energy source to melt and vaporize the target (in this case $CaPO_4$) material in a vacuum environment and condense its vapors on a substrate placed parallel to the target holder. Due to the high-power density of the focused pulsed laser, the ablated material thermally decomposes to form a plasma plume consisting of a gas cloud of highly excited molecules, molecular clusters, atoms, ions, electrons, and in some cases droplets or target fragments [294]. For ArF laser ablation of HA a comprehensive investigation of plasma plume expansion is well described in the literature [316]. Rotating the target prevents crater formation, rapid erosion of the target surface and plasma plume deflection. Therefore, the target is rotated during deposition to achieve a constant ablation rate. If the laser energy density is high enough, each laser pulse vaporizes or ablates a small amount of material, which is emitted from the target as a highly forward-directed plume. This plume becomes a flux of material deposited on the substrate. The substrate is usually placed at a fixed distance parallel to the target on a heated substrate holder. One of the main advantages of PLD technology is that the stoichiometry of the target can be maintained in the deposit. This is due to the simultaneous vaporization of all compounds and elements due to the high ablation rate.

Numerous schematic diagrams of experimental setups for PLD technology can be found in the literature [32,179,181,298,304]. The most basic setup consists of a vacuum chamber, a laser source, a focusing lens, a motorized target holder, a substrate placed on a heater to increase film adhesion and promote crystallinity, and a pump system. The substrate is usually placed in a plane parallel to the target with a distance of 2–10 cm between them. Ultraviolet (UV) excimer lasers with pulse durations of ~10 ns and power densities within 10–500 $MW/cm^2$ are commonly used for ablation. Methods using two laser beams (so-called laser-assisted laser ablation methods) are also used. In the latter

case, one laser beam of a KrF laser (ablation laser) is used to ablate the HA target. The other ArF laser beam (auxiliary laser) is used to irradiate the Ti substrate surface during deposition. The auxiliary laser is important for the formation of crystalline HA deposits and the improvement of the adhesion strength to the Ti substrate [317].

The PLD process can be carried out in ultra-high vacuum and in the presence of background gases such as oxygen, which is commonly used to fully oxygenate the deposited material for depositing oxides. Ar [318] and water vapor [319] can also be used. In addition, various types of lasers with different pulse repetition rates, energy streams, wavelengths, etc. are used. All these processing variables have been found to affect the structure, composition, properties, crystallinity, crystallographic orientation, morphology and roughness of the deposited $CaPO_4$. For example, a significant difference was found between Nd:YAG and KrF lasers and $CaPO_4$ deposits: KrF deposits have a columnar structure, while Nd:YAG ones are granular [287,288]. This example clearly shows that the mechanism of laser ablation of $CaPO_4$ is very complex and it is described in reference [320].

A PLD method is characterized by high particle flux densities (up to $10^{20}$ cm$^{-2}$s$^{-1}$) and thus high condensation velocities (0.01–1.0 Å per pulse) and kinetic energies of 10–200 eV for particles in the flux [292]. Moreover, local temperatures reach ~$10^4$ K due to the laser vaporization mechanism, where an accelerated (>$10^6$ cm/s) flux is generated from the plasma resulting from the interaction between the laser beam and the target surface. These features determine the structure and properties of the deposit: the presence of amorphous phases (ACP), high hardness and bond strength, and the reproduction of the initial elemental composition of the target in the deposit. Mass transfer with preservation of composition is due to the high heating rate and thermal ablation of the target in the laser spot region [320].

The PLD process is used to deposit thin (0.05–5 μm) coatings of $CaPO_4$ on various substrates [61,164,283–298]. The method includes ablation of a $CaPO_4$ (usually HA) target using pulsed (typically 30 ns, 120 mJ pulses at a repetition rate of 10 Hz) KrF excimer laser light (λ = 248 nm) in a 0.3 Torr/$H_2O$ atmosphere and the removed HA material is heated (400–800 °C) for deposition on the substrate; the deposition rate of PLD is approximately 0.02–0.05 nm per laser pulse [8]. The effect of high laser fluence (2.4 J/cm$^2$ to 29.2 J/cm$^2$) on the properties of $CaPO_4$ deposits was investigated. The deposits produced at 2.4 J/cm$^2$ appeared to be partially amorphous (ACP) and had a rough surface containing many droplets. On the other hand, higher laser fluence was found to result in higher crystallinity and lower surface roughness. Moreover, higher laser fluence decreases the Ca/P ratio of the deposited films and possibly increases their density [295]. Substrate heating is necessary to guarantee the formation of highly crystalline and phase-pure deposits. Furthermore, the substrate temperature can be varied to form deposits with the desired fine texture and roughness [143,296,321].

PLD is usually performed at elevated temperatures of substrates, which can result in the formation of oxide layers on the surface of substrates prior to $CaPO_4$ deposition and thus affect its adhesion to the substrate [286]. Like thermal spraying processes, PLD of $CaPO_4$ always results in formation of complex mixtures of various compounds, including additives (e.g., CaO, calcium pyrophosphate) and amorphous phases [285,322]. Among the known $CaPO_4$ compounds, HA, OCP (the process was carried out in hot steam flow) [289], calcined bone [323] and biphasic formulations such as HA + TTCP [324] have been deposited by PLD technique. It was also found that TTCP in the PLD deposits was not formed by partial conversion of previously deposited HA. Instead, it was formed by nucleation and growth of TTCP itself from the ablation products of the HA target [324]. Furthermore, $CaPO_4$ deposited by PLD can be of various morphologies (e.g., granular and columnar) that differ in their resistance to exfoliation [285]. Moreover, various types of oriented structures can be produced [290,293,325]. Finally, it should be noted that various process improvements have also been introduced, such as in situ UV-irradiation-assisted PLD [164], discharge-assisted PLD [326], high-frequency plasma-coupled PLD [327] and eclipsed PLD [328] techniques.

Additional information on the PLD of $CaPO_4$ is available in the references [181,292,304,329].

### 4.2.3. Magnetron Sputtering

Magnetron sputtering is a third example of PVD technology. As far as I have found, the idea of depositing $CaPO_4$ using a magnetron was first introduced in the USA. On 11 October 1975, a patent (filed on 13 May 1974) was issued devoted to "a system of coating prostheses with ground bone particles" [330]. The patent describes the deposition of bone particles and there is no need to explain that bone particles are mainly composed of $CaPO_4$ of a biological origin. However, the first deposition of $CaPO_4$ by sputtering took place in 1992 [270].

Magnetrons are high-power vacuum tubes that generate microwaves through the interaction between electron flow and a magnetic field. Magnetron sputtering originated in the mid-1960s [331] and is considered a high-rate vacuum coating technique for depositing miscellaneous compounds up to 5 µm thick on various materials (Table 3). The process takes place in ambient conditions. The sputtering system consists of a magnetron, a wave generator, a vacuum chamber, a cooling system, a target and a substrate. A specially shaped magnetic field is applied to the sputtering target. When the substrate is placed in the vacuum chamber, the target $CaPO_4$ material is released into the chamber as a gas. A strong magnet ionizes the particles of the target $CaPO_4$. The negatively charged target material is then aligned on the substrate to form a deposit [332]. A deposition system involving two opposing magnetrons has also been proposed. The authors confirmed that when two opposing magnetrons are arranged such that the magnetic fields are additive to each other, low substrate recoil and high deposition rates can be achieved simultaneously due to the right-angled geometry. Using this system, HA thin films with desired stoichiometry and crystallinity could be produced by sputtering without the need for annealing [333]. Furthermore, a multi-target system was proposed to deposit $CaPO_4$ with varying Ca/P ratios. The authors prepared three independent $CaPO_4$ targets, each compacted with DCPA, $\alpha$-TCP and HA powders in various combinations, in a set of three high vacuum sputtering sources mounted at 75° to the substrate surface normal at a distance of 100 mm from the substrate holder. These were fed into the system. As a result, $CaPO_4$ deposits with Ca/P ratios of 1.82, 1.72, 1.47, 1.43 and 1.0 were formed after annealing [279]. Experimental details on preparation of HA targets are available elsewhere [334].

Magnetron sputtering can be performed in RF (radio frequency) or DC (direct current) modes, but RF mode is only used for $CaPO_4$ deposition as DC mode is only possible with conductive materials. Typically, RF mode uses a sinusoidal generator operating at 13.56, 5.28 and 1.78 MHz. Discharge power, operating pressure, substrate temperature, gas flow rate, deposition time, thermal post-treatment and negative substrate bias are the key parameters directly affecting the integrity and quality of $CaPO_4$ deposits [181]. Namely, the deposition rate increased with rising Ar gas pressure up to 2 Pa but significantly decreased with increasing pressure up to 5 Pa, with simultaneous decreasing of the Ca/P ratio at higher Ar gas pressures [335]. When the negative substrate bias, deposition time and RF power were varied, the Ca/P ratio changed from 1.53 to 3.88 and the structure of the deposited material was found to become crystalline or amorphous [276]. A schematic setup of the magnetron sputtering equipment can be found in the literature [32,179,181].

Magnetron sputtering is currently a viable method to deposit $CaPO_4$ [270–282]. The advantages include high deposition rates, excellent adhesion and the ability to coat implants with difficult surface geometries (Table 3). Namely, deposition rates of 0.29–1.75 µm/h have been found in various studies [276]. However, to routinely deposit pure, crystalline $CaPO_4$ on implants by magnetron sputtering, several issues such as durability and Ca/P ratio need to be resolved. Videlicet, both the mechanical properties and a microstructure of HA deposits on Ti-5Al-2.5Fe alloys produced by RF magnetron sputtering have been investigated [272]. Deposition was carried out from HA targets in low-pressure Ar or Ar+$O_2$ mixtures at substrate temperatures within 70–550 °C. Smooth (coarseness ~50 nm) and homogeneous $CaPO_4$ deposits were produced. Regarding the effect of temperature, the deposits grown at substrate temperatures below 300 °C appeared to be amorphous (ACP) and contained few crystalline phases. In contrast, the deposits produced at a

substrate temperature of 550 °C or grown at ambient conditions and afterwards annealed at 550 °C were composed of HA [272]. Similar results were obtained in another study [282]. Remarkably, regardless of the type of $CaPO_4$ source used (DCPA, $\alpha$-TCP and HA were used), the sputtered deposits were found to be composed of ACP and transformed into apatite-like structures after annealing at 500 °C after deposition, with Ca/P ratios of 1.57, 1.84 and 2.14, respectively [278]. Thus, the Ca/P ratios of the deposits are always higher than that of the initial target. Furthermore, RF magnetron sputtering can be combined with other deposition techniques. Namely, plasma-assisted RF magnetron co-sputtering technology has been used to deposit $CaPO_4$ on Ti-6Al-4V orthopedic alloys [336,337]. Magnetron sputtering of $CaPO_4$ onto $CaPO_4$ sub-coatings deposited by another method has been performed as well [338]. More details about this technology can be found in other reviews [181,339,340].

As a result, magnetron sputtered $CaPO_4$ deposits on implants are produced commercially. An example is the BioComp HAVD (Hydroxy Apatite Vapour Deposition) implant (BioComp Industries BV, Netherlands). Also commercially available is the μ-ONE HA implant (Yamahachi Dental, Japan), in which a 1–2 μm thick film of HA is sprayed onto the surface of a titanium implant dental fixation device (implanted in the maxilla during surgery) using an undisclosed technology.

### 4.2.4. Electron-Cyclotron-Resonance (ECR) Plasma Sputtering

ECR plasma sputtering is the fourth example of PVD technology but was initially considered as a CVD process [341]. It is an off-axis low-temperature deposition method in which raw materials are supplied by sputtering with microwave ECR-excited plasma and the plasma flow is attracted to the sample by a divergent magnetic field. Using the microwave ECR method, highly ionized plasma is generated and a plasma stream is drawn in at gas pressures as low as 0.001–0.1 Pa. The sputtered gas molecules and particles are ionized by ions in the plasma stream and transported to the sample substrate with an energy of 10–30 eV without the need for substrate heating. Fully oxidized, dense, high-quality deposits are obtained at room temperature [341].

A schematic of the ECR plasma sputtering system is shown in Figure 14a [342]. Briefly, a 2.45 GHz microwave (500 W power) was split into two waves of equivalent magnitude and transmitted to the plasma source through two facing quartz glass windows; Ar or Xe gas (flow rate 8 $cm^3$/min) was introduced into the plasma source and a magnetic field was generated simultaneously to produce ECR plasma. The plasma density was highest near the "ECR point" (Figure 14b). Afterwards the plasma was transported along the dispersive magnetic field to a cylindrical HA target. When an RF voltage (500 W) was applied to the target, Ar or Xe ions stroked the inner cylindrical surface of the HA target, and sputtered atoms and clusters accumulated on the substrate surface. The substrate was a Si(100) wafer that was not intended to be heated from the back side, but its surface temperature was raised to 70 °C by irradiation of the ECR plasma stream [342].

To produce cylindrical HA targets, HA powder was dispersed in a polyvinyl alcohol binder, formed into cylindrical shapes and sintered at temperatures between 400–800 °C. When the HA targets were placed in the sputtering chamber, the background pressure increased from $3 \times 10^{-5}$ to ~$10^{-3}$ Pa; it took more than a month for the background pressure to drop below $5 \times 10^{-5}$ Pa, which was a very long time. Stainless steel bottles filled with distilled water were connected to the deposition chamber through a variable leak valve; water vapor was fed through the variable leak valve during spraying to supplement oxygen to produce $PO_4^{3-}$ and $OH^-$ functional groups. The pressure range of water vapor was $10^{-4}$ to $10^{-1}$ Pa and the deposition rate was 1.3 nm/min. Crack-free ACP deposits with a thickness of 300–500 nm and smooth morphology were produced at room temperature and transformed into HA deposits when annealed in the presence of oxygen [299]. However, crack-free HA deposits with high hardness and specific crystallographic orientation were produced when Si(100) substrates were heated from the back surface to 450–500 °C using

Xe gas [300]. Similar results have been obtained with sapphire substrates [301]. Alas, the biomedical properties of such deposits remain unreported [299–302].

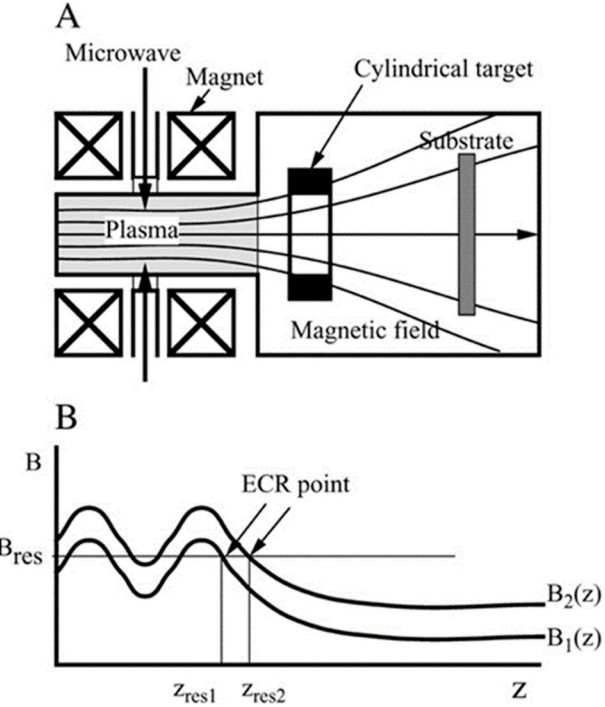

**Figure 14.** (**A**) Schematic of the ECR plasma spray system; (**B**) magnetic field strength as a function of distance from the plasma source tip. Reproduced with permission from [342].

In conclusion, productivity of ECR plasma sputtering technique is low. Therefore, it may be feasible in certain applications where epitaxial growth of thin $CaPO_4$ deposits with a specific crystallographic orientation is sufficient. Nevertheless, ECR plasma sputtering can be followed by additional deposition by a variety of more productive methods.

4.2.5. Metalorganic Chemical Vapor Deposition (MOCVD)

As the name suggests, MOCVD is a CVD technology. Historically, this technology was developed to place deposits in the semiconductor industry and has demonstrated several advantages, including the ability to adjust precursor concentration during deposition with the purpose of creating functionally graded deposits. According to MOCVD technology, a heated gaseous stream of a carrier gas + organometallic precursors (including reactive components of the coating material) is directed onto the surface of a heated substrate, on which a reaction occurs to produce a solid deposit. Simultaneously, reaction byproducts are pumped out. After the desired thickness is reached, the process is terminated [343].

For the first time $CaPO_4$ was deposited by MOCVD in 1996 [344], followed by other studies [345–351]. Deposition requires precursors containing Ca and P, and in many cases the inorganic metal precursors containing Ca have rather complex compositions. Examples include $Ca(tmhd)_2$ (where tmhd = 2,2,6,6,-tetramethylheptane-3,5-dione) [344], $Ca(hfpd)_2(tetraglyme)$ (where hfpd = 1,1,1. 5,5,-hexafluoro-2,4-pentadione) [350], calcium dibenzoylmethane [346] and $Ca(dpm)_2$ (where dpm = *bis*-dipivaloylmethanato) [347,349,351], but simpler compounds may also be used, such as calcium lactate [348,350]. Precursors containing P are $P_2O_5$ [344], trimethylphosphate [346,348,350] and triphenylphosphate [347,349,351]. Naturally, the precursors must be volatile at relatively low temperatures (<~600 °C), and pressures between 1 Torr and atmospheric pressure.

To perform MOCVD, precursors containing Ca and P are heated to evaporate and fed with a carrier gas (such as Ar) into a reactor in which a heated substrate is placed [344–351]. To accelerate oxidation of the precursor, oxygen may be separately introduced into the reactor and

mixed with the precursor vapor. Commonly, the substrate is heated to 600–800 °C. Depending on the Ca/P ratio, $\alpha$-TCP, HA or biphasic $\alpha$-TCP + HA formulations are deposited [347].

The deposited $CaPO_4$ are dense and homogeneous, but the grain dimensions of the deposits decrease as the substrate temperature decreases. Thus, the grain growth of $CaPO_4$ crystals was found to be faster at higher temperatures. This temperature dependence of the microstructure allows for the deposition of $CaPO_4$ with controlled surface properties. The main drawback of the MOCVD process is due to the precursors. That is, since organometallic precursors are used, carbon contamination can be a problem. Additional information on MOCVD is available in the literature [344–351].

### 4.2.6. Molecular Precursor and Thermal Decomposition Techniques

Since a thermal decomposition [352] and a molecular precursor [353–357] methods are similar, they have been combined in this section. Both technologies belong to CVD and have much in common with MOCVD, which was discussed earlier. That is, both techniques are based on the deposition of a mixture of Ca- and P-containing organic compounds with a desired Ca/P ratio (molecular precursors) on a substrate, followed by drying, firing and/or sintering. High temperatures burn off the organic components and the remaining Ca and P oxides combine to form solid $CaPO_4$ deposits [353–357]. The difference between MOCVD and these techniques lies in the substrate temperature. In these methods, the substrate is coated at room temperature, followed by sintering to decompose the Ca- and P-containing precursors, whereas, in MOCVD, the substrate is heated, so formation of $CaPO_4$ deposits occur in situ during deposition. Since the molecular precursor method is a wet process, even complex geometries can be coated.

To deposit HA, calcium 2-ethylhexanoate was stoichiometrically mixed with bis(2-ethylhexyl)phosphite in ethanol. The mixture was stirred for 2 days under ambient conditions, followed by dip coating. The coated substrates were then air-dried and calcined at 1000 °C in air [352]. Similarly, to deposit carbonate-containing CDHA, solid dibutylammonium metaphosphate salt was added to an ethanol solution of Ca-ethylenediamine-N,N,N′,N′-tetraacetic acid/amine complex (Ca-EDTA complex) in an amount that gave a Ca/P ratio of 1.67 [353–357]. The prepared molecular precursor solution was then dropped onto the Ti surface to cover the entire area of the substrate. The substrate was then dried at 60 °C for 20 min and then sintered at a temperature exceeding 500 °C for 2 h under atmospheric pressure. After calcination, carbonate-containing CDHA with a thickness of 0.44 μm was formed. After immersion in phosphate buffer solution (PBS), tensile bond strength measurements and scratch tests revealed excellent adhesion of the $CaPO_4$ deposits [353,355]. Subsequently, in vivo experiments showed that coated implants exhibited significantly higher bone contact rates compared to uncoated implants [354–356]. Cell-based experiments have also shown good results [358].

### 4.3. Wet Techniques

As can be seen from this section sub-title, all types of wet deposition techniques occur from aqueous and non-aqueous solutions or suspensions at moderate temperatures [359]. Depending on the solution pH, various $CaPO_4$ (Table 1) may be formed and deposited. Generally, the precipitation process usually occurs by means of heterogeneous nucleation, the kinetics of which depends on many parameters, including temperature, solution supersaturation, reagent concentrations, hydrodynamics, and the presence of nucleating agents, inhibitors and/or other admixtures. Regarding the precipitation mechanism of $CaPO_4$ from aqueous solutions, the entire process is rather complex and, for biologically relevant $CaPO_4$ (HA, CDHA, OCP), it seems to occur via the formation of one or several precursor $CaPO_4$ compounds such as ACP, DCPD/DCPA and OCP [360,361]. To get further details on this specific topic, the interested reader is referred to the specialist literature [362–364].

Some types of wet techniques seem to require specific surface treatments. Namely, for biomimetic deposition of $CaPO_4$ on Ti and its alloys, a surface layer of titanium oxide, hydroxide or titanate must be created prior to deposition [365]. This is done by thermal [366]

or alkali [52,155,367,368] treatments, oxidation with $H_2O_2$ [155,368], micro-arc oxidation (MAO) [369], pre-calcification in boiling $Ca(OH)_2$ solutions [370,371], under hydrothermal conditions [372] or using a water steam treatment [373], as well as by other oxidation techniques. The same is valid for other chemically inert metals such as Zr, Nb and Ta: prior to the biomimetic deposition of $CaPO_4$, surface layers of zirconium hydroxide, niobium hydroxide, tantalum hydroxide or their Na- or K-containing salts should be formed on their surfaces [374,375]. Regarding $CaPO_4$ deposition on Mg and its biodegradable alloys, detailed information on the surface treatment can be found in the literature [43]. For polymers, prior to $CaPO_4$ deposition by the wet method, the surface of polyetheretherketone substrates was treated with 98 wt% $H_2SO_4$ followed by $O_2$ plasma generated by a glow discharge system [376,377]. It is also known that the presence of hydrated silica [378], sodium ions [379] or both (i.e., sodium silicate) [380] has a positive effect on $CaPO_4$ precipitation onto substrates. Additional details on this topic can be found in the literature [32,381–383], among which the newly growing integrated layer strategy [383] is outlined.

4.3.1. Electrophoretic Deposition (EPD)

As written in Wikipedia: "Electrophoretic deposition is a term for a broad range of industrial processes, which includes electrocoating, e-coating, cathodic electrodeposition and electrophoretic coating or electrophoretic painting" [384]. The process is characterized by the migration (electrophoresis) of charged colloidal particles suspended in a liquid under the influence of a DC electric field and their deposition on an oppositely charged conductive substrate. This entire process involves mass transport, accumulation of particles near the electrode and agglomeration of particles to form a deposit [385]. To my knowledge, the idea of using electrodeposition to deposit $CaPO_4$ was first proposed in the United States. Explicitly, on 1 June 1975, the following patent (filed on 16 April 1974): "a method of improving orthopedic implant materials by the simultaneous electrodeposition of bone and collagen onto a prosthesis" was issued [386]. The patent describes a deposition of bone particles, and it is needless to explain that the bone particles contain biogenic $CaPO_4$ as a major component. Thus, electrodeposition is the oldest technique for depositing $CaPO_4$.

EPD is designed to apply the material to any conductive surface and is therefore used to deposit $CaPO_4$ only onto metal substrates. This method is particularly useful for porous metal structures. Deposition involves suspending $CaPO_4$ powder in water, alcohol or other suitable liquid to create a coating bath, which is then deposited on the metal surface [387–396]. Water is not recommended as a dispersion medium for $CaPO_4$ because the suspension is not stable without the aid of a dispersing agent and precipitates quickly. Thus, alcohols are often used instead. Butanol is preferred over ethanol because it reduces the evaporation rate and subsequently reduces cracking of the $CaPO_4$ precipitate when it dries. In addition, the appropriate particle size is very important because the powder must remain suspended during deposition. EPD usually involves submerging a metal substrate in a container holding a coating bath and applying electricity using electrodes, with the substrate being one of the electrodes (anode or cathode). The driving force of deposition is the applied electric field [385]. Depending on the mode and sequence of the applied voltage, EPD of $CaPO_4$ can be performed with either alternating current (AC) or DC fields [393] and with either constant [391] or dynamic [392,394] voltage. Of these, AC deposition results in a denser and less cracked coating compared to DC deposition at similar thicknesses [393], while dynamic voltages are suitable for placing gradient deposits [392].

After deposition, the substrate is typically rinsed to remove the undeposited bath and then sintered at 850–950 °C in a high vacuum ($10^{-6}$–$10^{-7}$ Torr) [385]. The resulting deposit consists of several $CaPO_4$ phases and various random admixtures. For example, in the case of deposited CDHA, sintering transforms it into a biphasic (HA + β-TCP) coating [387]. Its thickness can be varied by changing both the field intensity and the deposition time. In addition, various metal–phosphorus compounds may be formed at the coating/substrate interface due to interdiffusion of $CaPO_4$ and metal substrate atoms. Unfortunately, densification during sintering can cause shrinkage and cracking of the

deposit. In addition, thermal stresses caused by differences in the coefficient of thermal expansion of the core, and the deposit during sintering and cooling can lead to cracking [8].

A morphology of the deposited $CaPO_4$ depends on the applied voltage [390], deposition time [392] and powder concentration [393]. That is, at 200 V dimensions of the deposited particles ranged from 0.20 to 0.35 µm, at 400 V the particle size range increased from 0.35 to 0.80 µm and at 800 V the particle size range increased from 0.80 to 1.20 µm. Furthermore, the amount of deposited $CaPO_4$ increased with increasing voltage. Furthermore, porous and coarse deposits were obtained at higher electric fields, while denser deposits with finer particle size were obtained at lower electric fields. A similar effect was found for the deposition time, with shorter times resulting in smaller particles deposited [392]. With respect to the powder concentration in suspension, at low HA concentrations, the deposits were very coarse and showed large levels of agglomeration. At higher HA concentrations, they were uniform and crack-free, with less agglomeration. At very high HA concentrations, many cracks were seen [393]. Thus, deposition time and powder concentration, as well as applied potential, appear to influence the coating morphology.

This is a bit off subject, but certain types of $CaPO_4$ bioceramics may be prepared by EPD [388,389,397,398]. Namely, hollow HA fibers of various diameters were prepared. First, submicron HA powder was deposited on individual carbon fibers, their bundles and felts. The carbon substrates were then oxidized and calcined to leave the corresponding $CaPO_4$ replicas [388]. Similarly, uniform HA tubes were fabricated by EPD of HA powder on carbon rods, repeated deposition at room temperature and subsequent calcination [389]. In addition, the same approach was used to fabricate both porous $CaPO_4$ scaffolds [397] and coatings [399]. In addition, various modified and hybrid techniques have also been developed, including plasma-assisted EPD [400], and combinations of micro-arc oxidation (MAO) and EPD [401]. Additional information on the EPD of $CaPO_4$ is available in references [402,403].

### 4.3.2. Electrochemical (ECD) or Cathodic Deposition

As can be seen from the Wikipedia definition (see previous section), ECD (also called electrochemically assisted deposition [97,404]) appears to be a sub-division of EPD [384]. It uses electrical energy from a generator to trigger a series of chemical reactions in aqueous solution that result in $CaPO_4$ deposition on conductive surfaces. To accomplish this, a supersaturated or metastable aqueous solution containing both calcium and orthophosphate ions is used. The solution is electrified using either a two-electrode system (galvanostatic ECD) or a three-electrode system (potentiostatic ECD). Typically, a platinum electrode (anode) and a metal implant (cathode) are connected to a current generator. The process is based on various electrochemical reactions that occur in the electrolyte, resulting in an increase in the pH of the solution around the cathode. This causes the acidic orthophosphate ions $HPO_4^{2-}$ and $H_2PO_4^{-}$ to change to $PO_4^{3-}$, simultaneously producing $CaPO_4$ with a low solubility. The electrochemical reactions among ions during $CaPO_4$ deposition are described in the literature [405–408]. After the solution supersaturation reaches a critical value, $CaPO_4$ crystals nucleate and grow on the cathode to form deposits. Thus, ECD occurs in three successive steps: electrochemical reactions, pH rise and $CaPO_4$ deposition [407–411].

The crystal growth mechanism of $CaPO_4$ deposits on Ti by ECD was studied [407]. It was found that the mechanism changed during precipitation, resulting in diverse morphologies depending on the precipitation stage. Namely, in the first stage ($t$ = 1 min), the electrolyte was highly supersaturated and the first $CaPO_4$ deposits formed consisted of randomly oriented and highly branched nanoplates of CDHA. In the second stage ($t$ = 3 min), crystal growth progressed along the $b$- and $c$-axes and micro-sized plates were formed. In the third stage ($t$ > 10 min), the supersaturation of the electrolyte decreased and the deposited crystals propagated along the $c$-axis to form ribbon-shaped single crystals [407].

The ECD process can be performed under constant current (galvanostatic), voltage or potential (potentiostatic) modes, while the choice of parameters strongly affects both the morphology and phase composition of the prepared deposits. However, the use of high

current densities leads to the formation of large amounts of $H_2$ bubbles in the vicinity of the cathode, resulting in the formation of non-uniform and weakly adherent coatings [412]. Although DC current has been in common use for more than 20 years, pulsed current [413–421], pulsed countercurrent [422] and cyclic voltammetry [417,418,423] deposition techniques have been proposed to overcome these problems. Since ECD always occurs on negatively charged electrodes, it might be referred to as cathodic deposition [83,424,425]. The earliest paper on the ECD of $CaPO_4$ was published in 1991 [426].

The main advantage of ECD is that, when the ceramic film coats some portion of the metal substrate, the local current density is automatically reduced due to substantially higher resistance of the ceramics. Therefore, the current density is concentrated in the metal substrate. This negative feedback of the current density results in high uniformity in both the form and thickness of the coating, even for complex geometries. To compare ECD and EPD technologies, one can argue that the absence of electrochemical reactions in EPD is a major difference between them but both technologies can be used to coat only conductive materials. Also, because of the substantial dimensional difference between suspended particles and dissolved ions, EPD is mainly used for the preparation of thicker deposits, while ECD allows the formation of thinner deposits (Figure 15) [425].

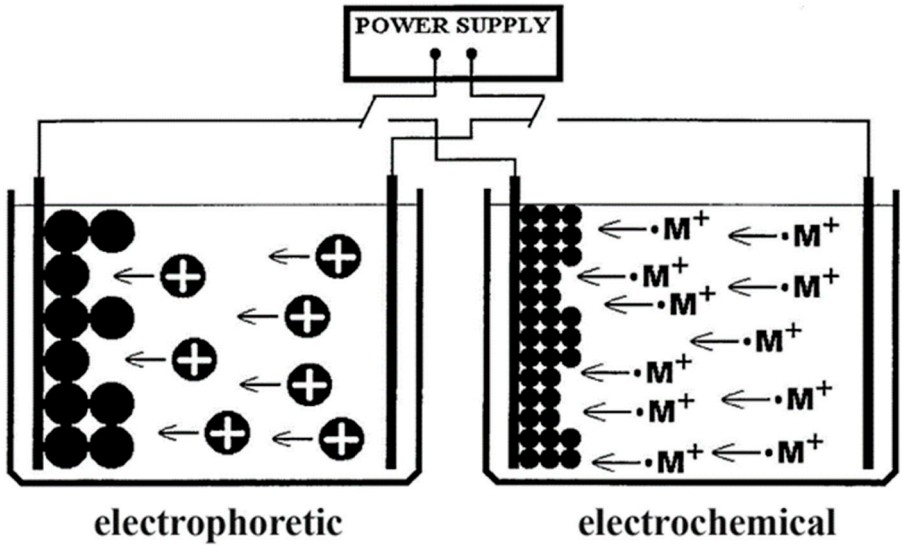

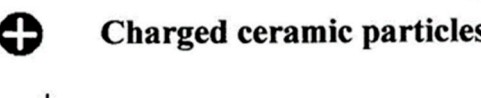

**Figure 15.** Schematic diagrams of electrophoretic and electrochemical deposition showing the movement and deposition of positively charged ceramic particles (**left photo**) and $M^+$ ions (**right photo**) on the cathode. Reproduced with permission from [425].

ECD of $CaPO_4$ occurs in aqueous solutions and may be performed at both room temperature [97,413,418,427–430] and at temperatures from 50 to 200 °C [35,36,40,155,416,417,423,431–437], for which autoclaves are used, and the method is called "hydrothermal electrochemical deposition" [434,435] or "hydrothermal electrodeposition" [436,437]. Regarding a choice of electrolytes, ECD may be performed in both a simple system of calcium nitrate and ammonium orthophosphate in water [155,423,424,427–433], and much more complicated simulating solutions, such as SBF [438–440]. In the latter case, ion-substituted forms of $CaPO_4$ are deposited. Experimental parameters such as solution pH, electrolyte composition, temperature, and time were found to affect both the structure and properties of

the deposits (Figure 16) [431]. For example, at solution pH = 3–5, DCPD can be deposited and then converted to HA by treatment with a 1 M NaOH solution at 80 °C for 1 h and to CDHA by treatment with SBF for 7–15 days under ambient conditions [421,429]. Coating thicknesses of less than 1 μm can be achieved. Reducing the thickness improves resistance to delamination frequently observed with thicker coatings [440]. Deposition of nano-sized crystals [441–444] and biphasic CDHA + DCPD formulations [445] are also possible. Natural materials such as shells have been tested as a source of calcium for ECD [446]. The magnetic field has been found to induce magnetohydrodynamic convection due to Lorentz forces, greatly reducing the formation of volcanic structures and producing more uniform deposits without changing the Ca/P ratio [447]. Unfortunately, ECD requires a sufficient amount of electrolyte for the surface area (20 mL of electrolyte is needed to coat 1 cm$^2$ of an implant). In addition, the formation of hydrogen gas inhibits deposition due to the formation of bubbles on the metal surface, resulting in a non-uniform coating [180]. A modulated technique was proposed to overcome the latter problem [413].

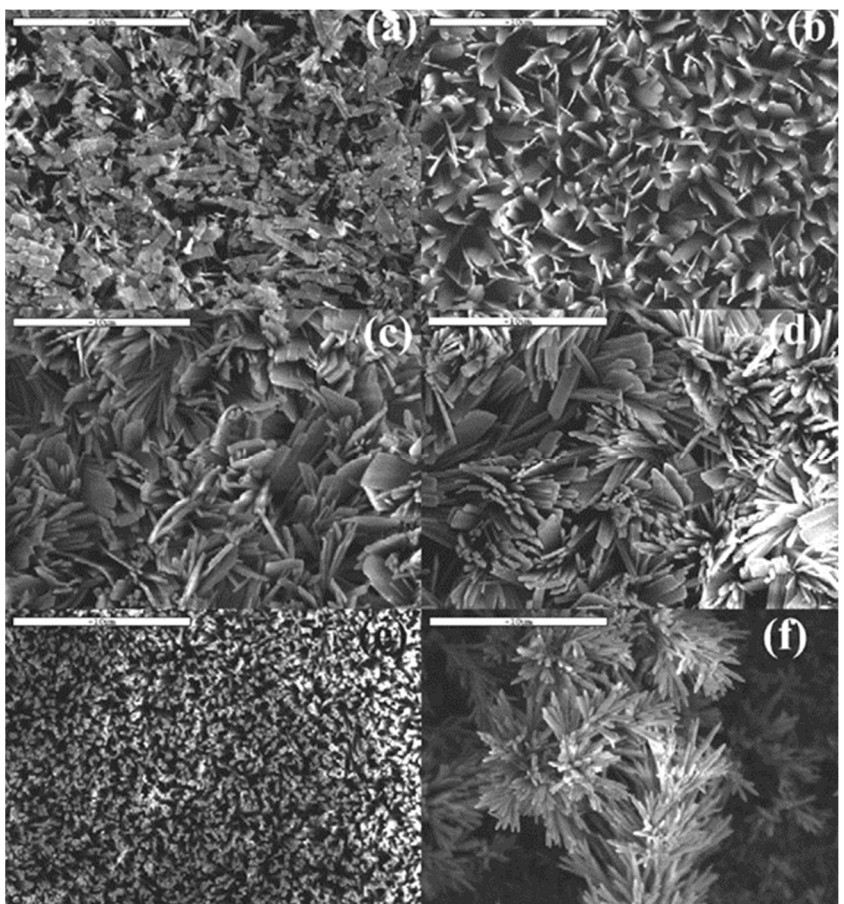

**Figure 16.** Typical SEM images of electrochemically deposited CaPO$_4$ coatings under different conditions: (**a**) pH = 4.2, 60 °C, 2 h; (**b**) pH = 4.2, 85 °C, 4 h; (**c**) pH = 6.0, 80 °C, 3 h; (**d**) pH = 6.0, 90 °C, 3 h; (**e**) pH = 6.0, 90 °C, 3 h, 0.01 M KCl added; and (**f**) pH = 6.0, 90 °C, 3 h, 1.00 M NaNO$_2$ added. All scale bars 10 μm. Reproduced with permission from [431].

According to the literature, nucleation of CaPO$_4$ crystals in ECD can occur either instantaneously or progressively [448]. Nucleation is said to be instantaneous when the rate of nucleation at a site is expected to be at least 60 times faster than the expected rate of coverage of that site by growth alone. Nucleation is said to be progressive if the rate of coverage by growth at a site is expected to be at least 20 times the rate of coverage by nucleation at the same site. In any case, once formed, CaPO$_4$ nuclei can grow in one, two or three dimensions, resulting in deposits of different shapes such as needles, discs and

hemispheres, depending on the deposit/substrate binding energy and crystallographic misfit. In the case of CDHA, nucleation was instantaneous for the first 12 min, with two-dimensional growth. Subsequently, nucleation progressed and was accompanied by three-dimensional growth [448]. Experimental data indicate that the ECD of CDHA occurs via intermediate formation of the OCP precursor phase [360]. Perhaps this may be the reason why ECD formed CDHA coatings were found to consist of two distinct layers [449].

Commonly, ECD formed $CaPO_4$ have a homogeneous structure because they are formed gradually through nucleation and growth processes at relatively low temperatures [8]. The deposits may be porous [410]. In addition, HA deposits can be produced by ECD from non-apatitic $CaPO_4$ with subsequent additional processing [148–150]. The deposited $CaPO_4$ would then be heat treated at 125 °C in water vapor [450] and further densified by sintering at temperatures up to 800 °C to improve bonding to the substrate. In addition, ECD can be combined with other deposition techniques. Namely, ECD has been used to deposit $CaPO_4$ seeds, followed by secondary growth of $CaPO_4$ crystals under hydrothermal conditions [451–453]. An electrical polarization of $CaPO_4$ deposits has been found [453]. Further details on this topic can be found in references [340,408,454–457].

With regard to biomedical applications, ECD formed $CaPO_4$ coatings on implants are commercially produced for clinical use. Examples include BONIT® (phase composition: >70% DCPD, <30% CDHA) (DOT GmbH, Rostock, Germany) and BoneMaster® (BIOMET Corp., Warsaw, IN, USA). In addition, MedicalGroup (Velin, France) and Lincotek Medical (Trento, Italy) are contract manufacturers, who, among other types of business, also use ECD to put down DCPD (trade name SALTINA™) and undisclosed $CaPO_4$, respectively, onto customers' orthopedic implants. Among others, BONIT® has been in use since 1995 and, to date, approximately 4.7 million orthopedic and dental implants have been implanted in patients of various ages [458].

To conclude this section, it should be emphasized that all the above applies to the ECD of $CaPO_4$ onto metals. However, in 2019 the first paper on the successful ECD of $CaPO_4$ (i.e., HA) on marble (i.e., $CaCO_3$) was published. The results of that study showed that the ECD technique can promote and enhance the formation of HA coatings on marble surfaces, although the protective effect obtained was not perfect [459]. Although it is difficult to draw definitive conclusions, it should be noted that this approach seems to be a combination of ECD and chemical conversion deposition (see Section 6. Conversion-formed $CaPO_4$ deposits for the details).

### 4.3.3. Sol–Gel Deposition

By classification, a sol is a biphasic suspension of colloidal particles in a liquid, while a gel is a composite because it consists of a solid skeleton or network that encloses a liquid or an excess of a solvent. Therefore, as the name suggests, sol–gel deposition is a wet chemical technique that involves a transition from the liquid "sol" phase to the solid "gel" phase. Since colloidal particles are between 1 and 1000 nm in size, the gravitational force on these particles is usually negligible. Interactions between particles are therefore governed by both short-range forces and surface charges.

The sol–gel processes date back to the origins of modern chemistry and were first recognized as an applied technology in 1846 when Ebelmen observed a hydrolysis and a polycondensation of tetraethyl orthosilicate [460]. In the preparation of sols, inorganic salts and/or organic elemental compounds such as alkoxides are usually used as precursors. For example, when synthesizing HA, orthophosphate precursors consist of $H_3PO_4$, $P_2O_5$, $P(OC_2H_5)_3$ and $(NH_4)_3PO_4$ dissolved in ethanol, while calcium precursors are $Ca(CH_3COO)_2 \cdot H_2O$ and $Ca(NO_3)_2 \cdot 4H_2O$ [320,461]. In addition, Ca glycosides and $Ca(OC_2H_5)_2$ precursors can also be used, but the latter must be dissolved in a non-aqueous solvent. In any case, a homogeneous solution of Ca- and P-containing precursors in an organic solvent should be prepared so that they can be mixed with the reagents or water used in the next step. After mixing, sols are formed by hydrolysis or condensation reactions. Sol can also be prepared by dispersing colloidal particles in a liquid and then destabilizing the sol to produce a par-

ticulate gel. The sol then condenses into a continuous liquid gel phase. Further drying and heat treatment transforms the gel into a dense ceramic or glass material [462]. The deposited gel forms a $CaPO_4$ coating, film or layer, typically using dip coating technique [463–475], but other deposition techniques such as spraying [466] and spin coating [475–480] are also used (details of these techniques are given below).

Sol–gel deposition of $CaPO_4$ is therefore carried out by precipitating a sol containing Ca and P onto the substrate at a low reaction temperature. After a short time, the sol turns into a gel. After drying, a solid deposit remains on the surface. To produce a thick deposit consisting of several layers, the deposition and drying cycle is repeated several times. Sol–gel $CaPO_4$ are porous, have a low density and a poor adhesion to the substrate. To improve their properties, the coated substrate is annealed at temperatures between 400 and 1000 °C [428,464–480]. Different $CaPO_4$ compounds are obtained depending on both the Ca/P ratio and the firing temperature. The resulting deposits are very dense and can adhere strongly to the underlying substrate [8]. Two-phase deposits can also be prepared where particles of one type of $CaPO_4$ are embedded in a continuous coating of another type of $CaPO_4$ (Figure 17) [469]. To increase the bond strength between the deposit and the substrate, another layer of compound can also be applied in between before sol–gel deposition of $CaPO_4$ [481,482]. More details on this topic can be found in references [454,461].

**Figure 17.** A schematic drawing of biphasic deposits prepared by sol–gel method with nano-sized β-TCP particles embedded into a continuous coating of fluorinated HA (FHA). Reproduced with permission from [469].

As a result, sol–gel deposited $CaPO_4$ coatings on implants are commercially available. For example, NanoTite™ (BIOMET 3i, Palm Beach Gardens, FL, USA) is produced by dipping OSSEOTITE® Surface (BIOMET 3i, Palm Beach Gardens, FL, USA), dental substrates into alcohol-based sols containing dispersed colloidal particles of undisclosed $CaPO_4$ of 20–100 nm in size. After removal, the substrates are dried at 100 °C. With this method, discrete crystalline deposits of nanosized $CaPO_4$ particles cover about 50% of the substrate surface and the remaining surface is covered with a layer of $TiO_2$.

### 4.3.4. Wet-Chemical and Biomimetic Deposition Techniques

By definition, biomimetics (synonyms: biomimicry, bionics) attempts to apply biological methods and natural systems. Therefore, biomimetic deposition is considered as a process of forming biologically active, bone-like $CaPO_4$ deposits on a substrate by immersion in various artificial simulated solutions, such as Hank's balanced salt solution (HBSS), SBF and PBS [37,39,54,59,72,73,75,76,366,369,483–489]. Wet chemical deposition is very similar to biomimetic deposition. The differences between the two are solution composition (wet chemical deposition solutions can contain any ions or additives, while biomimetic deposition solutions need to contain only biologically relevant ones), solution pH (wet chemical deposition can be performed at any pH value, while biomimetic deposition should

be performed at pH values from ~5.0 to ~8.5) and temperature (wet chemical deposition can be performed at temperatures from 0 to 100 °C, while biomimetic deposition must be performed at temperatures from ~20 to ~40 °C). Therefore, common reagents such as $Ca(NO_3)_2 \cdot 4H_2O$ may not be usable as a source of calcium for biomimetic deposition. As for deposition temperatures, their narrow range does not affect the deposition step in the case of biomimetic deposition but it really does in the case of wet chemical deposition. Namely, under similar experimental conditions, DCPD was found to be deposited at temperatures below 70 °C, DCPA at temperatures above 80 °C and a mixture of both phases at 75 °C [490]. The solution pH strongly influences the chemical composition of the deposited $CaPO_4$. Obviously, any type of $CaPO_4$ may be deposited by wet-chemical deposition (except those that cannot be deposited from aqueous solutions, see Table 1) but only CDHA (pure and ion-substituted, which depends only on the chemical composition of the solution) can be deposited by biomimetic deposition [490].

Historically, $CaPO_4$ was first biomimetically deposited on substrates in 1990 [483]. This method includes heterogeneous nucleation and growth of bone-like $CaPO_4$ (mainly, CDHA) crystals on the implant surface under physiological conditions (temperature 25 °C or 37 °C, solution pH 6–8) for several days to weeks. Furthermore, both OCP [14,438,488,489,491–493] and a thin layer of "apatite precursor", i.e., ACP [376,377] can be deposited in a biomimetic manner. The thickness of such $CaPO_4$ deposits is in the range of a few microns (Table 3) and, according to XRD measurements, most biomimetic deposits appear to be amorphous or poorly crystalline [8]. It should be noted that there have been studies where one type of $CaPO_4$ (i.e., CDHA) was biomimetically deposited on the surface of another $CaPO_4$ (i.e., β-TCP) [493].

The mechanism of bone-like apatite formation on the oxidized surface of Ti has been investigated in detail [494,495]. Fleetingly, the process is as follows. Initially, an amorphous sodium titanate layer is formed on the Ti surface due to an alkaline pretreatment. Afterwards, immediately after immersion into a simulating solution, the sodium titanate exchanges $Na^+$ and $H_3O^+$ ions in solution to form Ti-OH groups on its surface. The Ti-OH groups then took up $Ca^{2+}$ ions from the SBF, forming an amorphous calcium titanate layer. After a longer soaking time, the amorphous calcium titanate took up orthophosphate ions in the SBF and formed ACP precipitates with a Ca/P atomic ratio of about 1.4. ACP was then transformed into bone-like ion-substituted CDHA with a Ca/P ratio of about 1.65 [494]. A subsequent study clearly indicated that various titania gels can form after the exchange of $Na^+$ ions with $H_3O^+$ ions, but only those with rutile or anatase structure persuaded CDHA formation [495]. Although Ti and Zr belong to the same group in the periodic table of elements, the surface reactions were found to differ between Ti and Zr substrates [496]. Concerning the influence of surface roughness, the in vitro deposition process of $CaPO_4$ is preferentially initiated in surface cavities rather than in the plane of the sample, indicating that cavities have a strong template effect on deposition [376,377,497]. More specific details on this topic can be found in reference [498].

Biomimetic deposition of $CaPO_4$ is time-consuming, so faster methods have been sought. The most popular option is using concentrated types of the simulation solutions. Namely, the time of apatite induction in 1.5-fold SBF was significantly reduced compared to standard SBF [39,59,487]. Therefore, the SBF concentration was further increased. That is, 2× [487,499–501], 5× [502–505], 7× [485] and even 10× [70,485,506–509] SBF solutions were used to accelerate precipitation and increase the amount of $CaPO_4$ precipitates. Nevertheless, precipitation can be very rapid. To reduce the rate of the rapid precipitation from the concentrated SBF solution, acidification to pH ~5.8 by bubbling $CO_2$ has been considered; as $CO_2$ gas evaporates from the solution, the pH value slowly increases, causing $CaPO_4$ precipitation on the substrate [438]. A similar effect can be achieved by the addition of a combination of urea and urease. Urease enzymatically converts urea to $CO_2$ and $NH_3$ and increases the pH [510]. Furthermore, HA powder can be dissolved in deionized water saturated with $CO_2$ and precipitation starts due to $CO_2$ exhalation [77]. In another study, stable solutions containing high amounts of dissolved calcium and orthophosphate ions

were prepared. After the addition of $NaHCO_3$ that solution became supersaturated and, after 24 h of immersion, a uniform deposit with a thickness of about 40 µm and adjustable composition from ion-substituted CDHA to DCPD was obtained [511]. However, it should be noted that the application of SBF concentrates changed the chemical composition of the precipitates, i.e., the carbonate concentration increased and the orthophosphate concentration decreased [512]. It has also been found that precipitates produced from low concentrations of SBF solutions have high initial $Mg^{2+}$ uptake and form a relatively smooth surface profile, while high concentrations of SBF solutions lead to low $Mg^{2+}$ uptake and form a very rough surface profile consisting of micrometer-sized plates [485]. The disadvantages of this method are the long processing time, the need for chemical activation of the surface and the need for daily renewal of the simulated solution.

The nucleation and growth of $CaPO_4$ deposits on Ti-6Al-4V alloys with $5\times$ SBF were studied in detail by both atomic force microscopy and circumferential scanning electron microscopy [504]. It was found that immersion in $5\times$ SBF for only 10 min resulted in the appearance of dispersed precipitates with a diameter of about 15 nm. With increasing immersion time, the packing of $CaPO_4$ precipitates of several tens of nanometers in diameter formed larger spheres, forming a continuous coating on the substrate. That coating consisted of nanodimensional crystals and direct contacts between $CaPO_4$ and the metallic surface were observed [504].

Simplifying the ionic composition of standard simulated solutions is another option to improve deposition kinetics [491,492,510,513–519] and heating is an additional option [508,517–520]. For example, rapid (several hours instead of 14 days for SBF) biomimetic deposition of CDHA on Ti-6Al-4V alloy substrates can be achieved using slightly supersaturated $CaPO_4$ solutions with a simpler ionic composition than SBF. Thin-film XRD showed that the deposits obtained after 3 h consisted of ion-substituted CDHA with low crystallinity and its content increased as the immersion time was extended to 3 days [513].

While somewhat off-topic, it is worth noting that some types of $CaPO_4$ bioceramics can be prepared by wet chemical vapor deposition. For example, hollow $CaPO_4$ tubes were synthesized by this method. First, $CaPO_4$ was deposited on the surface of an anodic aluminum oxide membrane by mixing diluted $H_3PO_4$ and $Ca(OH)_2$ solutions. To remove the membrane and collect the separated tubes, the coated membrane was soaked in an etching agent ($NH_3:H_2O_2$:purified water = 1:1:3 by volume) for 15 days at pH = 11. A $CaPO_4$ colloid was prepared, which afterwards was centrifuged for 20 min at 3500 rpm, followed by washing, drying and calcination [521].

At the end of this section, it should be noted that biomimetic and wet-chemical $CaPO_4$ coatings, films and layers on implants are commercially available. Examples include Peri-Apatite™ HA (Stryker Corporation, NJ, USA) and HA*nano* Surface (Promimic AB, Sweden). According to the manufacturer, HA*nano* Surface with a thickness of 20 nm can be applied to substrates by spraying, dripping or dipping, and the excess coating solution is removed by a short heat treatment after spinning and compressed gas. Therefore, it can be said that techniques such as dip coating, spin coating and cold spraying (a description of these techniques is given below) are also used commercially for the deposition of $CaPO_4$.

### 4.3.5. Dip Coating Technique

Dip coating is a common deposition technique suitable for a wide range of substrates due to its simplicity. Dip coating consists of a series of sequential steps. The substrate is immersed in a $CaPO_4$ suspension at a constant rate. The wet precipitate spontaneously deposits on the substrate as the substrate is lifted up. The pull-up process is usually done at a constant speed to avoid vibrations. This speed determines the thickness (the faster, the thinner). The thickness also depends on the solids content and the viscosity of the liquid. Excess suspension is drained from the surface. At the same time, the solvent evaporates, making the deposit denser [522]. In the case of volatile solvents such as alcohol, evaporation has already started during the deposition and drainage phase. After drying, a solid deposit is obtained [523,524].

For dip coating there are two mechanisms governing the deposition. The first one is liquid entrainment. This occurs when the sample is pulled out of the slurry faster than it is removed from the surface, leaving a fine deposit [525]. Slip casting is the second mechanism, a filtration process in which the porous substrate is immersed in a suspension for a period of time and then removed from it. Due to capillary forces, the liquid phase is sucked out of the suspension (slip) and solid particles condense at the substrate–suspension interface, forming a wet film (cake) on the surface [526]. Therefore, the liquid entrainment mechanism is influenced by the draw velocity and the properties of the suspension (solid volume fraction, viscosity), while the slip casting mechanism is influenced by the surface microstructure of the substrate (porosity, pore size) and the suspension properties. $CaPO_4$ deposits from 2 µm to 0.5 mm thickness can be formed by varying those parameters [525,526].

$CaPO_4$ has been deposited on various substrates by immersion [55,62,105,106,523–532]. Examples of dip-coated substrates are shown in Figure 18 [55]. When the deposition is followed by sintering [523], this technique can be called "frit enameling" [8]. Moreover, immersion steps are also widely used in other deposition techniques (e.g., the sol–gel deposition mentioned above and some of the techniques described below), so the total number of studies on immersion coating exceeds the cited examples.

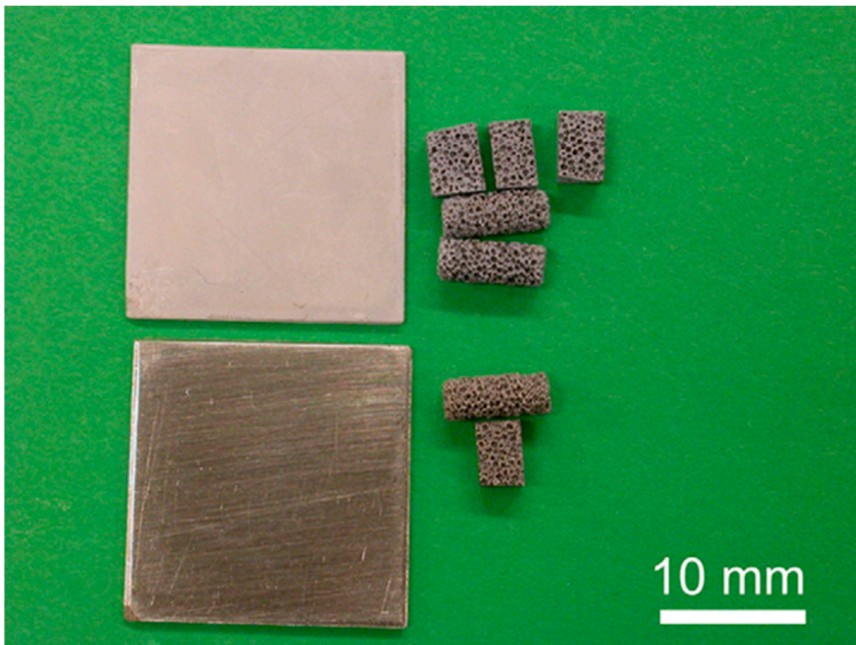

**Figure 18.** A photograph showing the application of the dip–dry deposition method onto metals. A stainless steel square and five porous (pore size ~500 µm) Ta samples in the upper half were treated eight times with supersaturated $CaPO_4$ solution and then dried. The samples show that the deposited coatings are uniformly white. In contrast, the deposited sample in the lower half shows a metallic luster. Reproduced with permission from [55].

Finally, the method of immersing the substrate into molten $CaPO_4$ should be briefly mentioned. Since the melting point of the thermally stable $CaPO_4$ (HA, FA, α-TCP, TTCP) exceeds 1500 °C, substrates with higher melting points (Ti, Ta, $Al_2O_3$, $ZrO_2$, etc.) can only be coated using this technique. Those experiments have been carried out and this technique has been called "immersion coating" to distinguish it from normal dip coating as it does not "wet" ("wet" means "moisture" and high temperature melts cannot contain moisture). As with the thermal spray techniques described above, the deposits obtained from the molten HA powder appears to be a complex mixture of various compounds, mainly β-TCP. Moreover, the adhesion of the deposits to the substrate was low and could be removed by light friction [523].

### 4.3.6. Spin Coating Technique

Spin coating is a method used to apply uniform thin deposits to a flat substrate such as a disk or a plate. It is very similar to dip coating. The process consists of four stages: deposition, spinning up, spinning off and vanishing. The substrate is immersed in a solution, suspension or sol, and pulled at a constant speed, usually by a motor. Since the substrate is first immersed and then rotated, such a method is sometimes referred to as the dip–spin method [533]. Alternatively, a solution, suspension or sol can be poured or dripped onto the flat surface of the spinning substrate [534]. The solution is expelled from the spinning substrate and the solvent evaporates, resulting in deposition (Figure 19). Deposition by spin coating is mainly driven by two independent parameters: viscosity and spin speed. The thickness of the deposited CaPO$_4$ is indirectly proportional to the spin rate, $h \sim \omega^{-n}$, where $h$ is the thickness, $\omega$ is the angular velocity and $n$ is a parameter dependent on the solvent evaporation rate.

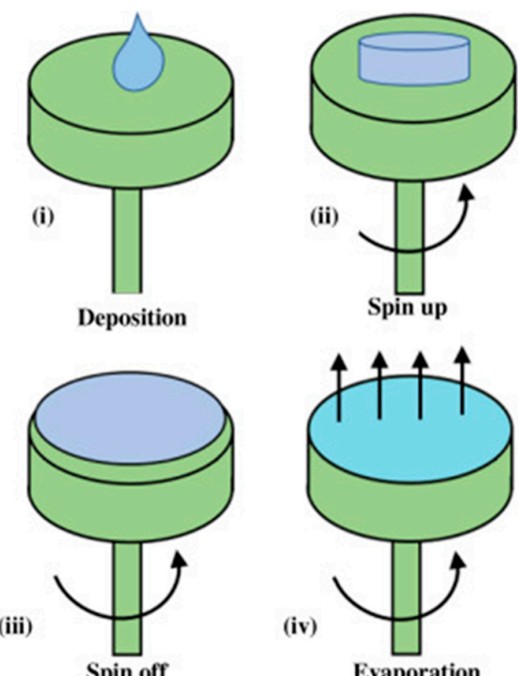

**Figure 19.** A schematic diagram of a spin-coating technique.

After drying, this procedure can be repeated to prepare thicker deposits. Afterwards, a final heat treatment is commonly performed to improve the physical properties of CaPO$_4$ deposits. The machine used for spin-coating is called a spin-coater or simply spinner [535]; there are only a few publications on spin-coating of CaPO$_4$ [475–480,533–536].

### 4.3.7. Hydrothermal Deposition Method

Hydrothermal deposition is one of the most cost-effective techniques. However, because the hydrothermal process is carried out at high temperatures (>80 °C) and for relatively long times (>1.5 h), CaPO$_4$ deposits are usually crystalline.

Hydrothermal methods have been used for the deposition of CaPO$_4$ on metallic [88,89,451,452,537–545], inorganic [546] and polymeric substrates [547,548], and even CaPO$_4$ [549]. This method has sometimes been referred to as the "chemical solution deposition method" [550–552] or as the "chemical bath method" [553]. In most of these studies, the deposition process was based on the formation of EDTA-Ca$^{2+}$ chelate compounds by co-dissolution of EDTA with Ca-containing salts under ambient temperature conditions. With increasing temperature, thermal decomposition of the EDTA-Ca$^{2+}$ chelate compound occurs, providing a Ca$^{2+}$ ion concentration high enough to allow precipitation in the presence of PO$_4{}^{3-}$ ions. In other words, hydrothermal treatment at 90 °C for 2 h

successfully formed well crystalline precipitates of OCP and HA on both Mg and Mg-Al-Zn alloys from 0.25 M $KH_2PO_4$ and EDTA-$Ca^{2+}$ treatment solutions in the pH range 5.9–11.9. The crystal phase and microstructure of those precipitates were found to vary with the pH of the treatment solution. That is, in the weakly acidic (pH = 5.9) solution, a two-layer deposit was formed with an outer coarse layer consisting of plate-like OCP crystals and a dense inner layer consisting mainly of HA crystals. A two-layer structure was also formed in weakly alkaline (pH = 8.9) solutions: the coarse outer layer contained rod-shaped HA crystals while the dense inner layer contained well-packed HA crystals. Both layers were observed to grow with increasing processing time. Needle-like HA crystals were formed in strongly alkaline (pH = 11.9) solutions. A thin layer of $Mg(OH)_2$ was similarly formed at the boundary between the Mg substrate and the deposited $CaPO_4$. OCP and HA improved the corrosion resistance of pure Mg and Mg-Al-Zn alloys in both HBSS and 3.5 wt% NaCl solutions. It was also found that the level of protection of the $CaPO_4$ film varies with its crystal phase, microstructure and thickness. Moreover, at cyclic stresses below the fatigue limit, the films showed good adhesion under slight plastic deformation [539–541,551].

Furthermore, the hydrothermal treatment can be carried out without EDTA below a solution pH of ~4.0, resulting in DCPA deposition [554,555], and in the presence of ammonium citrate and ammonium fluoride below pH about 8.0, resulting in FA and fluoridated HA deposition [545]. In addition, hydrothermal deposition of $CaPO_4$ can be realized both from SBF [88,89,547] and onto previously deposited $CaPO_4$ seeds [451,452].

### 4.3.8. Thermal Substrate Deposition Technique

Thermal substrate deposition takes advantage of the difference in solubility between low and high temperatures. This means that solid deposits can be deposited directly on the substrate by heating the substrate in a suitable saturated aqueous solution. Various heating methods have been proposed. Namely, it can be heated by passing an electric current through a conductive substrate such as metal foil or wire. Materials with complex shapes can be heated by non-contact techniques such as induction heating deposition [63,165,556–559]. There is also ultra-high frequency induction heating deposition, referred to as "chemical liquid vaporization deposition" [560]. In both cases, crystallization can be achieved by heating a sample immersed in solution to 160 °C and achieving local supersaturation.

$CaPO_4$ have been deposited on various substrates by thermal substrate deposition [63,165,556–564]. Namely, an alternating current was applied to Ti substrates immersed in an aqueous solution containing calcium and orthophosphate ions. Precipitation occurred at a solution pH of 4 to 8 and temperatures up to 160 °C for 10–30 min. The type of deposits varied according to the pH, temperature and ionic concentration of the solution. Namely, high quality precipitates were obtained on Ti consisting mainly of CDHA (pH > 6) or DCPA (pH = 4) [561–564]. Likewise, DCPA was deposited on carbon at solution pH ~4.5 [559] and subsequent hydrothermal treatment in alkaline solution converted DCPA to HA [63,165,557–560]. In all studies, the apatite content in the deposits increased with increasing heating time, solution pH and temperature.

### 4.3.9. Alternate Soaking Deposition

This method was developed in 1998 [565]. The process consists of several successive deposition cycles: in one cycle, the substrate is immersed in a Ca-containing solution, rinsed with water, then immersed in a $PO_4$-containing solution and rinsed again [566]. After repeated applications, $CaPO_4$ accumulates [567–573], using simple inorganic salts such as calcium chloride or nitrate as a Ca source and sodium or ammonium hydroorthophosphate as a $PO_4$ source. The duration of each immersion step can vary from 1 min [569] to 2 h [571] depending on the substrate. For even shorter immersion times (<1 min), the deposition technique is referred to as "alternating dipping" [574] or "alternating immersion" [575]. With regard to examples, alternating immersion has been used to deposit $CaPO_4$ on Ti substrates [569]. The pretreated substrate was immersed in 0.5 M $CaCl_2$, washed with distilled water and then immersed in 0.1 M $Na_2HPO_4$ solution (Figure 20). The amount of $CaPO_4$

deposited increased with the number of reaction cycles but was independent of solution temperature and immersion time [569], indicating that the deposition is dependent on ion exchange and adsorption on the pretreated surface [567–575]. Other studies confirmed those findings [567–575]. In 2011 the process was automated to deposit significant amounts of $CaPO_4$ with minimal effort and energy [576]. Moreover, flower-like $CaPO_4$ deposits were prepared on titanium surfaces by this technique [577].

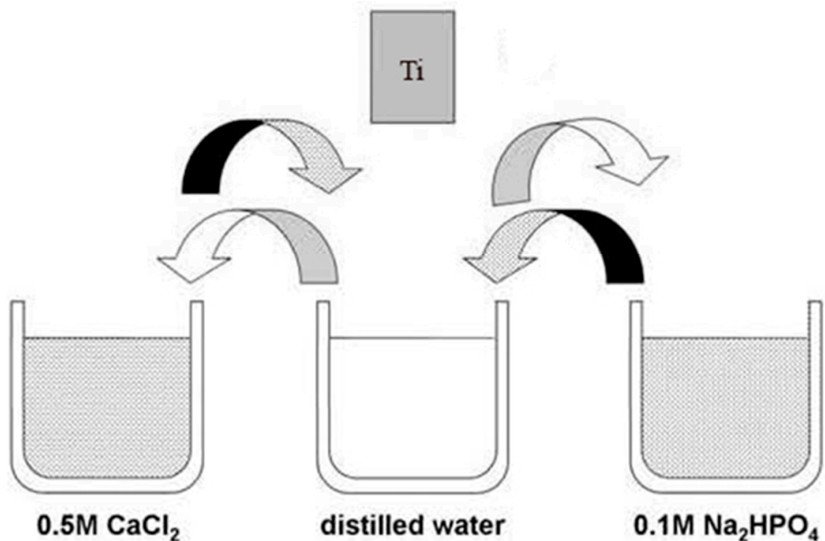

**Figure 20.** A schematic diagram of the alternate soaking deposition method. Reproduced with permission from [569].

At the end of this section, it should be noted that alternate soaking might be performed solely at the initial stages just to deposit precursors, such as ACP or $CaPO_4$ seeds, followed by the deposition from SBF [34,74,81]. In addition, alternative soaking can be performed under hydrothermal conditions [42].

4.3.10. Micro-Arc Oxidation (MAO) Technique

MAO (synonyms: anodic spark deposition, micro-arc discharge oxidation, plasma electrolytic oxidation (PEO), anodic plasma chemical treatment, spark anodizing) is considered a combination of plasma chemical and electrochemical processes capable of producing ceramic coatings on various metals and alloys. The process combines electrochemical oxidation of metal surfaces with a high-voltage (up to 500 V, often AC) spark treatment carried out in aqueous electrolytic baths. However, in 2022, the first paper on the PEO process carried out in a molten salt electrolyte (the authors used molten nitrates) was published [578]. In addition, ultrasound might be applied during the MAO process (called UMAO = ultrasound MAO) [579]; there is also a more complicated hybrid UMAOH (=UMAO hybrid) process, which consists of two stages: UMAO with ultrasound treatment for 8 min and subsequent MAO without ultrasound for 2 min. The latter process produces $CaPO_4$ deposits more suitable for drug delivery applications [580].

In the MAO process, sparks arise and move rapidly along the treated surface, while the temperature and pressure in the discharge channel are high enough to trigger thermochemical interactions between the substrate and the electrolyte, reaching ~$10^4$ K and ~$10^3$ MPa, respectively [581]. The electrolytic bath usually also contains modified elements in the form of dissolved salts (such as borates and silicates), which are incorporated into the deposits. Therefore, depending on the chemical composition of the electrolytic bath, MAO deposits represent complex mixtures of metal hydroxides and/or oxides with silicates, borates, etc. Moreover, the nature of the substrate material has been found to influence both the morphology and composition of MAO deposits [582]; while in the presence of Ca- and P-containing reagents, such deposits also contain various Ca- and P-containing compounds

(but rarely $CaPO_4$) [369,583–586]. Namely, MAO deposits on Ti were found to consist of a complex mixture of titanium oxides (rutile and anatase phases), $CaTiO_3$, $Ca_2Ti_5O_{12}$, $\beta$-$Ca_2P_2O_7$ and $\alpha$-TCP [369]. Other studies confirmed those findings [583,584,586]. Also, for the same purpose, nano-sized $CaPO_4$ powder was dispersed in an electrolytic bath and thus the MAO process was carried out from an aqueous suspension [587]. Since formation of $CaPO_4$ in the MAO coating was not detected or only appeared as a miscible phase, all these cases should be considered as another pre-deposition technique (see Section 3 above). A schematic setup of the MAO system can be found in references [32,179].

Further deposition of $CaPO_4$ on the aforementioned types of MAO coatings can take place by continuous or hybrid processing routes. Namely, after MAO treatment of a metal, EPD [401,588–590], ECD [591–593], wet chemical and/or biomimetic (Figure 21) deposition [369,594,595], alternating immersion [575], electron beam evaporation [596] and sol–gel coating [597] are used to deposit $CaPO_4$. In the case of EPD, $CaPO_4$ could be incorporated in situ into growing MAO deposits if electrolytes contained suspended nanoparticles [588].

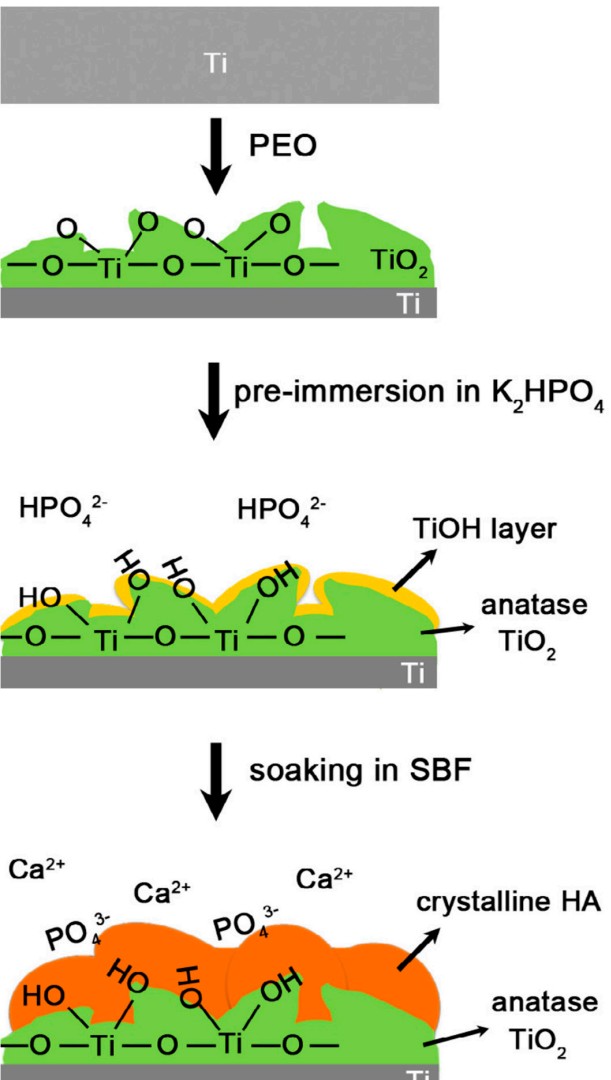

**Figure 21.** A schematic diagram illustrating a plausible mechanism for CDHA deposition on the surface of MAO (PEO)-produced anatase $TiO_2$ deposits on Ti substrates pre-dipped in $K_2HPO_4$ followed by SBF. Reproduced with permission from [595].

However, the MAO process can be carried out with aqueous electrolytes containing solely dissolved Ca- and $PO_4$-containing salts [592–604]. In this case, $CaPO_4$ is deposited in a single step. Namely, MAO deposition on Ti was carried out using DC power supply in an electrolyte containing calcium acetate and calcium glycerophosphate. As a result, porous, coarse deposits consisting of a homogeneous mixture of $TiO_2$ and ACP were formed. The substrate was then subjected to hydrothermal treatment in an autoclave for 10 h at 190 °C in an aqueous solution with pH 7.0–11.0 (adjusted with NaOH). This treatment converted ACP precipitates into HA and the amount of HA precipitated increased with increasing solution pH. The researchers suggested that the hydrothermal treatment allowed ACP to diffuse from the inner layer to the surface, where it hydrolyzed and deposited as HA [598]. Similar findings were discovered for Ti alloys [601,602] and Mg alloys [604], although in the latter case the presence of MgO was also detected. However, when an aqueous electrolyte containing dissolved $CaCl_2$ and $KH_2PO_4$ was used, $CaPO_4$ was deposited without additional treatment [599]. Furthermore, the MAO process can be carried out with electrolytes representing suspensions of HA powder in aqueous KOH or KOH + $K_3PO_4$ solutions, and HA precipitation was observed in both cases [605]. As a modification, there is a study in which the following electrolyte was used: β-TCP 50 g/L, NaOH 10 g/L, NaF 5 g/L [157]. A list of available electrolytes can be found elsewhere [606].

The elemental composition of $CaPO_4$ coatings deposited on Ti by MAO was studied (Figure 22). Those deposits consisted of a CDHA-based outer layer and a $TiO_2$-based inner layer, with small amounts of α-TCP and $CaCO_3$ in the outer layer, and $CaTiO_3$ in the inner layer [600]. Similar results were obtained in another study [601]. Additional details on MAO accumulation of $CaPO_4$ are available in recent reviews [607,608]. BioCera Medical Ltd. (Suffolk, UK) has initiated the production of such deposits on Ti, Mg and their alloys.

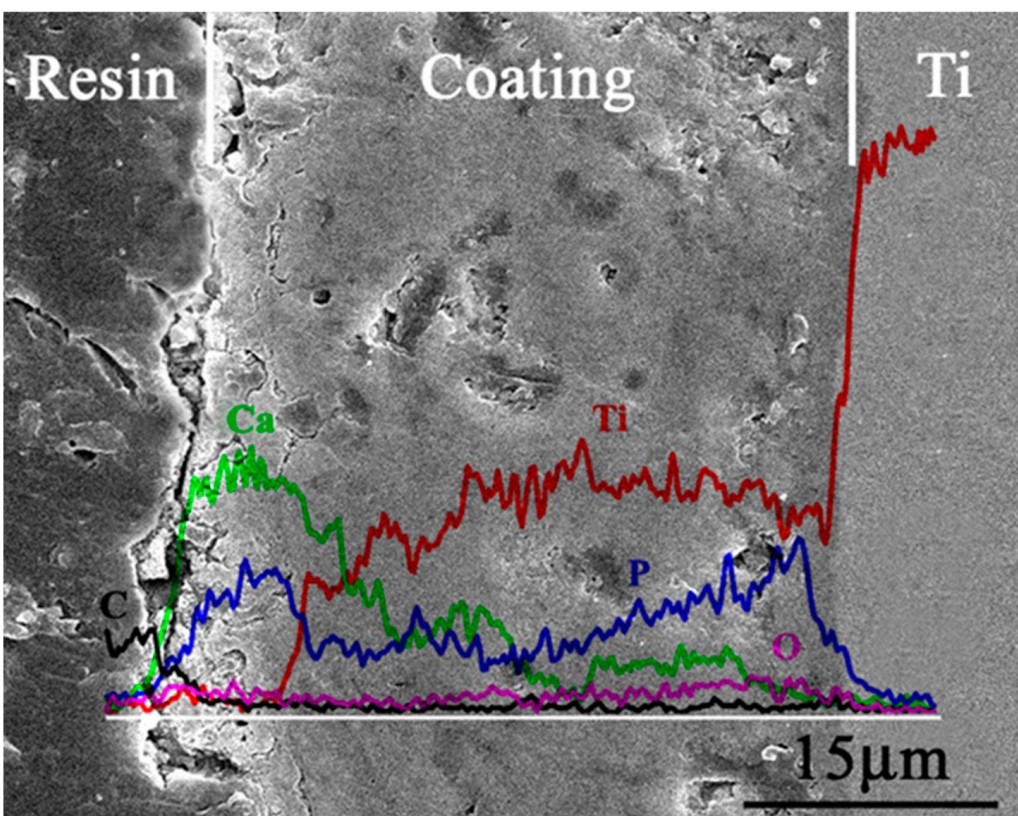

**Figure 22.** A cross-sectional morphology and elemental distribution of CaPO4 coatings deposited by MAO on Ti. Reproduced with permission from [600].

### 4.4. Other CaPO$_4$ Deposition Techniques: Miscellaneous

Before discussing other types of CaPO$_4$ deposition techniques, one must stress that a detailed description is not always possible as those techniques are rare and most of them are mentioned in a few papers only, sometimes just in one article.

#### 4.4.1. Hot Isostatic Pressing (HIP)

HIP is a process used to reduce porosity and increase the density of various materials. The HIP process exposes the components to high temperatures and isotropic gas pressure in a high-pressure containment vessel. To accomplish deposition, the solid core is first coated with CaPO$_4$ (usually HA) powder. Organic binders and other additives are often used to enhance stabilization. To remove these additives, a furnace is built in a high-pressure vessel and the coated substrate is placed inside. It is then heated at 700–1200 °C and simultaneously pressurized with a pressure of 20–100 MPa. The resulting deposits are usually thick (0.2–2.0 mm) and dense [609]. In addition, HIP can be used as a post-deposition technique to reduce the porosity of CaPO$_4$ deposits made by other techniques [610].

The HIP method has been used to deposit CaPO$_4$ on various substrates [9,523,611–613]. However, most of the CaPO$_4$ deposits produced by the HIP method are contaminated with metal and SiO$_2$ particles due to the use of glass-sealed tubes [8]. Moreover, it is difficult to coat complex substrates with this method. Furthermore, there is a thermal expansion mismatch between the substrate and the CaPO$_4$ deposit due to the high temperatures and pressures required.

#### 4.4.2. Implantation into the Surface of Superplastic Alloys

The implantation (or embedding) technique is likely to be similar to the HIP process described earlier. The main difference between the two is that in HIP CaPO$_4$ powder is used whereas here CaPO$_4$ granules are used and granule implantation requires the superplastic properties of substrates; therefore, the implantation technique is only used for CaPO$_4$ deposition on the surface of superplastic alloys. In early studies in the 1990s [614,615], CaPO$_4$ powder was granulated prior to deposition. The granules (32–50 μm in diameter) were then spread on the surface of a superplastic substrate and embedded in the substrate by hot pressing with a piston at 750 °C and 17 MPa for 10–180 min. Heating is required to reach the superplastic temperature, which depends on the material from which the substrate is produced. In the case of Ti alloys, after 10 min of treatment the implantation rate was ~20% and some granules were not on the substrate, while after 60 min of treatment the implantation rate was 100% and the upper region of the granules was exposed. Unfortunately, the CaPO$_4$ deposits produced in the first study were discontinuous [614,615].

In the 2010s, however, this technique was significantly improved. Namely, the injection temperature was increased to between 875 °C and 925 °C, the pressure was increased to 60.3 MPa [616] or a strain rate of $1 \times 10^{-4}$ s$^{-1}$ was applied [617–619], and the particle size of CaPO$_4$ was reduced below 10 μm. These modifications resulted in the formation of continuous CaPO$_4$ deposits 10–25 μm thick with good adhesion properties.

#### 4.4.3. A Double Layered Capsule Hydrothermal Hot Pressing

There are only a few publications on the subject and they are devoted to HA deposition on AZ31 Mg alloys [620] and Ti [9,621,622]. A double-layer capsule was used to create suitable hydrothermal conditions: the inner capsule encapsulated the deposited material and substrate, while the outer capsule was subjected to hydrostatic pressing under hydrothermal conditions. For the Mg alloys study, the alloy rods and DCPD + Ca(OH)$_2$ mixed powder were placed in polyethylene tubes. The mixed powder was loaded into the tube with the Mg alloy rod concentric to the tube axis. Both ends of the tube were fixed with a paper stapler. The entire tube was then sealed with polyvinylidene chloride film. Alumina powder was then placed between the tube and the film. The entire structure was placed in a batch type high temperature, high pressure vessel and subjected to hydrothermal treatment. The vessel was then heated to 150 °C for three hours while the

pressure was maintained at 40 MPa using a pressure regulator. After treatment, the vessel was cooled to room temperature and the HA-coated AZ31 sample was removed. A tensile test was then performed to determine the adhesion of the HA coating to the AZ31 substrate. The average maximum shear stress was measured to be $6.1 \pm 1.0$ MPa. It was found that HA remained on the surface of the Mg alloy after the tensile fracture test. Therefore, this technique allowed HA deposits to bond to Mg and its alloys with good adhesion [620].

In the Ti study, the authors demonstrated that 50 μm thick HA deposits can be deposited on the surface of Ti cylindrical rods at a temperature of 135 °C under a confining pressure of 40 MPa. The deposited HA had a porous microstructure with a relative density of about 60%. The results of tensile tests showed that the shear strength was in the range of 4.0–5.5 MPa. The results also showed that crack propagation occurred within the HA deposits but not along the HA/Ti interface. Therefore, the fracture properties of the HA/Ti interface were found to be higher than HA bioceramics alone [9,621,622].

### 4.4.4. Detonation Gun (D-Gun) Spraying

D-gun spraying is a high-velocity thermal spraying technique that provides a high degree of melting to the starting powder. Until recently, only three papers have been published on this subject [115,185,186], the most recent of which was published in 2001. The spraying equipment, as the name suggests, looks like a gun and consists of a 1.4 m long, water-cooled barrel. A mixture of acetylene and oxygen gases is fed into the barrel along with $CaPO_4$ powder. The hot gases generated in the blasting chamber travel at high velocity along the barrel, heating the $CaPO_4$ particles to the plasticization stage and a maximum velocity of 800 m/s. The high kinetic energy of the hot $CaPO_4$ particles hitting the substrate resulted in the formation of a very dense and solid deposit with a high proportion of amorphous phase (ACP) and small amounts of β-TCP. The thickness formed on the substrate in a single shot depends on the combustion gas rate, carrier gas flow, dimensions of $CaPO_4$ particles, frequency, and the distance between the barrel tip and the substrate. The D-gun spraying cycle can be repeated at a rate of 1–10 shots per second, depending on the thickness required. The low crystallinity and high residual stresses found in $CaPO_4$ deposits have resulted in faster dissolution rates both in vitro and in vivo [115,185,186]. However, it should be noted that this technique was rediscovered in the 2010s by Ukrainian researchers under the name $CaPO_4$ "gas detonation deposition" [623–625]. In recent publications, this process is referred to as "detonation spraying" [626,627].

### 4.4.5. Aerosol–Gel Deposition

Aerosols are colloidal suspensions of small solid particles or droplets in a gas. Since most droplets and solid particles are less than 1 μm in diameter, they are often called microdroplets and microparticles respectively. Various types of air pollution such as dust, clouds and smoke are examples of aerosols. Gels are considered complex because a solid skeleton or network covers the liquid phase or excess solvent. Thus, the aerosol–gel process appears to be a gas–chemical technology that makes a transition from gaseous aerosols to solid gel phases.

The aerosol–gel method has been applied to deposit highly porous $CaPO_4$ deposits on the surface of various substrates [628–631]. $Ca(NO_3)_2$ and $H_3PO_4$ [628] or triethylphosphate dissolved in ethanol [629] were used as precursor solutions. Aerosols can be produced by ultrasonic pulverization of both solutions simultaneously and independently [629] or by pulverization after their previous mixing, resulting in the pulverization of $CaPO_4$ suspensions [628]. In both cases, the microdroplets were fed into the deposition chamber by an air stream. After deposition, the coated substrates were fired at temperatures of 500–1000 °C. Multilayer deposition can be performed to produce thicker deposits [628]. The structure, morphology and composition of the final deposits were found to be consistent with highly porous polycrystalline HA. The adhesion strength measured by the indentation technique was approximately 100 MPa [628–631].

### 4.4.6. Aerosol Deposition (AD)

According to AD technology, fine (~0.1 μm to ~1 μm) $CaPO_4$ powder is mixed with a high-speed (~100 m/s to 400 m/s) carrier gas stream to form an aerosol, which can be deposited on the substrate surface by impact adhesion. The impact due to the high kinetic energy of the particles results in the formation of dense deposits of $CaPO_4$ and the deposits can exhibit high adhesion strength. Since deposition takes place at room temperature, the chemical composition of the starting powder and the deposit remains the same. Therefore, the composition of the deposits can be precisely controlled by changing the composition of the powder; thus, multicomponent deposits can be produced. This deposition technique is sometimes referred to as "a room temperature spray" process or method [632,633].

The AD technique has been used to deposit HA on the surfaces of dense zirconia [632], Ti [15,634,635], Ti alloys [636], $MgF_2$ [84] or Mg pre-coated with poly(ε-caprolactone) (PCL) [637], and various polymers [638,639]. For example, to perform AD, HA powder was sputtered onto Mg samples in a deposition chamber containing oxygen carrier gas at a flow rate of $5 \times 10^{-4}$ $m^3/s$ under a pressure of 9.2 Torr. SEM observations showed that, when HA was deposited on Mg with a PCL interlayer, HA was partially embedded in this interlayer and formed a complex-like structure [637], while no composite-like structure was observed when HA was deposited on Mg with a $MgF_2$ interlayer [84]. Of these, HA deposits on Mg with PCL interlayer showed better stability during deformation compared to those on Mg without interlayer [637].

Depending on the ambient processing conditions, various $CaPO_4$-based biocomposites can be deposited by AD [640–644]. For example, to deposit HA/chitosan biocomposites on AZ31 Mg alloy substrates, researchers used a slit-type nozzle with a rectangular aperture of $10 \times 0.5$ $mm^2$ and air at a flow rate of 30 L/min as the carrier gas. A 5 μm thick HA/chitosan biocomposite was spread over the entire surface of the substrate by scanning with a motorized X-Y table at a scanning speed of 1 mm/s for 1 min. That biocomposite deposit was found to have good corrosion resistance with a high bond strength of 24.6–27.7 MPa [643].

To conclude, a study is available on the opposite process: aluminum oxide was deposited onto HA plates by means of the AD technique [645].

### 4.4.7. Cold Spraying (CS)

CS deposition (also called cold gas spraying [646–648], supersonic powder deposition and kinetic spraying) appears to be quite similar to AD described earlier. The main differences consist of coarser (size 1–50 μm) $CaPO_4$ particles (particles of this size cannot form aerosols), higher velocities of the carrier gas (from ~300 m/s to supersonic ~1200 m/s) and higher (100–800 °C) deposition temperatures [649,650]. For gas pressure, low- and high-pressure cold spraying (LPCS and HPCS, respectively) technologies can be used: LPCG is limited to the use of air as propellant and operates at a maximum operating pressure of 10 bar and temperature of 600 °C, while HPCG operates at a maximum operating pressure and temperature of 40 bar and with nitrogen as a propellant [647]. However, it should be emphasized that, in contrast to thermal spraying techniques (Section 4.1), CS deposition can significantly reduce many of the risks associated with high-temperature techniques, such as thermal decomposition and phase transformation of $CaPO_4$ compounds, since it is only carried out by deformation of solid particles, which occurs at temperatures well below the melting point. This is achieved due to the high kinetic energy of the particles when they hit the substrate. As for the requirements of the powder, it must exhibit excellent flowability to avoid fluctuations in feed rate and inhomogeneous deposition, and therefore particle shape and size distribution are considered to be very important. Regarding the substrate, the most important property seems to be its hardness. This is because hard substrates promote granular fracture by increasing the cohesion and deposition rate of the coating, whereas adhesion is a priority for softer substrates [650]. The proposed bonding mechanism for $CaPO_4$ particles is shown in Figure 23 [646].

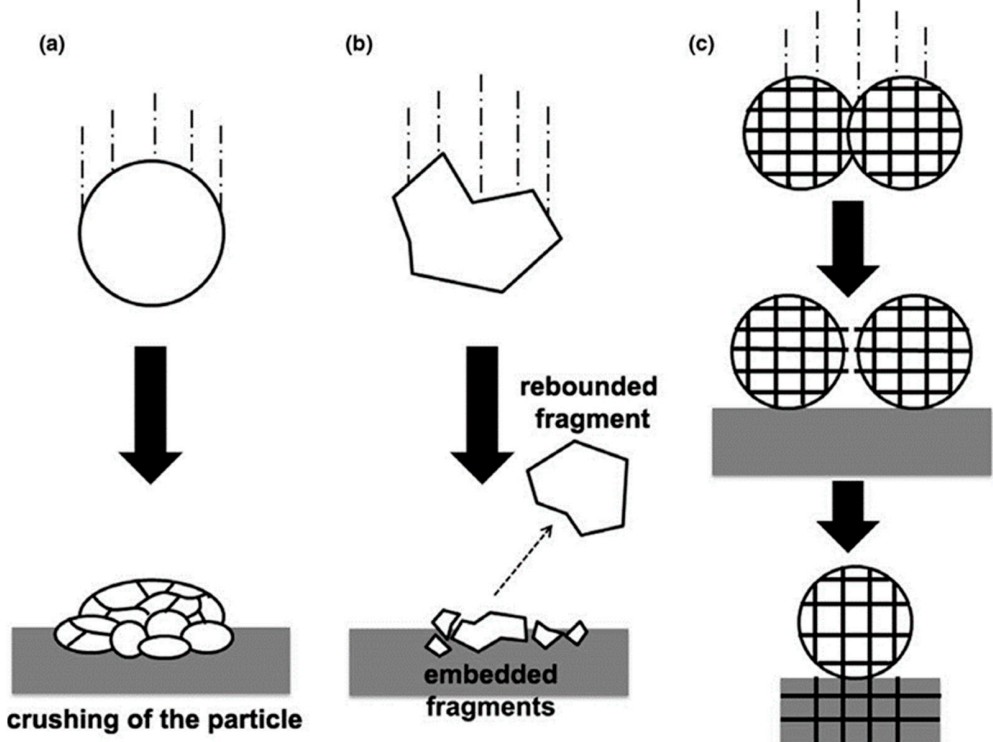

**Figure 23.** Proposed binding mechanisms for single $CaPO_4$ particles during CS deposition: (**a**) impact consolidation at room temperature, (**b**) embedding or mechanical fixation, (**c**) chemical bonding. Reproduced with permission from reference [646].

Like AD described above, in the CS process, atomized $CaPO_4$ particles that experience both slight changes in microstructure and slight oxidation and/or decomposition are accelerated to a very high velocity by a supersonic jet of pressurized gas flow through a Laval-type nozzle. The stream of particles in the gas is propelled towards the deposition chamber where the substrate is located. The deposition system consists of a spray gun, a gas heater, a gas pressure regulator and a dust feeder. The particle temperature at impact depends on several factors such as a heat capacity of the particles, a nozzle design and a gas temperature. The fixation of $CaPO_4$ particles to the substrate occurs through severe plastic deformation. This ensures that the original powder chemistry and phase composition in the deposit is maintained [649–653]. However, for successful adhesion, a speed of the deposited particles must exceed a critical velocity, which depends on the properties of the thermal spray material [649]. Composite materials containing $CaPO_4$ has been shown to be good candidates for CS deposition [654–656].

To finalize this subtopic, it is necessary to mention the development of a new aerosol cold spray (ACS) $CaPO_4$ deposition process that combines several advantages of AD and CS technologies. According to the authors, the disadvantages of AD technology, which takes place at ambient temperature, can be overcome by using radial injection into the divergent zone of a Laval-type nozzle, as used in a low-pressure CS. In this case, gas heating of 500–600 °C is possible and the parameters of the fluidized bed aerosol chamber (gas pressure, oscillation amplitude and frequency) can be controlled over a wide range [657].

### 4.4.8. Blast Coating

Abrasive blasting is the process of pressing abrasive material against a surface at high pressure to smooth rough surfaces, roughen smooth surfaces, shape surfaces or remove surface contaminants. Pressurized fluids (usually air) or centrifuges are used to propel the blasting material. Researchers who have studied this process in detail have discovered that blasting media can adhere to the surface. This is because when particles hit the surface of

the substrate, both their momentum and kinetic energy are transferred. When this energy is absorbed by the substrate surface, it melts on a microscopic scale, immobilizing the particles. The area of lattice defects is also observed to increase. The sizes of the melted and lattice defect areas depend on the material properties of the substrate material and the energy of the detonated grains [658].

Therefore, the idea of using this property to deposit $CaPO_4$ on metal (Ti and its alloys) substrates was introduced [659–665]. This process was carried out using a room temperature shot blasting machine and HA powder alone [659,660,662], or a mixture of HA powder and blasting media (CoBlast$^{TM}$ process developed by ENBIO (Dublin, Ireland)) [662–665]. Furthermore, blast coating can be realized with a composite ceramic consisting of an alumina core (carrier material) coated with a porous HA shell [661]. As a result, the surface of the metal substrate is completely and homogeneously coated with HA. The researchers noticed that the deposited HA particles stick together as if sintered at room temperature. The $CaPO_4$ deposits were stable to sonication in water for at least 5 min and hard to remove by scratching with a fingernail. All coated surfaces showed higher adhesion strengths compared to plasma-sprayed HA, which can be attributed to a combination of mechanical and chemical bond formation. In addition, the composition of the HA deposits was found to match the composition of the raw HA, while the coated substrates showed promising in vivo results [659,660,662]. Similar findings were discovered with the CoBlast$^{TM}$ process [662–665]. Regarding known improvements, there is ball peening deposition, which differs from the CoBlast$^{TM}$ process only in the nature of the blasting medium (silica is used for ball peening, alumina for CoBlast$^{TM}$) [666], and super-high-speed (SHS) blasting at high speed (100 m/s) resulting in finer HA particles for fragmentation on the substrate surface. The latter resulted in the formation of more uniform and finer HA deposits with improved roughness, crystallinity, wettability and bond strength [667].

Furthermore, BrainBase Corporation (Tokyo, Japan) introduced the development of an ABS process (ABS = Apatite Blasted Surface) in which Ti dental implants are blasted with biphasic $CaPO_4$ (HA + β-TCP) bioceramic powder and ultrasonically cleaned in pure water for a long time. Although there is very little trace of $CaPO_4$ on the Ti surface (chemical analysis of the blasted surface showed that less than 1% $CaPO_4$ remained over the entire surface), this value seems to be sufficient to improve biocompatibility. The blasting of metallic dental implants with biphasic HA + β-TCP bioceramic powder (BCP$^®$ surface treatment) was also developed by Anthogyr (Sallanches, France) [668]. Both techniques produce a biocompatible rough surface free of contaminants, and guarantee fast and effective osseointegration. Finally, the Ossean$^®$ surface developed by Intra-Lock International (Boca Raton, FL, USA) is a resorbable intermediate blasting surface for metal implants and is also treated to contain low amounts of $CaPO_4$ to form fractal structures.

To finalize, blast coating appears to be similar to the aforementioned CS and, perhaps, both techniques could be combined.

### 4.4.9. Direct Laser Melting

First, the starting $CaPO_4$ powder is thoroughly mixed with a solvent to form a suspension. Afterwards, the suspension is deposited onto the substrate surface using any suitable technique such as spray, spin or dip coating techniques. The coated substrate is then air dried to remove moisture and the $CaPO_4$ deposit is directly laser melted using a continuous wave or pulsed laser beam to form a strong bond between the deposit and the substrate. The surface texture of the resulting deposit can be controlled by varying the overlap of the laser dots while varying the speed of the laser movement. Unfortunately, due to the high process temperatures required to melt $CaPO_4$, thermal decomposition of the latter occurs. Thus, the residue is always a complex mixture of various phases and/or compounds [179,669–673]. A schematic configuration of the direct laser melting system can be found in the literature [179].

### 4.4.10. Transmission Laser Coating

The transmission laser coating process appears to be analogous to the direct laser melting process described hitherto. The difference between them lies in the material melted. That is, while direct laser melting melts the $CaPO_4$ deposit, transmission laser coating melts the substrate surface. The latter process is based on the significant difference in light interaction between ceramics and metals. That is, many ceramics, including $CaPO_4$, appear opaque in UV and transparent in IR, while the laser absorption of metals is highest in the IR region. Thus, infrared laser beams can easily penetrate the deposited $CaPO_4$ with little or no heating, and melt the surface of the metal substrate. After solidification, the $CaPO_4$ particles appear to be squeezed by the metal. This process has been found to provide both strong coating/substrate interface strength and low processing temperatures to limit $CaPO_4$ degradation [674]. When supersaturated $CaPO_4$ solution is used instead of deposited $CaPO_4$, this technique is referred to as laser-induced hydrothermal synthesis (Section 4.4.16. below).

### 4.4.11. Laser Cladding

This method has striking similarities with the aforementioned direct laser melting and transmission laser coating techniques. Depending on the powder feeding method, laser cladding can be classified as a pre-deposition process or a simultaneous feeding process. The former is similar to direct laser melting, while the latter is a technique in which $CaPO_4$ particles or precursors are injected into a carrier gas stream. This powder stream is then injected into the area to be irradiated by the laser beam. The laser beam heats the deposition material, creating a molten pool on the metal substrate where the particles impact. A shielded inert gas is usually applied to prevent oxidation in the interaction area. Rapid quenching occurs as the molten pool leaves the laser irradiated area and deposits on the substrate [675–677]. Sometimes, this deposition process is called "laser surface alloying" [678].

Other than $CaPO_4$ themselves [679], precursors such as DCPA/DCPD and $CaCO_3$ mixed powders can be used for laser cladding [678,680–686]. Depending on the Ca/P ratio, well crystalline HA (when Ca/P ~ 1.67), $\alpha$-TCP, $\beta$-TCP or their mixtures (when Ca/P ~ 1.50), and complicated mixtures of TTCP, $\alpha$-TCP, $\beta$-TCP, ACP, HA, $Ca_2P_2O_7$ and CaO can be prepared when the ratio deviates from both values. Furthermore, when deposited on metals (e.g., Ti), other mixtures such as $CaTiO_3$ can be formed through oxidation. Porous deposits are also prepared due to the gaseous by-products ($CO_2$ and water vapor) produced by the reactions between the precursors. As the laser power was increased, the amount of TTCP, HA and CaO in the deposits gradually decreased and eventually only $\alpha$-TCP and $CaTiO_3$ remained. However, since TTCP and $\alpha$-TCP are completely converted to HA, the amount of HA can be significantly increased by sintering at 800 °C for 5 h after deposition followed by furnace cooling [680–686]. A computer-controlled version of the laser cladding technique called "laser rapid forming" has been able to deposit films on complex geometries [687]. Furthermore, materials containing ion-substituted $CaPO_4$ [688] and $CaPO_4$-based composites [686,689] can be deposited by this technique.

### 4.4.12. Laser-Engineered Net Shaping (LENS™)

LENS™ (Optomec, Albuquerque, NM, USA) is a commercially used rapid prototyping process developed in 1996 [690] and used to deposit $CaPO_4$ on metal substrates [691–693]. It also has striking similarities to the direct laser melting, transmission laser coating and laser cladding techniques mentioned above; unfortunately, full details of this process have not been published. According to the known information on the LENS™ process, laser power is focused on the substrate, creating a molten layer on its surface. $CaPO_4$ powder is then injected onto the molten metal layer from the deposition head using a carrier gas (Ar) to melt it. As the laser head advances, the molten metal and $CaPO_4$ rapidly solidify. The $CaPO_4$ particles solidified in this way appear to be embedded in the surface of the metal substrate and the biocomposites form. The entire substrate is scanned back and forth,

creating a pattern of $CaPO_4$ deposits with a finite thickness. This procedure is then repeated many times until the desired thickness is prepared. The deposition process can be controlled by varying the laser power, scanning speed and powder feed rate. $CaPO_4$/metal composite deposits have been produced by this technique [691–693]. The formation of calcium titanate due to partial oxidation of the Ti metal was detected [691]. To limit oxidation, the LENS™ process can be performed in a controlled, oxygen-free atmosphere. In the case of stainless-steel substrates, a small amount of $Fe_3P$ phase was found on their surface [693]. A schematic setup of the LENS™ deposition system can be found in reference [691].

As good biocompatibility of deposited $CaPO_4$-Ti biocomposites was found [691,693], the LENS™ deposition system was combined with an RF-induced plasma spray process [694]. The advantages of such a combination were clear: the laser-processed compositionally graded Ti-HA interlayer acted as a strong diffusion interface between the HA coating and the Ti substrate, eliminating problems associated with sharp interfaces, and improved the crystallinity of HA deposits due to lower cooling rates. In vitro biocompatibility studies with human fetal osteoblasts revealed a significant increase in cellular activity compared to LENS™-TCP coated titanium substrates [694].

### 4.4.13. Matrix Assisted Pulsed Laser Evaporation (MAPLE)

The MAPLE process was developed at the US Naval Research Laboratory in the late 1990s as a gentler pulsed laser vaporization process for functionalized polymers as an alternative to PLD. In the traditional MAPLE process, the laser target consists of a frozen matrix of polymeric compounds dissolved in a relatively volatile solvent. This is achieved by a cooling system in the chamber and instant freezing of the target by immersion in liquid nitrogen. The concentration of solvent and solute is chosen so that the target material is completely dissolved to form a dilute, homogeneous solution. The solvent is selected to absorb the tuned laser wavelength to prevent damage to the target, thus avoiding interaction between the laser and the material, and allowing undamaged, sensitive active material to be deposited on the substrate. As in conventional PLD, both solute and solvent molecules are released from the target region irradiated by the laser pulse and deposited on the substrate surface. Therefore, the MAPLE technique provides a smooth mechanism for the transport of various compounds, including species with large molecular weight, and is expected to provide improved stoichiometric migration, more precise thickness control and high homogeneity of the deposit. A schematic set-up of this process can be found in references [695,696].

Relevant to the topic of this review, the MAPLE technique has been used for the sensitive and accurate deposition of both $CaPO_4$ alone [696,697], and $CaPO_4$ and organic and/or biological materials (biocomposites) [698–701]. This example includes the deposition of CDHA hydrate [696] and ion-substituted OCP [697] as well as $CaPO_4$-based biocomposites with sodium maleate [698], alendronic acid [699], silk fibroin [700] and chitosan [701].

### 4.4.14. Liquid Phase Laser Deposition

This technique involves the nucleation of $CaPO_4$ precursors on the surface of various substrates by laser irradiation followed by wet chemical or biomimetic deposition. The latter is referred to as a "laser-assisted biomimetic process" (LAB process) [702,703]. This process has been applied to $CaPO_4$ deposition on various inorganic [704,705], polymer [129,706–708], metal [709–711], HA [712] and ivory [702,703] substrates. The experimental setup was an open system in which supersaturated precursor aqueous solutions containing $Ca^{2+}$ and $PO_4^{3-}$ ions (e.g., SBF) were poured into a container (Figure 24). The substrate was then immersed in the solution and irradiated with a laser beam at room temperature. The laser beam was focused through the solution onto the substrate surface. A scanning system generated a surface pattern of seven concentric squares with 200 μm intervals at the edges of the inorganic substrate, as seen in the upper left corner of Figure 24, while the center of the substrate was not irradiated and was used as a control. The processing mode of the pulsed laser can be both focused and defocused [713]. After irradiation, the

substrates were immersed in SBF under physiological conditions for 24 h. The results showed that ion-substituted CDHA rapidly deposited from SBF to the irradiated area of the inorganic substrate and did not deposit on the surface of the non-irradiated area. This was due to the formation of small $CaPO_4$ nuclei during irradiation, which promoted the growth of thicker deposits on the irradiated regions of the substrate [705]. Comparable consequences were obtained for polymeric [129,706–708], metal [709–711] and HA [712] substrates, indicating that $CaPO_4$ deposition is due to laser absorption by the substrate surface, which causes modification of the substrate and increases the temperature of the surrounding solution. $CaPO_4$ formation facility appeared to increase with increasing laser power and irradiation time. The overall deposition process was found to be simple, mild and site-specific [129,702–713].

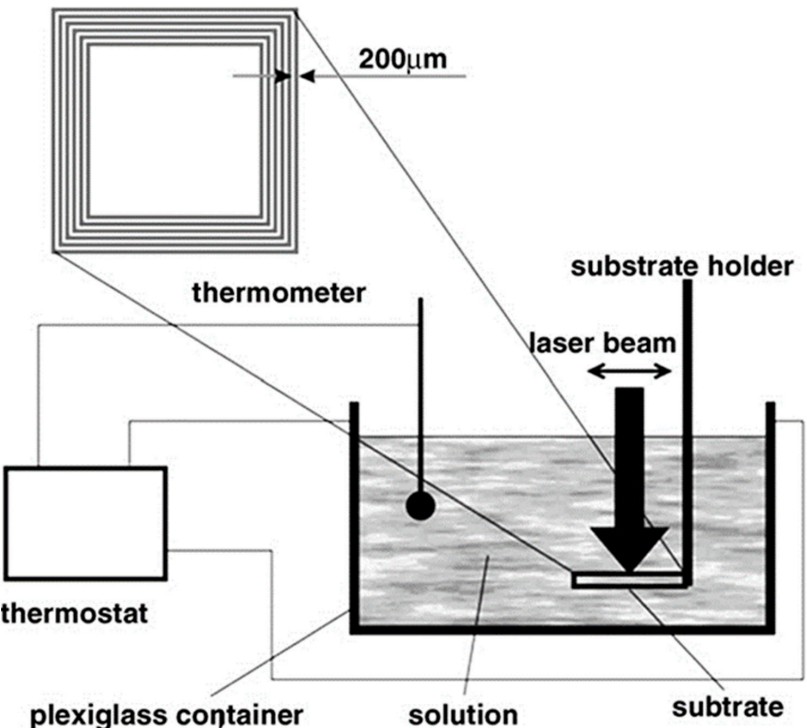

**Figure 24.** A schematic setup and a laser irradiation design (top left) used for a liquid phase laser deposition. Reproduced with permission from [705].

4.4.15. Laser-Induced Forward Transfer with Optical Stamp (LIFTOP)

LIFTOP deposition technology appears to be a further development of the previously described LAB process. Initially, laser-induced forward transfer (LIFT) technology was developed in 1986 as a simple and unique additive manufacturing method [714]. The LIFT process usually involves the irradiation of a single laser pulse through a transparent support onto a donor material or sacrificial layer that absorbs laser light, which causes heating, melting and ablation, and other laser-induced phenomena. In general, laser-induced changes cause a transient excitation field at high temperature and/or high pressure, thereby moving the donor material toward the acceptor substrate placed against the donor [715]. In 2020, a novel LIFT process, a soft shock-absorbing polydimethylsiloxane-coated transparent support, which the authors called LIFT with OPtical stamp (LIFTOP), was developed to imprint fragile $CaPO_4$ microchips onto a rigid acceptor substrate. The LIFTOP process was initially driven by a laser-induced process such as sacrificial layer ablation to drive the shock-absorbing layer. It was developed to enable the transfer of material from the optical stamp to the target substrate [715]. The description of LIFTOP deposition seems complicated, so let me quote the authors: "As an optical stamp, we used a transparent polyethylene terephthalate (PET) substrate coated with a polydimethylsiloxane

(PDMS) shock-absorbing layer. In this work, a carbon sacrificial layer was deposited on the PDMS shock-absorbing layer; this sacrificial layer effectively absorbed the laser pulse in place of the apatite without absorbing light at the laser wavelength. Prior to the LIFTOP process, a thin apatite film was prepared as a raw material on the optical stamp by a conventional biomimetic process (immersion in a $CaPO_4$ solution without laser irradiation). Then, the LIFTOP process was performed for the tooth substrate by irradiating the apatite film/carbon sacrificial layer/PDMS/PET (from the PET side) with a single laser pulse. As a result of the LIFTOP process, an apatite microchip with a shape corresponding to the laser beam spot was transferred onto the tooth substrate [715]. The processing time was fairly short; for example, about 80% of a 1-mm-square region was coated with apatite within about 0.1 s using a laser beam with a diameter of 100 mm, and a pulse repetition rate of 1 kHz. Cross-sectional observations showed that the transferred apatite microchip attached to the tooth surface without noticeable gaps" ([703] p. 220). To learn more about this deposition technique, interested readers are advised to read the publications mentioned above.

### 4.4.16. Laser-Induced Single-Step Coating (LISSC)

The LISSC deposition technique, also called laser-induced hydrothermal synthesis, appears to be a combination of liquid phase laser deposition and transmission laser coating, described earlier (Sections 4.4.10 and 4.4.14, respectively). In other words, as in liquid phase laser deposition, LISSC uses $CaPO_4$ supersaturated solutions (150 times higher than body fluids) and, as in transmission laser coating, the nanosecond laser beam melts the substrate surface. Thus, according to LISSC, $CaPO_4$ synthesis and deposition occur simultaneously through melting of the substrate (i.e., Ti, Mg, polyetheretherketone, polycaprolactone), forming gradient $CaPO_4$-substrate composite deposits (Figure 25) [716,717]. According to the authors: "The technique developed in this study shortened the existing multiple-step process to a single process, thus reducing the total process time from several days to a few minutes. By melting the substrate at the same time as HA nucleation by laser irradiation, a gradient HA–substrate intermediate layer in which HA penetrates into the substrate was observed. The coating thickness could be controlled from $20.3 \pm 4.6$ μm to $53.3 \pm 11.5$ μm, and it achieved a coating strength of 31.7–47.2 N, which is sufficient for application in medical implants." ([716], Conclusions).

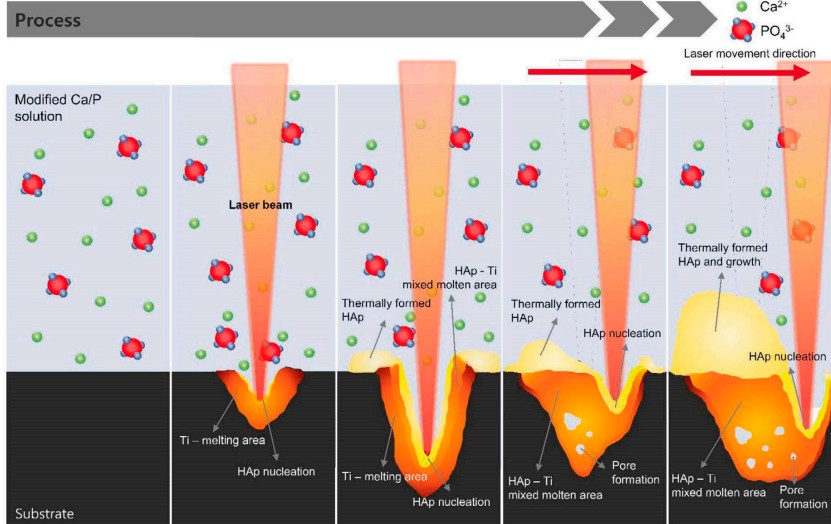

**Figure 25.** A schematic representation of $CaPO_4$ synthesis, $CaPO_4$-Ti substrate mixed melt layer formation and $CaPO_4$ deposition on Ti substrates using LISSC deposition technology. Green and red circles indicate $Ca^{2+}$ and $PO_4^{3-}$ ions, respectively. HAp indicates HA. Reproduced with permission from [716].

### 4.4.17. Electrostatic Spray Deposition (ESD) Technique

Electrostatic spray deposition (ESD, synonyms: electrohydrodynamic atomization (EHDA) spraying, electrostatic atomization, electrospraying) was developed in the early 1990s at the Institute of Inorganic Chemistry at Delft University of Technology (Netherlands) to produce fine porous ceramic deposits with controlled morphology for solid electrolytes and lithium batteries [718]. This method is based on producing an aerosol of an organic solvent comprising inorganic precursors under the influence of a high voltage. This is achieved by feeding a liquid through a nozzle. Micro-sized spherical droplets are usually formed at the nozzle's tips but, when a high voltage is applied between the substrate and the nozzle, each droplet deforms into a cone and spreads out forming a spray of the charged droplets. As a result, the droplets from the nozzle are dispersed in a spray and this spray adheres to the substrate, which is grounded and heated by the applied potential difference. The droplets therefore appear to lose their charge as they hit the substrate. After the solvent has completely evaporated, a fine residue remains on the surface. Processing parameters such as flow rate, concentration, voltage, distance from the nozzle to the substrate and relative humidity can be adjusted to control the droplet size and hence the final properties of the deposit. An overview of the setup of ESD technology can be found in the literature [304,719–723].

To perform ESD, soluble calcium salts (acetate, chloride or nitrate) and phosphate compounds ($H3PO_4$ or triethyl phosphate) are dissolved in alcohol. The resulting solution is pumped, rapidly stirred in front of a nozzle and electrostatically sprayed onto the substrate, which can be heated up to 300–450 °C [719–724]. Besides solutions, $CaPO_4$ powder can also be suspended in alcohol and the resulting suspension electrosprayed [725–729]. Of these, the solution method allows the deposition of finer precipitates compared to the suspension method. The chemical and morphological properties of $CaPO_4$ precipitates strongly depend on the composition of the precursor solution (absolute and relative concentration of the precursor, pH) and deposition parameters such as distance from nozzle to substrate, liquid flow rate, temperature and spray nozzle geometry. By varying those parameters, various phases and phase mixtures such as DCPA, α-TCP, β-TCP, carbonate apatite, β- and γ-calcium pyrophosphate, calcium metaphosphate, CaO and $CaCO_3$ have been deposited by ESD techniques [722–724]. Since ESD can take place at room temperature, thermally unstable compounds can also be deposited. As seen in Figure 26, electrostatically sputtered $CaPO_4$ deposits can be porous [723–726,729,730]. However, after deposition, the substrate can be calcined at high temperatures. The calcining step is necessary to agglomerate and/or melt the deposited $CaPO_4$ particles to form a dense and homogeneous coating. Template-assisted electrohydrodynamic atomization (TAEA) appears to be a further development of the ESD technique, which allows the creation of patterned surfaces by using mask templates to selectively coat the substrates. Patterns of pillars and tracks of various dimensions of $CaPO_4$ were created by application of the TAEA technique [731,732].

To finalize this section, several types of combined techniques, such as sol–gel assisted ESD [733] and ESD vapor deposition [734], were developed. Furthermore, $CaPO_4$ deposits by ESD might be commercially available. For example, NanoMech (Springdale, AR, USA) patented a NanoSpray® coating system, which was developed based on ESD technology [735]. Further details on ESD can be found in the literature [304].

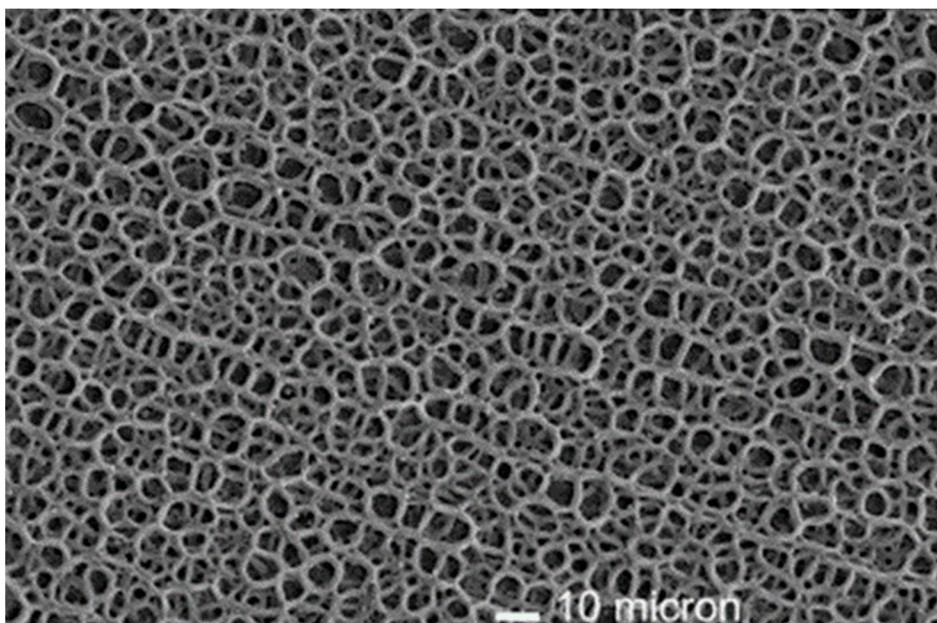

**Figure 26.** A scanning electron micrograph of electrostatic spray deposited CaPO$_4$ coatings characterized by a porous surface morphology. Reproduced with permission from [723].

### 4.4.18. Spray Pyrolysis (Pyrosol) Technique

Spray pyrolysis (pyrosol) deposition is a CVD technique similar to MOCVD (Section 4.2.5). The difference between the two lies in the evaporation system of the precursor solution, which is ultrasonic in spray pyrolysis and thermal in MOCVD. In the former, an aerosol is created [736–744]. For this reason, spray pyrolysis is sometimes called "aerosol CVD" [739] or "ultrasonic spray pyrolysis" [741,742]. As in MOCVD, organic precursors containing Ca and P can be used. For example, a solution of calcium bis(2,2,6,6-tetramethyl-3,5-heptanedionate) dihydrate and triethylphosphate dissolved in di(ethylene glycol) dimethyl ether is evaporated by sonication and, as in MOCVD, the resulting aerosol is heated and pre-fed into a reactor with a substrate in a container [739]. Ca acetylacetonate hydrate solution in aqueous solution of N,N-dimethylformamide and H$_3$PO$_4$ has also been used in some studies [743]. In addition, colloids containing nano-sized CaPO$_4$ can also be deposited [744]. Deposition can be carried out in both pulsed (intermittent) and continuous modes; the former reduce cracking and increase the binding strength of the CaPO$_4$ deposits [742]. Homogeneous, flat and dense CaPO$_4$ deposits were formed which could be continuous [742,743] and porous [736,737].

### 4.4.19. Polymeric Deposition Route

This is a CVD technique and quite similar to the molecular precursor and thermal decomposition method (Section 4.2.6), where a mixture of Ca- and P-containing organic precursors is deposited on a substrate in the desired Ca/P ratio, followed by drying, baking and sintering. The difference lies in the intermediate formation of a viscous polymer solution into which the substrate is immersed. In other words, for polymer deposition, phenyldichlorophosphine (C$_6$H$_5$PCl$_2$) was first mixed with acetone and hydrolyzed with water. Then, the exact stoichiometric amount of Ca(NO$_3$)$_2$ solution dissolved in acetone was added. The resulting mixture was then oxidized with air bubbles; after 1 h, an increase in viscosity was observed, indicating polymerization. The viscous solution was used to dip-coat various thermally stable substrates and then dried and baked. As a result, thin, almost completely dense deposits were produced with X-ray diffraction patterns showing the presence of HA and β-TCP [745].

### 4.4.20. Atomic Layer Deposition (ALD)

ALD is a variant of conventional CVD methods where reactive precursors are fed sequentially; ALD is based on self-limiting growth. This technique is particularly suitable for the preparation of microelectronic and nanoscale materials and for coating 3D structures where precise film conformability is required. $Ca(thd)_2$ (thd = 2,2,6,6-tetramethyl-3,5-heptanedione) and $(CH_3O)_3PO$ were used as Ca and $PO_4$ precursors, respectively, to deposit $CaPO_4$ on Si (100) and corning (0211) substrates by ALD. $CaPO_4$ were deposited in a commercial reactor at a pressure of 2–3 mbar; $Ca(thd)_2$ was evaporated at 188 °C. One-second pulses and purges were used for all precursors. Ozone produced from $O_2$ (99.999%) in an ozone generator was used as an oxygen source to oxidize $Ca(thd)_2$ after each pulse. Nitrogen (99.999%) was used as a carrier gas and purge gas. Water vapor from a container of water kept at constant temperature was used as the oxygen source for $(CH_3O)_3PO$ without any additional bubbling system. $CaPO_4$ membranes contained high levels of $CaCO_3$ impurities but annealing with a wet $N_2$ stream reduced the carbonate content even at 400 °C. $CaPO_4$ films were amorphous (ACP), but rapid heat treatment promoted the formation of carbonated CDHA [746]. Due to the large amount of by-product $CaCO_3$ formed during ALD deposition, the technique was converted to a two-step process. First, pure $CaCO_3$ was deposited by replacing the $PO_4$ precursor with a $CO_3$ precursor and then the $CaCO_3$ deposit was converted to $CaPO_4$ by immersion in 0.2 M $(NH_4)_2HPO_4$ solution at 95 °C [747,748]. See Section 6. Conversion-formed $CaPO_4$ deposits for additional examples.

### 4.4.21. Drop-On-Demand (DOD) Micro-Dispensing Technique

DOD micro-dispensing method was introduced in inkjet printers. According to this methodology, inkjet printheads digitally project tiny ink droplets (several picoliters) through nozzles to produce images directly on the substrate [749]. Lately, DOD microdispensing has been applied to deposit both pure and doped $CaPO_4$ coatings on numerous substrates [750–752]. A schematic setup of this technique might be found elsewhere [752]. Briefly, the deposition system consists of a microvalve drive, a pneumatic system, a precision XYZ stage controller, a dispensing unit, a vision system and an external heating patch. A microvalve printhead with an inner diameter of 300 μm is necessary to generate droplets at an applied constant pressure of 2 bar and a run time of 500 μs. The precision XYZ stage controller is necessary to achieve a droplet movement. The results showed that the DOD microdispensing technique did not change the properties of $CaPO_4$ powder during deposition. There was no observable change in the morphology of the powder after deposition. The deposits are phase pure, retain all functional groups and exhibit rod-like particles of ~60 nm in length and ~15 nm in width [751,752]. Uniform $CaPO_4$ deposits with a thickness of $34.5 \pm 1.0$ μm and a critical load of 69 mN before fracture [752], and ~3.5 μm thickness and a critical load of 160 mN before fracture were produced [751].

### 4.4.22. Vapor Diffusion Sitting Drop Micro-Method (VDSDM)

A VDSDM was developed using a device called a "crystallization mushroom" (Figure 27) [753]. The deposition process was carried out on the surface of mica sheets at room temperature. In each experiment, 12 mica sheets were placed concentrically in the upper chamber of the crystallization mushroom; a droplet of a solution formed by mixing 20 μL 50 mM $Ca(CH_3COO)_2$ and 20 μL 30 mM $(NH_4)_2HPO_4$ (Ca/P = 5/3) was placed on each sheet. Then, 3 mL of 40 mM $NH_4HCO_3$ solution was placed in the reservoir chamber. Both the glass lid and the upper reservoir of the corks were sealed with silicone grease. The experimental periods were 1, 7, 14 and 21 days. When the corks were closed and sealed, $NH_3$ and $CO_2$ released from the underlined $NH_4HCO_3$ solution slowly diffused into the upper chamber and dissolved in the solution droplets covering the mica layers, causing the pH to rise. This pH increase led to heterogeneous nucleation and crystallization of $CaPO_4$ on the mica layer surface. After the experiment, the corks were opened, washed with deionized water and allowed to dry at room temperature or 120 °C for 1 day. As a result, a deposit containing a mixture of OCP and CDHA was prepared in which the

amount of OCP increased with time. Similar experiments in the presence of polyacrylic acid resulted in the formation of ACP deposits [753].

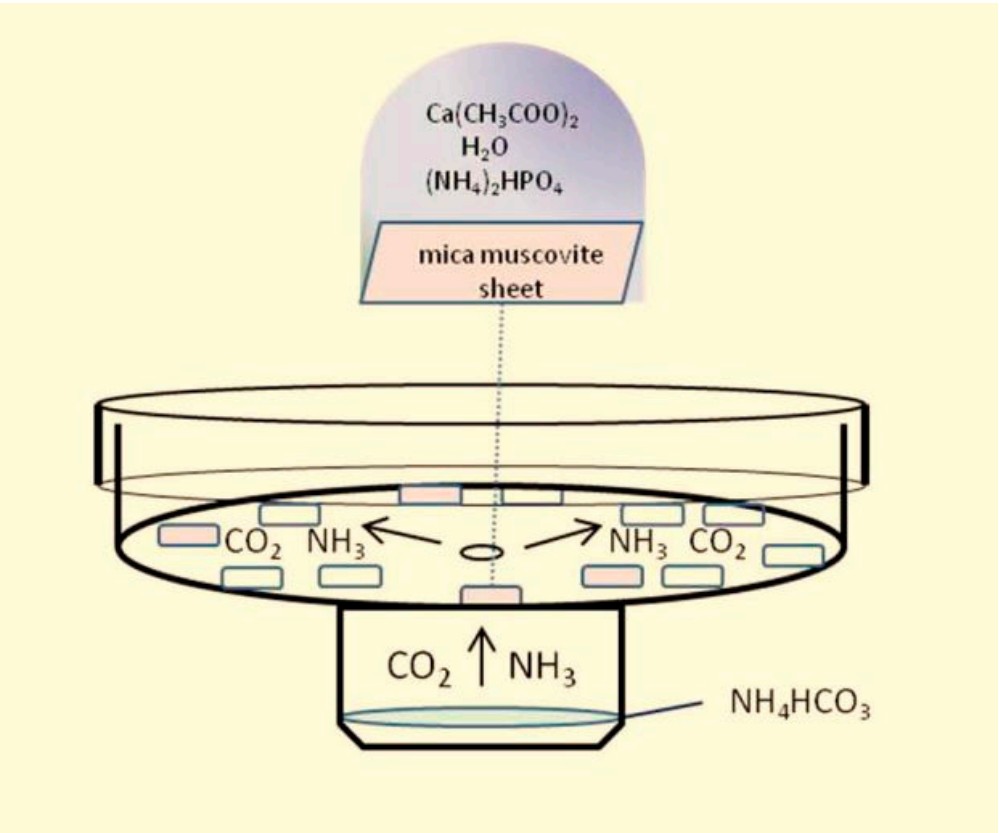

**Figure 27.** A crystallization mushroom set up for $CaPO_4$ deposition on mica muscovite sheets by VDSDM. Reproduced with permission from [753].

### 4.4.23. Mechanochemical Synthesis or Ball Impact Method

A highly unusual mechanochemical deposition method of $CaPO_4$ has been proposed in just two publications [754,755]. In both papers, the authors performed $CaPO_4$ deposition on metal substrates using high-energy ball milling. In the first study, Ti powder (250 μm), HA powder and $10 \times 8 \times 2$ mm Ti alloy substrates were used. The powders and substrates were placed in a vibrating steel flask (50 Hz) and processed for different times with an optimum ball-to-powder ratio of 40:1. The grinding process was carried out without any process control agents. The chamber was sealed to avoid contamination from the atmosphere. As a result, 50 μm thick deposits were obtained [754]. Similar results were obtained in the second study [755]. The authors discovered that the impacts of grinding balls significantly reduced the size of HA particles and activated the metal surface, resulting in tight cold welding of HA particles to the metal surface [754,755]. Nothing on the properties of the prepared $CaPO_4$ deposits has been reported.

### 4.4.24. Mechanofusion

Similar to the mechanochemical synthesis described above, mechanofusion systems use strong mechanical energy to trigger mechanochemical interactions between particles to form composite or coated particles with new properties. In other words, when fine cohesive powders are mixed with coarser-grained materials, the structure of the resulting mixture consists of a layer of fine powder attached to the surface of the larger particles. The mechanofusion (mixer) equipment consists of stationary internal parts and an internally rotating housing. The raw material is fed into a rotor with a fixed press head. Due to the rotation of the rotor, the powder is pressed against the rotor wall by centrifugal forces. The

powder is transported by the rotor wall to the stationary press head, where it is subjected to numerous compression and shear forces [756].

The process was applied to deposit HA particles on the surface of polymer (nylon, epoxy or polyethylene) beads (average particle size ~200 pm) [757]. Each mixture was processed in a mechanofusion system for 90 min. Among them, nylon beads showed the best coating efficiency with HA and a favorable specific gravity. To determine the optimum mixing ratio, HA coated beads with four different composition ratios were prepared from mixtures of nylon beads and HA powder with weight ratios of 100:0.5, 100:1, 100:2 and 100:3. Vero cells were cultured on them; HA-coated nylon beads with a weight ratio of 100:1 emerged as the most suitable cell carrier due to their specific gravity and the ability of Vero cells to grow effectively on them [757].

### 4.4.25. Autocatalytic Deposition

This deposition method appears to be a variant of the wet-chemical deposition method from supersaturated aqueous solutions (Section 4.3.4). The presence of $PdCl_2$ as a catalyst and $NaH_2PO_2 \cdot H_2O$ as a reducing P-containing compound is a feature of autocatalytic deposition. Two types of Ca-containing solutions for autocatalytic $CaPO_4$ coating were investigated: weakly alkaline (pH ~9.2) and weakly acidic (pH ~5.3). The process was based on the catalytic oxidation of hypophosphite ions to orthophosphate ions. Once produced, the latter ions reacted with dissolved $Ca^{2+}$ ions to precipitate CDHA on the polymeric substrate [49,758,759]. Shortly afterwards, this process was applied to the precipitation of $CaPO_4$ on metals [52,759]. In this case, the first authors used $H_3PO_4$ as P source and $PdCl_2$ or AgCl as catalyst [52], while the second authors used only a weak acid bath [759]. The proposed autocatalytic deposition mechanism is shown schematically in Figure 28.

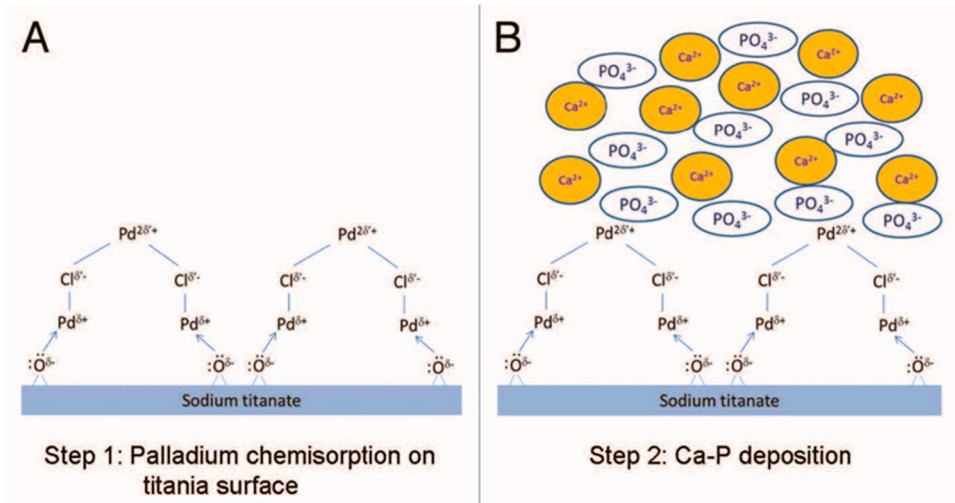

**Figure 28.** A hypothetical mechanism of $CaPO_4$ precipitation on Ti surface using $PdCl_2$ as catalyst. Note that an intermediate layer of sodium titanate has been formed on Ti prior to $CaPO_4$ precipitation. Reproduced with permission from [52].

An autocatalytic precipitation method without the use of a catalyst was also developed [760]. The authors used $CaCl_2$ and $NaH_2PO_2$ as Ca- and P-precursors, respectively, and sodium succinate $C_4H_4Na_2O_4 \cdot 6H_2O$ as reducing agent. All chemicals were dissolved in distilled water and the solutions were treated at pH 5.5 and 6.0 (slightly acidic), temperatures of 80 and 90 °C, and two immersion times of 60 and 180 min. The deposits were found to be a three-phase mixture of carbonate-substituted CDHA, chlorapatite and $CaCO_3$ [760].

### 4.4.26. Galvanic Deposition

Galvanic (synonym: sacrificial anode) deposition is based on a simple substitution reaction by galvanic contact of two metals with different redox potentials [761,762]. To

perform deposition, two metals with different redox potentials are connected by a conductive wire and placed in an electrolyte to form a single galvanic cell. In this cell, the more active metal acts as the sacrificial anode and the less active metal as the cathode, while ions of calcium and orthophosphate must be present in the electrolyte for $CaPO_4$ deposition to occur. The dissolution of the sacrificial anode generates electrons that flow to the cathode via an external ohmic junction. The reduction reaction occurring at the cathode causes a local increase in pH (a base electrogenesis) and the nucleation and precipitation of $CaPO_4$ (Figure 29). The electrochemical reactions leading to an increase in pH and $CaPO_4$ precipitation are well described in the literature [762]. Thus, galvanic deposition is similar to the constant current (galvanostatic) mode of conventional electrochemical deposition but, unlike the latter, galvanic deposition does not require an external power supply. Like electrochemical deposition, galvanic methods do not require expensive equipment or reagents, and can deliver the desired compound on the substrate surface in a short enough time.

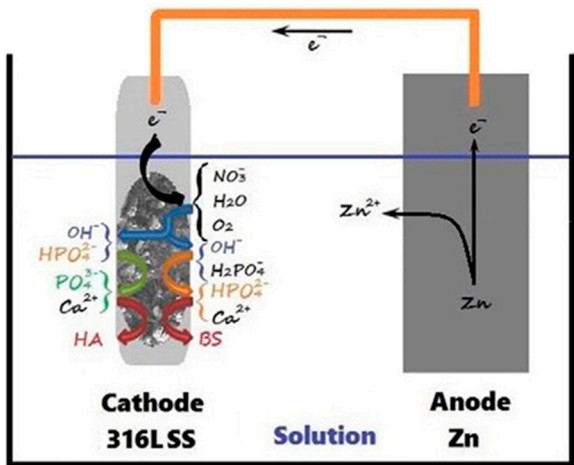

**Figure 29.** A schematic diagram of the galvanic deposition method. In this case, the sacrificial Zn anode is dissolved and $CaPO_4$ is deposited on the surface of the cathode made of commercial stainless steel 316LSS (<0.03% C, 16–18.5% Cr, 10–14% Ni, 2–3% Mo, <2% Mn, <1% Si, <0.045% P, <0.03% S and Fe in equilibrium). Reproduced with permission from [762].

Using an electrolyte containing 0.061 M $Ca(NO_3)_2 \cdot 4H_2O$, 0.036 M $NH_4H_2PO_4$ and 0.1 M $NaNO_3$, the authors carried out deposition at pH = 4.2 at three different temperatures (6 °C, room temperature and 50 °C) and three different deposition times (1, 3 and 7 days). Zn was chosen as the sacrificial anode and deposition was carried out on a commercial stainless steel 316LSS surface (Figure 29). At such an acidic pH, DCPD is in stable phase (Table 1) and its presence was detected in all deposits regardless of the deposition time and temperature. However, at 25 °C, traces of CDHA were detected only after a long deposition time and only after one day when deposited at 50 °C [762].

4.4.27. Anodization Technique

The anodization technique looks rather similar to ECD but, contrary to ECD, which is also called "cathodic deposition", this is an anodic deposition technique. For example, anodic oxidation of a magnesium alloy AZ31B substrate was performed using a platinum cathode with SBF solution as the electrolyte. The process was carried out at a constant current of 30 mA and temperature of 20 °C. The initial pH of the electrolyte was 7.4 and the final pH after 15 min of anodic oxidation was 7.8. To maintain the same pH value, the electrolyte was renewed after each 15-min anodization [763]. When the same process was performed at higher temperatures of 37, 50, and 80 °C, the pH values of the electrolyte increased from an initial 7.4 to 11.5 after the temperature rose to 50 and 80 °C [764]. The anodized deposits were found to consist mainly of the MgO phase intermixed with ion-substituted CDHA.

### 4.4.28. Simultaneous Precipitation and Electrodeposition

As seen from the title, this deposition technique appears to be a combination of ECD with a wet-chemical deposition and both deposition processes occur simultaneously. Namely, according to this method, one of the two precursors (either $Ca(NO_3)_2$ or $(NH_4)_2HPO_4$) is present in an electrolysis cell, while the other one is added drop-wise simultaneously with the application of an electrochemical potential. By alternating the addition order of the precursors and solution temperature, as well as the deposition time, $CaPO_4$ deposits that display a higher crystallinity compared to those obtained via the standard ECD can be obtained [765]. With some exceptions, major advantages of using this method are: better crystallinity, a more uniform and continuous surface, greater roughness and potentially higher anti-corrosion capabilities [766].

### 4.4.29. Electrical Stimulation

Electrical stimulation appears to be one more variation of the ECD process, in which the anode consists of a bone specimen, while the cathode is a metal, such as Ti or a vascular specimen connected to Ti. PBS was used as an electrolyte (presumably, other Ca- and $PO_4$-containing simulating solutions could be used as well). By this method, $CaPO_4$ was deposited on a Ti surface by electrically extracting the mineral components from a bone placed on the anode and re-crystallizing them on the surface of Ti cathode. The prepared $CaPO_4$ deposits were found to exhibit an XRD pattern resembling that of bone and, most importantly, contain trace elements. According to the authors: "This method requires only electrical stimulation and, therefore, it is a convenient method for collecting and supplying apatite as a biomaterial compared with conventional synthetic methods" [767].

### 4.4.30. Cyclic Electrodeposition

Cyclic electrodeposition (synonym: cyclic ECD) combines alternate soaking (Section 4.3.9) and ECD technology. It is based on independent cycles of electrodeposition of $Ca^{2+}$ and $PO_4^{3-}$ ions. In this method, the substrate is alternately immersed in Ca- and $PO_4$-containing solutions and the current is reversed accordingly, so that both species can be deposited during a certain number of cycles. The main advantage of cyclic electrodeposition is that the individual variables of each step can be independently controlled and studied. This means that basic ECD parameters such as current density, solution pH and reaction time can be adjusted independently in Ca- and $PO_4$-containing baths [768,769].

Some 0.1 M aqueous solutions of $CaCl_2$ (adjusted to pH 5.5) and $Na_2HPO_4$ (adjusted to pH 7.2) containing ethanol were used for cyclic ECD of $CaPO_4$ on porous silicon. The silicon substrates were alternately immersed in Ca-containing and $PO_4$-containing solutions under the following conditions: 1 $mA/cm^2$ cathodic current density for Ca solutions and 1 $mA/cm^2$ anodic current density for $PO_4$ solutions, controlled for 30 s with a potentiostat. The substrate was washed with ethanol between each step for the total of 20 cycles of Ca and $PO_4$ deposition. All processes were carried out in ambient conditions but were followed by thermal annealing at 800 $^\circ$C for 1 h to increase the crystallinity [768,769].

### 4.4.31. Cyclic Spin Coating

Cyclic spin-coating, like cyclic ECD, represents a combination of alternate soaking deposition and spin-coating. The method involves sequential spin-coating of the substrate with Ca- and $PO_4$-containing solutions based on independent coating cycles. $CaPO_4$ deposition was carried out for 60 s at 2500 rpm using the solutions described for cyclic ECD. The deposition process took place at room temperature and was repeated for 10 consecutive cycles [769].

### 4.4.32. Biomediated Deposition Technique (Biosynthesis)

Biosynthesis of $CaPO_4$ is based on the use of calcifying bacteria and there is only one publication on the subject [770]. To perform $CaPO_4$ deposition, fresh saliva samples were collected and calcifying bacteria were isolated. The total number of viable bacteria

was counted by the standard pour plate method by means of modified Mueller Hinton agar medium. The composition of the modified agar medium was casein acid hydrolysate 17.5 g/L, bovine tallow infusion 2.0 g/L, starch solubility 1.5 g/L, $CaCl_2$ 1 g/L and agar 20 g/L. To induce $CaPO_4$ precipitation, 300 mL of broth was poured into a clean 500 mL conical flask and 10 mL of calcified bacterial culture was inoculated into the broth. The metal substrate was then immersed in the medium under sterile conditions and kept in an incubator at 37 °C for 48 h. After performing incubation, the substrate was removed from the medium, washed with distilled water to remove adhering bacteria and then gently heated to form CDHA precipitates. CDHA showed improved corrosion resistance and low passivation current density [770].

### 4.4.33. Emulsion Route

To realize the emulsion route, a stable oil-in-water emulsion containing $CaPO_4$ particles should be prepared. For this purpose, aqueous dispersions of nano-sized HA particles with a solid content of 0.04 wt% were prepared by serial dilutions; 25 g of these dispersions were then manually agitated with poly(L-lactic acid) (PLLA) solutions in dichloromethane (total 2.5 g, 1.0–9.1 wt% solids) at 25 °C for 30 s. As dichloromethane is a volatile solvent, HA-coated PLLA microspheres were prepared by in situ evaporation from the emulsion at 25 °C (Figure 30, right). Cell-based experiments revealed that HA-coated PLLA microspheres promoted cell adhesion and spreading compared to uncoated PLLA microspheres [771]. HA-coated PCL microspheres were prepared by the same technique [772]. However, for polymers with lower melting points such as PCL (melting point ~60 °C), a solvent-free emulsion route was developed by preparing water-stable molten emulsions containing $CaPO_4$ particles. To perform the deposition process by the emulsion route, an aliquot of the same dispersion of nano-sized HA particles was mixed with 0.125 g of PCL pellet in a glass tube. The tubes were then heated in a water bath at 80 °C for 1 h to dissolve PCL. To prepare the emulsion, the hot mixture was homogenized using a homogenizer at 20,500 rpm for 1 min. The emulsion was then cooled in air to the ambient conditions, resulting in the preparation of HA-coated PCL microspheres [773]. The same is possible with polymers with a melting point of 100–160 °C but in such cases a stainless-steel high-pressure reactor and oil bath should be used [773]. In addition, fillers such as nano-sized $Fe_3O_4$ can be incorporated into polymer microspheres prior to deposition of $CaPO_4$ particles using the emulsion route to obtain additional properties such as magnetism [774].

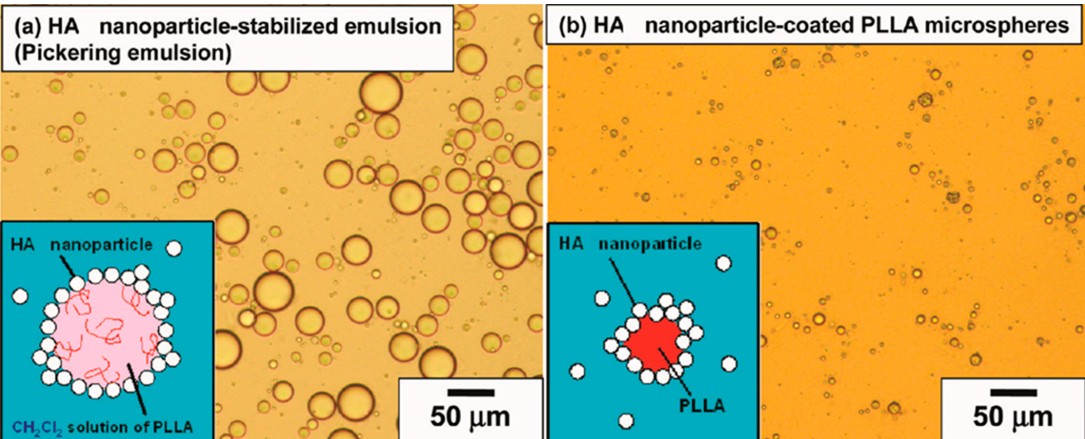

**Figure 30.** Representative optical micrographs (**background**) and schematic diagrams (**inset**) of dichloromethane emulsions of PLLA stabilized with nanoscale HA in water (**left**) and preparation of nanoscale HA-coated PLLA microspheres after dichloromethane evaporation (**right**). Reproduced with permission from [771].

### 4.4.34. Slurry Processing Technique

Slurry processing was carried out to deposit $CaPO_4$ on Ti. As the name suggests, a $CaPO_4$ slurry had to be prepared first: fine $CaPO_4$ powder and distilled water were mixed in approximately equal proportions. A Ti substrate was then fully embedded in the slurry in a ceramic crucible and heated in air at 450–750 °C for 2 h. This heating oxidizes the Ti surface and promotes the inter-solid diffusion of titanium oxide and $CaPO_4$, simultaneously fixing $CaPO_4$ deposits on the surface. After heating, the substrate was removed from the slurry, sonicated in distilled water for 10 min and dried in air at 60 °C. Composite gradient deposits of HA and $TiO_2$ with a thickness of 200–1000 nm were prepared on Ti substrates. At high temperatures, both the roughness and thickness of the $TiO_2$ layer increased as oxygen diffused more easily into the substrate [775,776].

### 4.4.35. Slip Coating Technique

Slip-coating deposition is largely similar to dip- and spin-coating techniques and slurry processing. Briefly, a $CaPO_4$ solution (suspension, slurry, etc.) is prepared and applied to the surface of a pre-sintered ceramic substrate with a soft brush, then dried and sintered to form deposits. Since the pre-sintered substrate has good water absorption, it is easy to form thicker $CaPO_4$ deposits. Numerous additives such as surfactants, dispersants and binders can be added to the $CaPO_4$ slips to improve adhesion. As with dip and spin coating, this process can be repeated several times to create thicker deposits. As a result, microporous $CaPO_4$ deposits with good adhesion strength can be produced [777–779].

### 4.4.36. Deposition by Solvent Evaporation

This simple and elegant method is based on dissolving HA in an "apatite dissolution solution" followed by evaporation of the solvent. First, an "apatite dissolved solution" should be prepared. To do this, 1 g HA powder was dispersed in 1000 mL of deionized water and with simultaneous bubbling of $CO_2$ gas for 1 h. $CO_2$ sparkling facilitated HA dissolution by lowering the pH from 11.6 to 5.6. The substrates (the authors used graphite [780] and polyetheretherketone [77]) were then immersed in the "apatite dissolved solution" and the whole system was heated with an external heater (90 °C for 1 h) or microwave oven (100 °C for 5–10 min) to simultaneously evaporate water and remove $CO_2$ by an external heater [780], and microwave heating to remove only $CO_2$ [77,780]. The combination of microwave and external heating techniques appeared to be effective in depositing large amounts of $CaPO_4$. This was due to the seed crystals formed by evaporation of solvent from the "apatite dissolution solution" by external heating facilitating $CaPO_4$ deposition by microwave heating. The deposit formed by 4.9 μm thick stacking of plate-like crystals was identified as carbonate-containing CDHA with a Ca/P ratio of 1.72. After implantation in the femur and tibia of Japanese white rabbits for four months, calcified bone formed at the interface and showed excellent biocompatibility [780].

### 4.4.37. Discrete Crystalline Deposition

This deposition technique appears to be quite similar to dip coating. Pure Ti or Ti-based alloy samples were immersed in an alcohol-based colloidal suspension of stoichiometric HA nanosized particles (size 20–100 nm, >95% crystalline, ACP was not detected) at room temperature. After removal from the solution, the samples were dried in an oven at 100 °C. As a result, about 50% of the metal surface was covered with $CaPO_4$ crystals, while the remaining surface was covered with metal oxides. Treatment of implant surfaces of titanium and its alloys with $CaPO_4$ impurity crystals was found to bond to bone [781]. Further surgical studies revealed that bone conduction was significantly enhanced when discrete nano-sized HA particles were deposited on Ti [782]. Unfortunately, it is still unclear why this technique results in discrete deposition of $CaPO_4$ crystals.

### 4.4.38. Powder Mixed Electrical Discharge Machining (PMEDM)

Electrical discharge machining (EDM, synonyms: wire burning, die sinking, spark eroding, spark machining, burning, wire erosion) is a material removal process that uses electrical discharge (sparks) to achieve the desired shape. Since electrical discharge machining involves no direct contact between the tool and the workpiece, conductive materials can be machined regardless of their hardness, shape or strength. The material is removed from the workpiece by a series of rapidly repeated current discharges between two electrodes separated by a dielectric fluid. The electric spark causes a localized increase in surface temperature and pressure above the boiling point of the material (up to 2000 °C and 200 bar respectively), creating a cratered surface. In addition, the dielectric liquid as coolant, brightener and insulator decomposes locally, its components react with the substrate components and the high-temperature generation simultaneously provides molten compounds. The addition of fine powder to the dielectric fluid is a relatively new development in the process known as PMEDM [783].

Several studies have been conducted on PMEDM treatment of various substrate surfaces by adding fine HA powder to water (dielectric fluid) [784–787], and it was found that the addition of HA significantly changes both the surface morphology and the chemical composition of the treated substrate, indicating that HA powder has a significant effect on the overall process. Furthermore, EDS analysis of the substrate surface clearly showed the presence of strong Ca and P peaks due to HA deposition. The authors therefore concluded that the PMEDM technique can be used for $CaPO_4$ deposition [784]. Other studies have also shown positive results. Namely, other researchers deposited $CaPO_4$-containing coatings on Ti-Ta alloys, which were converted into HA-containing coatings after hydrothermal treatment in an autoclave [785].

However, as with the MAO process (see Section 4.3.10. above), due to local extremes, the $CaPO_4$ deposits obtained from the PMEDM process are always a complex mixture of various products. In other words, to enhance the bioactivity of β-phase Ti-35Nb-7Ta-5Zr alloy, the PMEDM process was carried out in dielectric solutions containing fine (0.5–1 μm) HA powder. As a result, 18–20 μm thick deposits of Ti, Nb, Ta, Zr, O, Ca and P elements were formed and biocompatible phases such as TCP, $CaZrO_3$, $Nb_8P_5$, CaO, TiP, $Nb_4O_5$, $TiO_2$ and TiH were formed [786]. Similar complex mixed phases ($CaZrO_3$, $ZrO_2$, HA, ZrC and TiC) were also detected in Zr-based metallic glasses processed by PMEDM in dielectric solutions containing HA powder [787]. Regarding biomedical properties, bioactivity analyses have always shown that the $CaPO_4$-containing deposits produced by PMEDM promote cell attachment and high cell proliferation. More details on this technique can be found in a recent review [788].

To finalize this section, there is a gas-assisted electrical discharge coating technique, which appears to be a modification of the PMEDM through a combination of PMEDM, gas-assisted perforated electrodes and electrode rotation [789].

### 4.4.39. Investment Casting

This method is used for $CaPO_4$ deposition, as metal prostheses are usually produced by investment casting, where molten metal is poured into molds made of high-temperature resistant and chemically inert inorganic compounds (e.g., zirconia, silica, graphite and alumina) [790–793]. The technique is quite simple: the mold cavities must be coated with $CaPO_4$ before the molten metal is poured. Thus, an aqueous suspension of HA powder was prepared and applied to the cavity using a paint brush. When the water was completely dry, HA particles adhered to the cavity walls [790–792]. However, a more complex HA deposition technique has also been invented in which a wax pattern is used to deposit HA on the cavity walls followed by dewaxing [793]. The molten metal was then poured into a mold coated with HA, followed by cooling, demolding and recovery of the metal sample. This method appeared to be feasible and the cast samples were easily removed from the mold. $CaPO_4$ uptake was detected on the surface of the metal castings and analysis of the

deposits showed that they consisted of a mixture of various compounds, including α-TCP, β-TCP, HA and CaO [790]. Other studies have obtained the similar results [791–793].

### 4.4.40. Brush Painting

$CaPO_4$ deposition by brush painting is known as well. Namely, the coating process of Ti substrates with HA was carried out using a suspension of HA in oil of turpentine. The solid-to-liquid proportion was 1 to 3 (g/mL) and mixing was performed using a spatula. Then, the suspension was applied to the surface of Ti with a brush. Afterwards, the painted samples were heat treated in a furnace at 800 °C for 30 min [794].

### 4.4.41. Photocatalytic Deposition

This method is based on the photolysis of water-soluble orthophosphate anion precursors such as triethylphosphate, which takes place on the semiconductor surface in the presence of water-soluble $Ca^{2+}$ ions. Therefore, $CaPO_4$ can be deposited as a photocatalyst on surfaces with photocatalytic activity only when irradiated with UV light in aqueous solution. This activity was found to be possessed by $TiO_2$ in the form of highly ordered nanotubes. Therefore, to deposit $CaPO_4$ on the surface of the Ti substrate, this photocatalytic layer was first anodized. The deposition process was carried out using an electrochemical cell with a three-electrode setup. Platinum wire served as the counter electrode and an Ag/AgCl (saturated KCl solution) electrode served as the reference electrode. A Ti plate with a nanotubular $TiO_2$ layer grown on it was used as the working electrode. An aqueous solution containing triethylphosphate and calcium nitrate or calcium chloride was poured into the cell. The pH of the solution was adjusted with an aqueous solution of ammonia. The cell was equipped with a quartz window to irradiate the working electrode with UV light. The UV light focused from a high-pressure mercury lamp (250 W) irradiated the surface of the working electrode. As a result, CDHA was deposited [92].

### 4.4.42. Adsorption

Moreover, the deposition of $CaPO_4$ on specific substrates can be realized by adsorption. For example, to adsorb nano-sized HA crystals on the surface of PLLA microspheres, the microspheres were first treated with alkali (pH = 11.0, adjusted with 25% ammonia solution) for 1 h at room temperature, adding carboxyl groups to the surface, washed with water and dried under reduced pressure. The alkali treated and dried PLLA microspheres were then washed with ethanol and immersed in a 1.0% ethanol dispersion of nano-sized HA for 1 h at room temperature under shaking. The HA-coated microspheres were then washed five times with ethanol for 3 min under ultrasonic treatment and dried under reduced pressure. Due to the ionic interaction between the calcium ions of HA and the surface carboxyl groups of the alkali treated PLLA, the surface of the PLLA microspheres was found to be uniformly coated by HA [56].

### 4.4.43. Sonocoating

Sonocoating involves a physical interaction between the nanosized particles and the surface of the material, and creates a chemical bond between the particles and the substrate. To perform the deposition process, the substrate was immersed in a container containing a colloidal aqueous suspension of nano-sized HA at a concentration of 0.1 wt%. An ultrasonic horn (VCX750, 20 kHz, amplitude 50%) was placed 15 mm away from the substrate. The high-power ultrasound causes the growth of cavitation bubbles in the liquid, which implode at a critical size. This implosion energy pushes the nano-sized HA particles towards the surface of the material, where they adhere to the substrate. The process temperature was set at 30 ± 1 °C. The coating time varied between 1 and 8 min. After deposition, the samples were rinsed with demineralized water and dried [795]. A schematic of the method is shown in Figure 31 [796]. The sonocoating method allows homogeneous deposition under ambient conditions, in a short time and in water. It is a green technological process and has potential applications in bone tissue regeneration.



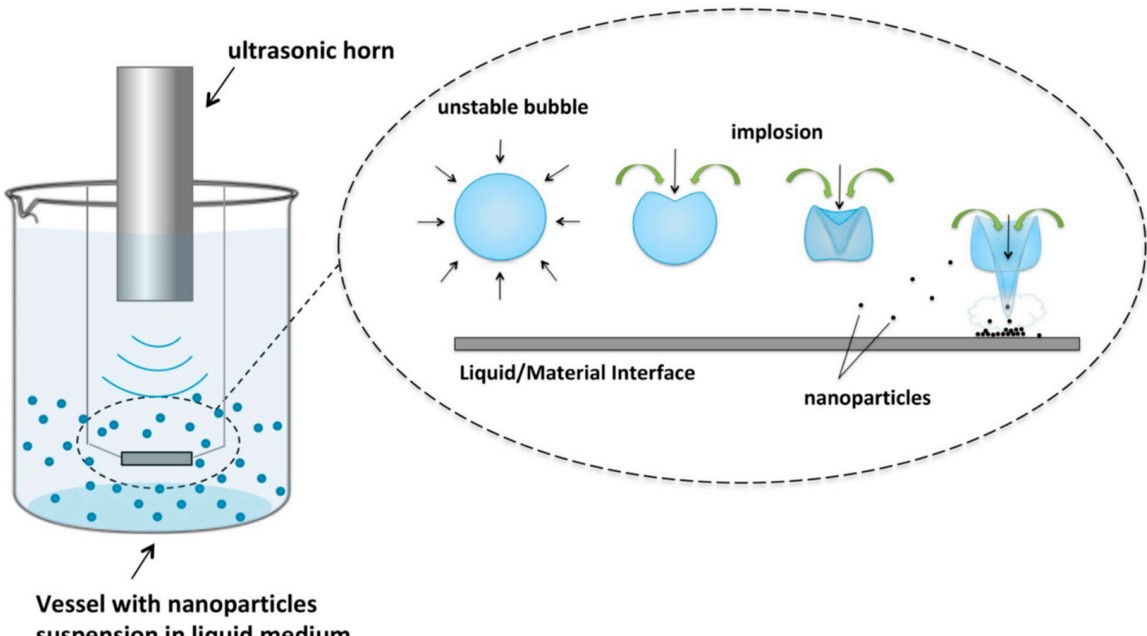

**Figure 31.** A schematic diagram of the sonocoating process. Reproduced with permission from [796].

### 4.4.44. Ultrasonic Mechanical Coating and Armoring (UMCA)

To deposit HA on Ti [797] and NiTi shape memory alloy [798] substrates, a high frequency ball bombardment method called the UMCA process has been developed. The deposition requires an ultrasonic device that delivers vibrations at frequencies above 20 kHz that rapidly induces the powder/ball effect on the substrate surface. The UMCA process involves the preparation of the HA slurry, pre-coating of the HA slurry onto the Ti plate and balling of the substrate to fix the pre-coating: impact treatment. Let me quote: "HA slurry obtained by mixing HA powder (1 μm) and ethanol was dropped onto Ti plates (Gr. 2, $10 \times 10 \times 1$ mm$^3$) and dried in air. Before pre-coating, the Ti plates were polished using #800 silicon carbide (SiC) paper, cleaned ultrasonically with ethanol for 10 min, and dried in air. The pre-coated Ti plates were fixed at the ceiling of the ultrasonic chamber, with the polished surfaces faced downward. The deposition process was performed using an ultrasonically assisted peening equipment supplied by SONATS. ZrO$_2$ balls with a diameter and weight of 1 mm and 1.5 g, respectively, were loaded into a Ti chamber. The bombardment of the ZrO$_2$ balls was initiated by a vibration generator at vibration frequencies and amplitudes of 20 kHz and 50 μm, respectively. The effects of the UMCA duration and pre-coating time on the deposition state were investigated. Pre-coating several times implies that both the pre-coating and ball-impact steps were repeated several times. After completing the UMCA process, the specimens were rinsed with alcohol and distilled water to remove the weakly attached powder" ([797], p. 5000).

### 4.4.45. Osteomimetic Deposition

Osteomimetic deposition appears to be a modification of biomimetic deposition (Section 4.3.4.) but using alkaline phosphatase (ALP) instead of SBF. Initially, in 2019, Polish researchers developed a CaPO$_4$ deposition technique on polycaprolactone (PCL) by means of its incubation with bovine intestinal ALP in sodium glycerophosphate and calcium chloride medium. To get an effective attachment of CaPO$_4$ deposits, the hydrophobic surface of PCL was modified by incubation at 50 °C (thermal activation) or at 37 °C after short exposition to lipase (lipase activation). An enzymatic hydrolysis of sodium glycerophosphate in the presence of calcium chloride was found to lead to a rapid formation of CaPO$_4$ deposits [114]. However, that method from 2019 did not contain HCO$_3^-$ ions in the media. In 2023, the same authors added HCO$_3^-$ ions to the phosphatase incubation

medium and called it "osteomimetic deposition" [799]. The addition of $NaHCO_3$ to the incubation medium produced A and B types of carbonate substituted CDHA. The authors concluded that "After further development, osteomimetic method will enable studies on the influence of bone specific phosphatases and various phosphorylated substrates on the HAP crystals formation and their deposition on the artificial scaffolds or collagen fibrils" (Ref. [799], p. 12).

### 4.4.46. Surface-Induced Mineralization (SIM)

Similarly to the osteomimetic deposition, SIM is also a modification of biomimetic deposition (Section 4.3.4.). There were just three papers on this topic published in 1996–2000 [800–802]; however, the basic paper on this approach was published in *Science* in 1994 [50]. SIM deposition is based on the observation that in nature organisms use various macromolecules to control the nucleation and growth of the mineral phase. Therefore, the SIM process mimics the idea of Nature's template-mediated mineralization by chemically modifying substrates to produce surfaces that are able to induce heterogeneous nucleation from aqueous solutions. After the surfaces of substrates have been chemically modified, they are placed into aqueous solutions containing soluble precursors of the desired ceramic material [50]. Namely, after several pre-deposition treatments, the authors performed surface modifications of Ti and its alloys with self-assembled monolayers by placing the substrates in a 1% si1ane solution in cyclohexane for 30 min, followed by rinsing in 2-propanol to remove residual silanes. Afterwards, the substrates were sonicated in chloroform for 5 min to produce mirror-bright surfaces and soaked in supersaturated aqueous $CaPO_4$-containing solutions, consisted of $KH_2PO_4$ and $CaCl_2$ with the addition of 0.01 M KOH to adjust the solution pH [800].

### 4.4.47. Ionized Jet Deposition (IJD)

IJD represents one more PVD technique based on pulsed electron ablation and seems to be a modification of EBAD (Section 4.2.1). IJD equipment was developed by Noivion Srl (Rovereto, (TN), Italy) and it employs a special pulsed electron source able to generate in vacuum ultra-short and electric discharges in the MW range. The discharge is supported by a gas jet and directed into a solid target, generating a superficial explosion and consequent emission of very energetic material in the form of plasma with the same composition as the target. The plasma emitted by the target produces a dense coating on any object it finds on its path with the same composition as the target or combined with a reactive background gas (reactive mode). To perform $CaPO_4$ deposition, three types of targets (HA, β-TCP and DCPD) were mounted on a rotating target holder and ablated by a fast pulse (100 ns) of high energy (10 J) and high-density ($109 \text{ W/cm}^2$) electrons. To ensure uniformity of deposition, the substrates were kept rotating during the whole process. The vacuum chamber was kept at a pressure of $2 \times 10^{-4}$ mbar and deposition was carried out in an oxygen flow. The working voltage and electron beam frequency were set to 18 kV and 7 Hz, respectively, and the deposition time was set to 30 min. The results showed that IJD provides tunable $CaPO_4$ deposits, the compositions of which resemble those of the deposition targets. All deposits were nanostructured comprising grains of about 80 nm in aggregate clusters up to 1.5 μm [803].

### 4.4.48. Undisclosed Proprietary Deposition Techniques

The IonTite[TM] method was developed by Spire Biomedical, Inc. (Bedford, MA, USA) in the mid-2000s. According to the literature: "The proprietary IonTite[TM] coating process was conducted at room temperature without additives and creates a uniform coating ~5–15 μm thick in less than 1 h. The IonTite[TM] process allows a coating to be applied conformably onto a wide range of substrates used for orthopedic, dental, and spinal applications" ([804], p. 266). In another publication, it is written that IonTite[TM] is a multi-step process [805]. In addition, there is a "transonic particle acceleration deposition process" also developed by Spire Biomedical, Inc. (Bedford, MA, USA) to deposit nanodimensional HA onto Mg

substrates [806], a description of which contains even less detail, when compared with that for the IonTite$^{TM}$ process.

### 4.4.49. Additive Manufacturing Techniques

Finally, any type of additive manufacturing techniques, including its subsets 3D printing and rapid prototyping, may be applied for $CaPO_4$ deposition as well [807]. As its name implies, additive manufacturing adds a material to create an object; therefore, these techniques are widely used to grow 3D objects by deposition of one superfine layer at a time. Here, each successive layer bonds to the preceding layer of a melted, partially melted or self-curing material. As the material cools or is cured, all layers fuse together to form a 3D object. Nevertheless, since all types of these techniques apply layer by layer deposition, one can take a substrate and put one or several layers of $CaPO_4$ onto it. However, to the best of my findings, nobody ever tried to deposit $CaPO_4$ onto a substrate by this approach.

## 5. Deposition of Ion-Substituted $CaPO_4$ and $CaPO_4$-Containing Biocomposites

Since bioceramics consisting from chemically pure $CaPO_4$ have some severe mechanical limitations [4–6], ion-substituted $CaPO_4$ [183,323,808–813] and $CaPO_4$-containing biocomposites [814–822] have been deposited on various substrates to improve their properties. Generally, such deposits can be prepared by two approaches: (i) direct deposition of the ion-substituted $CaPO_4$ and biocomposites or their precursors, and (ii) incorporation of the desired compounds and/or dopants into the already deposited $CaPO_4$. All of the above-mentioned techniques for deposition of the pure $CaPO_4$ can likewise be used for the deposition of ion-substituted $CaPO_4$ and biocomposites, although the real applicability of each technique is highly dependent on the nature and properties of the desired dopants and/or compounds. Namely, in the case of inorganic dopants, carbonates are thermally unstable, while chlorapatite is thermally stable. Therefore, it seems that no thermal sputtering technique is applicable for the deposition of carbonate apatite, which can be easily deposited by wet techniques, while the former can be easily deposited by plasma sputtering [823]. Furthermore, wet deposition techniques can also involve thermally stable additives. For example, consider the deposition of $CaPO_4$ doped with Ag or other metals. Namely, Ag-doped $CaPO_4$ has been deposited using sol–gel [824,825], electrochemical [826], electrophoretic [827,828] depositions, magnetron sputtering [829–831], thermal [832], plasma [833,834] and cold [835] spray, IBAD [836], PLD [837] and thermal substrate [838] techniques, while Sr [486,839], Zn [840] and Si [486,841] doped $CaPO_4$ were deposited using biomimetic techniques. However, this in no way implies that $CaPO_4$ doped with Sr, Zn, Si and other dopants can only be deposited by biomimetic techniques [833,842–844]. The same approach applies to the deposition of other calcium phosphate species such as calcium metaphosphate, calcium pyrophosphate and calcium polyphosphate [845]. $CaPO_4$-containing biocomposites can be produced by electrodeposition [846], dip coating [847], biomimetic methods ([848] and references therein) and hydrothermal-electrochemical co-deposition [849], as well as pulsed electrodeposition aided injection chemical vapor deposition [850]. In addition, MAO, MAPLE and AD techniques offer opportunities for the deposition of various composites. Namely, $CaPO_4$/hydrated titania biocomposites were deposited on Ti by MAO in combination with EPD [851] and nanocomposites containing diopside, bredigite and fluoride HA were coated on the surface of AZ91 Mg alloy treated by MAO using EPD technology [852]. Furthermore, various types of reinforcements can be added to $CaPO_4$ deposits [853]. However, due to the large number of possible dopants and/or additives and limited space, the deposition of both ion-substituted $CaPO_4$ and $CaPO_4$-based biocomposites will not be further elaborated, and interested readers are referred to the original publications.

For the incorporation of desired dopants into previously deposited $CaPO_4$ coatings, films and layers, sorption and ion exchange methods appear to be the most common ones. Electrodeposition can also be used [692]. For doping, $CaPO_4$-coated substrates are immersed in the solution for some time and then various post-deposition treatments are

applied. However, this method has several limitations. Namely, the incorporated dopants are usually located on the outer surface of the $CaPO_4$ deposit and are rapidly depleted both in vitro and in vivo without any long-term effects [692,843,854–858]. To improve fixation of dopants on the surface of $CaPO_4$ deposits, the process of depositing additional compounds can be carried out before the dopant deposition process [859].

## 6. Conversion-Formed CaPO₄ Deposits

Owing to a definite creation mechanism of the conversion-formed deposits, they can only form on the surface of substrates containing calcium or orthophosphate ions in their initial chemical composition. Therefore, for $CaPO_4$ to form on the surface, it must be treated with soluble chemicals containing the missing ions. Thus, Ca-containing solids are treated with an aqueous solution containing dissolved $PO_4$ ions. In other words, it has been found that the surface of marble (i.e., $CaCO_3$) samples treated gently and for a long time (72 h) with a dilute (1 g/L) aqueous solution of MCPM is coated with CDHA, possibly carbonate-containing CDHA [857]. Aqueous solutions of $(NH_4)_2HPO_4$ [746–748,860–866], $(NH_4)_3PO_4$ [865], $NH_4H_2PO_4$ [865], $K_2HPO_4$ [867,868], $KH_2PO_4$ [869] and others can be used instead of MCPM for similar purposes. Sometimes, a mixture of OCP and CDHA can be deposited on marble [865]. Interestingly, the well-known preparation process of coralline HA by hydrothermal conversion of the $CaCO_3$ skeleton of marine corals [870–872] appears to produce CDHA deposits of varying thickness on corals if stopped at intermediate stages. CDHA deposition on $CaSO_4 \cdot 2H_2O$ (gypsum) has also been produced by the same technique [873,874].

Techniques combining deposition and conversion are also known. That is, the surface of the substrate can be coated with $CaCO_3$ using any possible deposition technique and then the $CaCO_3$ deposit can be converted into $CaPO_4$ ones using any $PO_4$-containing conversion solution [747,748,875,876]. Namely, carbonate apatite/chitosan composites were deposited on Ti-6Al-4V alloy according to the following sequential steps: (i) deposition of $CaCO_3$ coating on the substrate by EPD; (ii) conversion to carbonate apatite in PBS; and (iii) modification of the coating with chitosan to form composites [875]. Similar results were obtained with ALD [747,748]. The same deposition-conversion approach is valid for $CaPO_4$: one can deposit one type of $CaPO_4$ (e.g., ACP, DCPD, OCP, etc.) and further convert it to another type of $CaPO_4$ (e.g., DCPA, CDHA, HA, etc.) by thermal treatments or soaking in alkaline or simulating solutions [63,148–152,163,421,429,494,557–560,785].

Mention should also be made of the possibility of the conversional formation of $CaPO_4$ deposits on the surface of another $CaPO_4$. For example, there have been studies in which α-TCP porous beads were prepared. Then, HA deposits were formed on their surface by hydraulic hydrolysis in ultrapure water at 120 °C in a heat-resistant pressure vessel. As a result, functionally graded porous beads consisting of α-TCP inside and HA on the surface were produced [877]. In addition, a slightly decalcified surface layer was created on porous HA bioceramics by a short soaking in phosphoric acid solution [878]. A similar approach was used to create surface conductivity for natural FA crystals to study their microtopography by scanning electron microscopy [879]. In both cases, thin layers of undisclosed acidic $CaPO_4$ were formed on the surfaces.

Moreover, for the same purpose $CaPO_4$ treatment by supercritical $CO_2$ in the presence of water might be used as well. Namely, biphasic (BCP = HA + β-TCP) $CaPO_4$ ceramics were immersed into deionized water and the mixture was inserted into a cell. The cell was filled at 5 °C with liquid $CO_2$ at the bottle pressure, before being heated to 37 °C. Upon this temperature rise, the pressure increased to ~80 bar. The system was maintained at 37 °C/80 bar for 4 h to allow for BCP surface dissolution/reprecipitation, then slowly depressurized for 30 min. After this treatment, the BCP/water mixture was extracted from the device and left to equilibrate for 1 h at room temperature, while still in contact with the medium. The samples were then retrieved, washed with deionized water and oven-dried 24 h at 50 °C. The surface of BCP samples was found to be composed of a mixture of OCP and carbonate-containing CDHA [880].

To finalize this section, it is worth noting that CDHA has been found to form spontaneously on ancient marble surfaces [881,882] and that $CaPO_4$ deposits formed by transformation have been proposed both to strengthen carbonate stone [861–863] and for the protection of outdoor marble artworks and artifacts [860,862,864].

**Part 2. Properties and Applications**

### 7. A Brief Description of the Most Important Properties

First, one should remember that the goal of this review is to inform readers of a large number of techniques and experimental approaches to produce $CaPO_4$ deposits (coatings, films and layers) on the surfaces of numerous substrates. Even cataloguing them with a brief description of their physical and chemical principles coupled with the key process parameters appears to be rather long. Second, although all techniques have been developed and/or adapted solely to deposit $CaPO_4$, the differences in physical and chemical principles commonly provide deposits, whose properties are difficult to compare (Figure 32) [883]. More to the point, application or non-application of various pre- and post-deposition treatments, as well as application of additional processing techniques such as a magnetic field [162,163], UV light [164] or ultrasonics [139,165,166] during deposition complicates the comparison even more. Therefore, let me limit myself just by brief descriptions of the most important properties of $CaPO_4$ deposits and their biomedical applications.

| Feature Coating method | Adhesion strength (MPa) | Porosity | Surface uniformity | Roughness (µm) | Temperature (°C) | Performance evaluation |
|---|---|---|---|---|---|---|
| Dip coating | 30 | Non | Mean | 1.00–1.50 | 600 | Positive |
| SIM | 15 | Low | Low | 2.00 | 450 | Negative |
| Plasma spraying | 38 | Adequate (3% to 20%) | High | 30.00 | 16000 | Positive |
| PVDMS | 80 | Low | High | 60.00 | 600 | Positive |
| MAO | 44 | Low | Mean | 40.00 | 200 | Positive |
| EPD | 12 | Low | Mean | 1.20 | 800 | Negative |
| CVD | 19 | Low | Low | 0.08 | 800 | Negative |
| IBAD | 45 | Low | High | 2.70 | 450 | Positive |
| Sol-gel dip coating | 25 | Non | Mean | 1.80 | 200 | Negative |
| PLD | 18 | High | Mean | 0.50–5.00 | 600 | Positive |
| EHD | 11 | Low | Mean | 0.10 | 20–40 | Negative |

**Figure 32.** Comparison of $CaPO_4$ deposits prepared by different methods. Abbreviations: CVD—chemical vapor deposition; EHD—electrohydrodynamic coating; EPD—electrophoretic deposition; IBAD—ion beam assisted deposition; MAO—micro-arc oxidation; PVDMS—physical vapor deposition magnetron sputtering; PLD—pulsed laser deposition; SIM—surface-induced mineralization. Reproduced with permission from [883].

#### 7.1. Introduction

Obviously, $CaPO_4$ deposition on substrates significantly changes their surface composition, structure and morphology, thereby improving the biofunctional properties of the surface such as bone bonding, corrosion and degradation performance. Therefore, such deposition is sometimes referred to as biofunctionalization [884,885] or (multi)functionalization [886]. However, the biomedical application of $CaPO_4$ deposits involves a number of important conditions. For example, non-absorbable or slow-release deposits with high crystallinity

have been recommended to maintain bond strength with the implant. Nevertheless, this contradicts the statement that the ideal interface between the implant and the surrounding tissue should coincide with the tissue being replaced. Namely, the crystallinity and bioactivity of HA are inversely proportional [887]. Therefore, in terms of bioactivity, $CaPO_4$ deposits should have low crystallinity and preferably contain numerous bone-mimicking dopants such as carbonate, Na and Mg. Therefore, deposits prepared from less crystalline and/or better absorbable $CaPO_4$ may be more beneficial for initial bone formation when compared with ones prepared from more crystalline HA, since a partial dissolution of the coating appears to be one of the first steps of binding [32]. Nevertheless, soluble $CaPO_4$ deposits will weaken the bond strength with the substrate. Namely, a comparative study was conducted on the biological stability and osteoconductivity of HA deposits on Ti produced by plasma spraying and PLD. After 24 weeks of implantation, plasma-sprayed HA showed significant instability and reduced thickness, but were not statistically different from the uncoated Ti control. In contrast, those of PLD remained largely intact but showed significantly higher bone attachment [888]. Therefore, the stability of $CaPO_4$ deposits outweighed their solubility. Furthermore, excessive amounts of ions leaching from soluble deposits can cause localized inflammatory reactions. Therefore, ideal deposits should be complex in both composition and structure. That is, the rapidly resorbing ACP phase should be located primarily on the surface where it can support the growth of bone-like apatite, while crystalline, slow dissoluble HA should be localized towards the implant/deposit interface to ensure long-term survival and provide adequate bond strength. However, some crystalline phase needs to be incorporated into the ACP phase so that when the other phase components dissolve, bone-like apatite becomes the base.

Additionally to inducing quicker bone regeneration and better bonding between the implants and the newly formed bones, the $CaPO_4$ deposits should adhere well without releasing particles that could damage other components of the implant. Furthermore, to reduce the intrinsic residual stresses at implant placement, ideally, its elastic modulus should have a value between those for substrate and bone. Besides the processing conditions and preparation techniques, which directly affect the structure, crystallinity, morphology and chemical composition, there are several factors that influence the mechanical, chemical and physical properties of $CaPO_4$ deposits. These include thickness (which affects adhesion and fixation, and currently accepted optimum values appear to be between 50–100 μm), fatigue resistance, adhesion, chemical and phase purity, and porosity [8,221]. Wear resistance may also be important [200]. Standard specifications for $CaPO_4$ deposits for implant materials are available [889].

### 7.2. Elastic Modulus and Hardness

Hardness is a property related to yield strength and is useful for predicting the strength of deposits, the stress state within the deposit and the wear resistance of the deposit surface. To minimize the stress shield effect (resulting from the mismatch between bone stiffness and implant stiffness), the mechanical properties of the $CaPO_4$ deposits should ideally be between bone and substrate. Therefore, a number of studies have been conducted on the mechanical performance of $CaPO_4$ deposits [33,58,174,211–215,228,252,285,477,890–896] but the values obtained seem to vary widely. This is due to several important reasons, such as the anisotropic behavior of $CaPO_4$ deposits, different properties and types of substrates, different techniques used for precipitation, various precipitation conditions, the use or non-use of additives (binders, surfactants, porous materials, etc.), and the application or non-application of pre- and post-deposition treatments. Due to those variables, the mechanical properties of $CaPO_4$ deposits prepared by different researchers under different conditions using different techniques (Table 3) seem to be largely incomparable. Furthermore, the measurement technique itself may also have an influence. For example, dynamic nanoindentation with a Vickers tester was used to measure elastic modulus and hardness of magnetron-sputtered $CaPO_4$. The immersion rate of the Vickers diamond pyramid with an angle of 136° between the opposing surfaces was 5 mN/min and the duration of the

loading/unloading cycle was 2 min. The numerical values of nanohardness and Young's modulus were ~10 GPa and ~110 GPa, respectively. The authors also noticed that nanohardness decreased with increasing indentation penetration depth, a phenomenon known as the indentation size effect [276]. In other studies, nanohardness values of 3.4–4 GPa and Young's modulus values of 122–150 GPa were measured [272,891]. Comparable numerical values were found for microplasma sprayed HA [211–215]. Correspondingly, the numerical values of Young's modulus, and hardness for crystalline and amorphous $CaPO_4$ deposits produced by PLD on both Ti and Si substrates were ~93 GPa (on Ti), ~127 GPa (on Si), ~1.6 GPa (Ti) and ~2.3 GPa (Si), while the values for amorphous coatings were 74–107 GPa (Ti), ~68.5 GPa (Si), 0.55–1.06 GPa (Ti) and ~0.40 GPa (Si) [890]. Nanoindentation tests have shown that the Young's modulus of IBAD coatings is higher than $91.7 \pm 3.6$ GPa and the microhardness is higher than $5.27 \pm 0.32$ GPa [268]. Moreover, thick plasma-sputtered HA deposits showed higher microhardness values on the top surface compared to those in the center of the cross-section [893]. However, plasma-sputtered HA coatings on tungsten were found to have Young's modulus values below 6 GPa in both tension and compression, as assessed by cantilever bending tests [31]. It has been found that the hardness of $CaPO_4$ deposits depends on the substrate temperature during deposition and the correlation between the two parameters is complex. For example, in one study, the hardness of $CaPO_4$ deposits increased with increasing substrate temperature, reaching values as low as 5 GPa at 30 °C and as high as 28 GPa at 700 °C [296], while in another study both the hardness and modulus of $CaPO_4$ deposits decreased with increasing substrate temperature [282]. This is probably due to the different deposition techniques used in these studies.

The numerical data on hardness and Young's modulus for $CaPO_4$ deposited by the wet method appear to be low. Namely, the values for HA deposited by sol–gel method are ~0.25 and ~28 GPa, respectively [477]. In contrast, Young's modulus values for $CaPO_4$ deposited by the biomimetic method were found to be ~4.5 GPa [485]. More details on mechanical testing methods for $CaPO_4$ deposits can be found in the literature [897].

### 7.3. Fatigue Properties

Several studies have shown that cyclic loading of $CaPO_4$-coated samples leads to fatigue failure of $CaPO_4$ deposits [102,103,549]. Furthermore, the combination of aqueous environment and stress can cause accelerated exfoliation or dissolution of deposited $CaPO_4$, affecting the long-term stability of the implants [186,898–905]. For instance, different types of $CaPO_4$-coated substrates were mechanically tested under either dry or wet (SBF [900–904], HBSS [905], Ringer's [186] solution) conditions. The results indicate that the fatigue properties of amorphous (ACP) and crystalline $CaPO_4$ deposits appear to be different. Furthermore, fatigue behavior revealed substantial differences when tested under either dry or wet conditions [900–905]. Nevertheless, some studies have not detected fatigue effects [906,907]. Namely, repeated fatigue testing of HA-deposited samples in lactate Ringer's solution for up to 5 million cycles showed no change in response to fatigue loading [906]. Other researchers found that, after up to 10 million cycles of bending in air and SBF, there were no significant microcracks or delamination of the plasma-sprayed HA deposits, nor significant changes in crystallinity, weight, residual stress or thickness [907]. All these large differences in fatigue properties can be attributed to a number of processing variables, including the nature of the substrate material, the deposition technique used, and the application or non-application of pre- and post-deposition treatments.

Surface modification of substrates is commonly used to improve the fatigue stability of $CaPO_4$ deposition. For example, we have found that HA/Ti bond coating on the surface of Ti substrates prior to $CaPO_4$ deposition extends fatigue life [102,103].

### 7.4. Thickness

As clearly shown in Table 3, the thickness of the $CaPO_4$ deposits varies from nanometer size to several millimeters depending on the manufacturing technique and this parameter seems to be very important. That is, if the $CaPO_4$ coatings are too thick, they are more

fragile. Moreover, the outer layer may easily detach from the inner layer, eventually compromising the mobility of the implant. Conversely, the resorbability of HA (Table 1), the second least soluble $CaPO_4$, is approximately 15–30 μm per year under physiological conditions so, if too thin, the deposits are more likely to dissolve [908]. Obviously, other $CaPO_4$, except FA, dissolve faster. Therefore, the optimal thickness of the $CaPO_4$ deposits is a reasonable compromise between the mechanical properties and dissolution. Furthermore, the choice of thickness with respect to absorption also depends on the phase and chemical composition, the atomic structure (crystalline or amorphous) and dispersion degree of the grain structure. To complicate matters further, the failure mechanism seems to be different for thin and thick films. For example, the failure mode of thinner (~50 μm) HA deposits on Ti alloys was conclusively found to be at or near the coating/bone interface, while that of thicker (~200 μm) HA deposits was found to be at the coating/bone interface, inside the HA lamella splat layer, and deposit/substrate interface [909,910]. Similar conclusions were reached in another study comparing the mechanical behavior of 0.1, 1, and 4 μm thick $CaPO_4$ deposits [911]. Furthermore, the thickness of $CaPO_4$ deposits was found to affect both mechanical fatigue behavior [912] and implant fixation in vivo [913]. Nevertheless, there were no statistically significant differences in the bone formation degree related to the thickness of HA deposits [914]. Furthermore, the corrosion resistance of metal substrates was found to depend on both grain size and thickness of the $CaPO_4$ deposits they protect [471,532]. Considering these points, commercially available plasma-sprayed HA coatings are typically 50–200 μm thick [221], but since cells and tissues interact only with the top surface, a thickness of ~10 nm is sufficient for cell activity.

The following examples can be mentioned to demonstrate the complex correlation between the thickness of $CaPO_4$ deposits, and their structure and chemical composition. Various thicknesses of $CaPO_4$ deposits were deposited by PLD, ranging from 170 nm to 1.5 μm [287] and from 40 nm to 5 μm [288], depending on the deposition time. Regarding the results of the first publication, the morphology was found to be granular particles and droplets. During growth, the granular particles grew larger and partially covered the droplets, forming a columnar structure. The thinnest deposits (~170 nm) consisted mainly of ACP, those 350 nm thick also contained HA and the thicker ones in addition to HA also contained α-TCP. All types of deposits failed in the scratch test due to spalling from the diamond tip but the failure load increased with decreasing thickness, and only plastic deformation and cohesive failure were observed for the thinnest coatings [287]. Comparable data were obtained in the second study [288]. Thus, both the phase composition and the structure of $CaPO_4$ deposits may depend on their thickness.

*7.5. Adhesion and Cohesion*

The performance and durability of an adhesive on a substrate is largely supported by two fundamental properties: adhesion and cohesion. Adhesion is considered bond strength, meaning the ability of an adhesive or coating to adhere to a surface and then bond the two surfaces together, and cohesion represents the internal strength of the material, which is the attraction of particles or other materials within the adhesive to hold the adhesive together [915].

In surgical procedures, when delamination of the coating occurs, both implant failure and undesirable tissue reactions occur. This leads to implant micromotion, increased fretting and debris particle formation [916]. A typical morphology of a fracture surface after $CaPO_4$ delamination is shown in Figure 33 [808]. Thus, both adhesive and cohesive fractures were present. Thus, $CaPO_4$ deposits of all types, regardless of their intended function, must be both strong enough to maintain cohesion and satisfactorily adhere to the underlying substrate. Since neither the substrate nor the $CaPO_4$ deposits is perfectly flat, the bottom of the deposits will never be in complete contact with a substrate. The area of contact is called the "welding points" or "active zones". There are voids of various dimensions and shapes between them. Commonly, the larger the contact area, the better the adhesion [917]. Since chemical bonding between the deposited $CaPO_4$ and the substrate is

rare, mechanical fixation is the main mechanism involved in adhesion. Thus, the size and shape of the grains in close contact with the substrate affect the adhesion of HA deposits, with hexagonal grains of increasing size (up to 250 nm side length) deteriorating adhesion and granular nano-sized grains improving interlocking by the substrate and increasing deposit adhesion [250]. Furthermore, adhesion strength has been found to be a linear function of average surface roughness [918] and, thus, when compared to smooth substrate surfaces, highly roughened substrate surfaces exhibit higher adhesion strength [372,919]. The schematic in Figure 34 [187] illustrates three possible cases for the interaction of plasma-sprayed $CaPO_4$ splats with the first few layers of substrate asperities, depending on the splat dimension/substrate roughness (SD/SR) ratio: for SD/SR >> 1 (case A) each splat covers several substrate asperities at once, improving interlocking, which may be advantageous for bond strength. However, some unfilled gaps remain; the SD/SR ~ 1 case (Case-B) appears to be better because the deposition of small splats can fill the gaps and improve interlocking. Finally, in the case of SD/SR << 1 (Case-C), the deposition of small spherical particles does not significantly fill the gaps between the particles, resulting in poor adhesion and weak cohesion near the substrate coating interface [187].

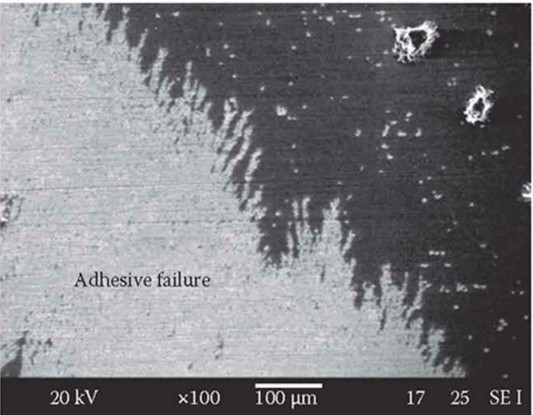 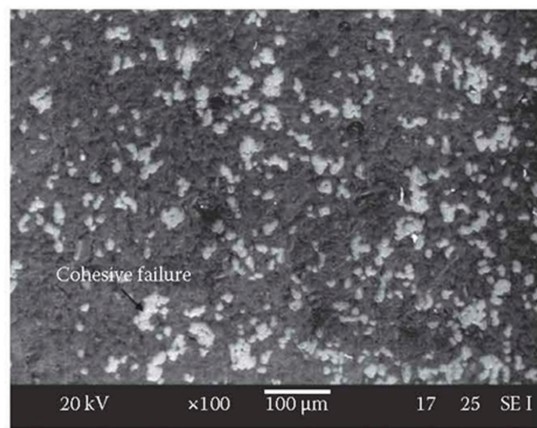

**Figure 33.** A typical morphology of the fracture surface of a $CaPO_4$ deposit. The image on the left shows a typical adhesive fracture where part of the coating has detached from the substrate. In the image on the right, a cohesive failure is observed. This occurs within the coating and is a mottled delamination on the surface of the remaining coating. Reproduced with permission from [808].

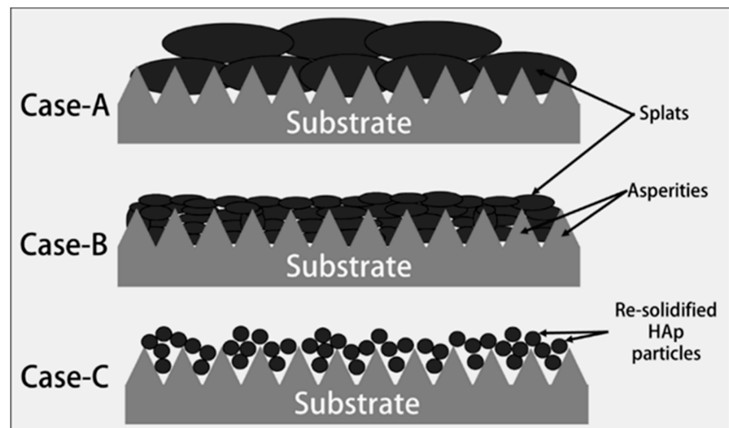

**Figure 34.** Three different possible deposition states on substrate asperities depending on the size and shape of the fully fused HA (denoted as HAp) particles affecting their adhesion to the substrate. Reproduced with permission from [187].

Therefore, surface preparation methods such as sandblasting and abrasion are used to increase roughness and improve adhesion strength before spraying (Section 3. Brief

knowledge on the important pre- and post-deposition treatments). In contrast, if a large amount of shrinkage occurs during particle solidification, the amount of mechanical fixation is reduced [182]. As the strength of human bone is about 18 MPa, all types of $CaPO_4$ deposits adhering to the implant surface should have either higher or, at least, comparable bond strength values. Therefore, according to ISO guidelines, the adhesion strength of $CaPO_4$ deposits should be equal to or exceed 15 MPa [19–23].

In particular, the adhesion of the $CaPO_4$ deposits must be enough to maintain its biological activity after implantation. In general, tensile adhesion tests according to ASTM C633 [920] and ASTM F-1147-05 [921] standards are the most frequent procedures to determine quantitative adhesion values. An illustrative sketch of the procedure is shown in Figure 35. In addition, fatigue [181,922], scratch [890,892,923–928] and tensile [923] tests as well as an abrasion resistance test [925] are among the most valuable methods providing supplementary information on the mechanical properties of $CaPO_4$ deposits. Moreover, there are scratch tests, which are performed with reference to ISO 20502:2005 [929]. Changes in surface topography can give an indication of abrasion resistance. Namely, $CaPO_4$ deposits with higher adhesion to the substrate show less variation in surface roughness and studies on different parameters have shown that deposition time is the most influential factor on wear behavior [925]. The latter parameter is correlated with the thickness of the coating. The load at which complete removal of the coating occurs is often taken as an indicator of adhesion strength.

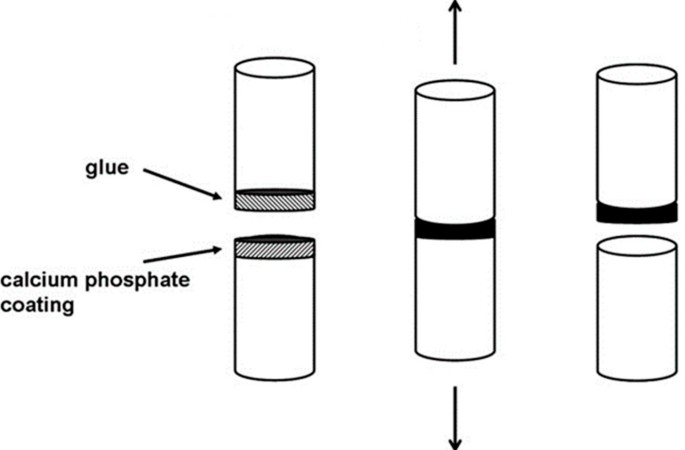

**Figure 35.** A schematic diagram of a standardized tensile adhesion measurement.

Like other properties, the adhesion strength of $CaPO_4$ deposits depends on many parameters [915]. First, it is highly dependent on the deposition technique (Figure 36) [930]. For example, PLD-deposited HA showed higher adhesion strength to titanium alloys compared to plasma sprayed HA [931]. Furthermore, the adhesion strength may depend on the thickness and chemical composition of the deposit. Namely, 50 μm thick deposits showed higher adhesion strength than 240 μm thick deposits [932]. On the other hand, scratch tests showed that the adhesion of sol–gel fluorinated HA (FHA) to titanium alloy substrates improved up to 35% with increasing fluorine concentration [468]. In another study on FHA deposits containing different amounts of fluorine, it was found that the adhesion strength increased up to 40% and fracture toughness increased up to 200–300% with increasing degrees of fluorination [470]. In addition, the structure, chemical composition and nature of the substrate surface also play an important role [36]. It was also found that the adhesion strength of plasma-bent deposits decreased with decreasing plate power (from 28 kW to 22 kW) or increasing working distance (from 90 mm to 130 mm) [224]. Furthermore, the adhesion strength of HA deposits was found to increase with increasing magnetic field strength [163]. The adhesion strength of $CaPO_4$ deposits on Ti plates pretreated with alkaline solution and then heat treated in air (600 °C, 1 h) was higher (~35 MPa) compared

to those heat treated in a vacuum (~21 MPa). That can be attributed to differences in the structure and composition of the interfacial layer of sodium titanate [933]. Moreover, a correlation between adhesion and residual stress was found [227]. For the plasma-assisted deposition technique, a good overview of the adhesion strength values for $CaPO_4$ is given in Table 3 of reference [181].

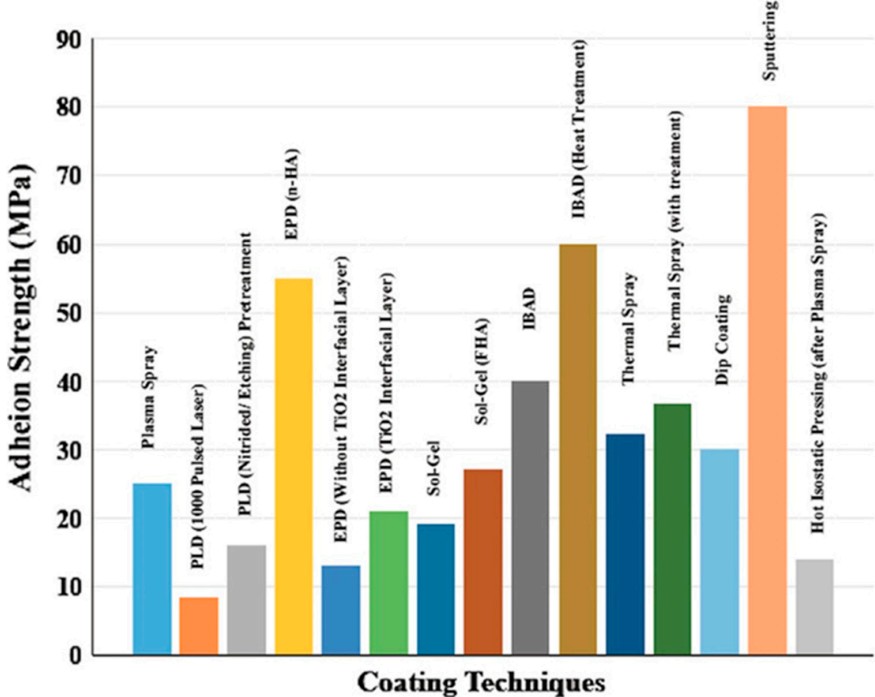

**Figure 36.** The bond strength values of HA deposits on Ti-6Al-4V alloy specimens by various techniques. Reproduced with permission from [930].

Nevertheless, the application of various intermediate coatings (synonym: buffer layers) is the most important method to influence the adhesion strength of $CaPO_4$ deposits [915]. The great diversity of available deposition techniques described in Section 4, multiplied by the large selection of different substrates, results in a large number of potentially suitable chemistries used as an intermediate layer between the substrate and the deposited $CaPO_4$. For example, chemicals such as $TiO_2$ [934], TiN [286,935–937], $ZrO_2$ [286] or $Al_2O_3$ [286] have been used as buffer layers in plasma-assisted deposition. $TiO_2$ [938,939] and TiN [272] have also been used as base layers for RF magnetron sputtering. Similarly, the formation of interlayers of hydrated titanium dioxide on Ti [365,372,477,940] and polar hydrophilic groups on polymers [54,56,63,73–76] is essential to improve the adhesion of wet-deposited $CaPO_4$. Namely, pretreatment of polyethylene terephthalate, polymethyl methacrylate and polyamide 6 polymer surfaces with NaOH increased the adhesion strength of the deposited apatite from 3.5 to 8.6 MPa, 1.1 to 3.4 MPa and 0.6 to 5.3 MPa, respectively [54]. Favorable effects were also found with glow discharge [73–75] and UV [49,75,76] pretreatment of the same polymers. Additional types of pretreatments of various surfaces have been found to have a similar effect [36]. To complicate things further, it should be noted that interdiffusion of atoms, ions and molecules of the deposited $CaPO_4$ on the one hand and atoms, ions and molecules of the deposited substrate on the other hand can occur. This is particularly useful for high-temperature technology but mutual diffusion is not a good solution for post-deposition annealing. As a result, various types of non-stoichiometric interlayers are formed. Namely, the width of such an interlayer between the HA coating and Mg substrate formed by IBAD technology was found to be ~3 μm [941]. This interlayer can reduce the mismatch in thermal expansion coefficients between the coating and the substrate, affect surface area, wettability and thermal conductivity, and thus improve adhesion strength

without affecting biocompatibility. As the topic of interlayers is very broad, no further details are provided.

Finally, a few more examples should be mentioned. In dense $CaPO_4$ deposits under tensile loading, the cohesive strength was higher than the bond strength, so that fracture occurred at the interface between the substrate and the deposit. For porous $CaPO_4$ deposits, the cohesive strength is low and fracture occurs in them [150]. Amorphous $CaPO_4$ deposits (ACP) appeared to have more brittle properties and less adhesion than their crystalline counterparts [285]. The adhesion strength between apatite layers and Ti metal surfaces has been reported to be 10–30 MPa [942,943]. Comparable data were obtained in another study where $CaPO_4$ was deposited on Ti substrates by a biomimetic method from two types of SBFs. The data showed that both a surface roughness of the substrate and an ionic concentration of SBF have a significant effect on the morphology, formation and binding strength of $CaPO_4$ deposits. The maximum value of bond strength for the deposited coatings was ~15.5 MPa [944].

At the end of this section, it should be noted that, besides adhesive failures, cohesive ones could also happen with $CaPO_4$ deposits [46]. More information on this topic is available in the literature [915].

### 7.6. Surface Characteristics: Crystallinity, Morphology and Roughness

Although all these parameters influence the behavior of $CaPO_4$ deposits after implantation, the surface characteristics of the coated implants appear to be indistinguishable from those of bulk $CaPO_4$ bioceramics with the same composition, crystallinity, morphology and roughness [945]. Namely, the crystallinity affects their solubility in the physiological environments: the higher the crystallinity, the more stable $CaPO_4$ deposits are in the solution [946]. The degree of crystallinity can be controlled by various types of post-deposition treatments (Section 3. Brief knowledge on the important pre- and post-deposition treatments for details). The international standard ISO 13779-2:2018 recommends a crystallinity of 45% or higher for thermally sprayed HA coatings for bone implants [21], while the detailed description of determining the degree of crystallinity and the amount of secondary phase in $CaPO_4$ deposits are given in the international standard ISO 13779-3 [947].

Regarding the surface morphology of $CaPO_4$, it affects bone cell attachment, growth, proliferation and differentiation [948,949]. By the functions of deposition techniques, experimental conditions and application/non-application of post-deposition treatments, the surface morphology of the $CaPO_4$ deposits can be modified. Normally, smooth surfaces are more effective than sharp morphologies for osteocyte adhesion [950].

Finally, roughness higher than 2 μm is not desirable because the long distance between valleys and peaks prevents the formation of osteoblast pseudopodia, which is necessary for osteocyte adhesion [951].

### 7.7. Biodegradation

Biodegradation (synonyms: biological degradation, abiotic degradation) is the chemical dissolution of materials by cells, bacteria or other biological means. As the chemical composition of all body fluids is considered to be constant, the biodegradation of any type of deposit is only controlled by the properties of $CaPO_4$, including purity, Ca/P ratio, crystal structure, chemical and phase composition, crystallinity, porosity, lattice defects and particle size [8,952]. Thus, until the $CaPO_4$ deposit becomes fine enough for body fluids to access the substrate surface, its biodegradation is likely to be similar to $CaPO_4$ bulk bioceramics [953,954] with the same structure, composition and properties (porosity, roughness, topography, etc.).

Generally, the biodegradability requirements of $CaPO_4$ deposits seem to be controversial. That is, which quality is optimal? Should it remain permanently on the implant surface or should it be a temporary property? There are two possible directions in this regard. One is to create a stable deposit to increase its bonding strength with the implant and the other is to create a resorbable deposit to increase its bioactivity.

Thus, studies on the behavior of CaPO$_4$ deposits in various solutions have been continued. Namely, the dissolution rate of deposited HA was studied using a binary constant composition method in which the dissolution rate decreases with increasing crystallinity of HA [955]. Comparable data have been obtained in vivo. After implantation, HA deposits with ~55% crystallinity degraded faster than those with ~98% crystallinity and were found to have superior osteoinductive properties [946]. Although various in vitro simulations have been widely studied, biodegradation is thought to be an in vivo process. However, to more closely approximate in vivo conditions, SBF [191,192,325,939,956–961], HBSS [275,937,962], saline [963,964], Ringer's solution [965,966], PBS [275,967,968], Eagle's minimum essential medium (EMEM) [969] and various other simulation solutions are used to assess the biodegradability of CaPO$_4$ deposits. Medium containing osteoclast-like cells has also been used [970]. The simulated solutions often contain dissolved calcium and orthophosphate ions, resulting in both partial dissolution and recrystallization of CaPO$_4$ deposits [192,957–959,969]. This is evidenced by the following quote: "The soaking in SBF homogenizes the morphology of coatings. The sintered zone disappears and the pores get filled by the reprecipitated calcium phosphates" [192]. It should be emphasized that ion-substituted CaPO$_4$ crystallizes in all cases due to the presence of other ions in the chemical composition of the simulated solution.

Commonly, the biodegradation rate of deposited CaPO$_4$ is proportional to the solubility values of the individual CaPO$_4$ compounds listed in Table 1. Namely, both bone union and osteogenesis of plasma-sprayed α-TCP, HA and TTCP were evaluated by histological observations and mechanical extrusion tests after 3, 5, 15 and 28 months of implantation. Of these, the most soluble phase α-TCP showed the most significant degradation ~3 months after implantation, while HA and TTCP showed the first significant degradation signs ~5 months after implantation [971]. The authors of another study got comparable results [961]. The results showed that implants coated with α-TCP had lower mechanical push-out test values compared to those coated with HA or TTCP [972]. Plasma-sprayed HA deposits appeared to dissolve faster than the stoichiometric HA. This is due to the dissolution of HA powder at elevated temperatures and its partial decomposition into more soluble compounds such as hot ACP, TTCP and OA [973]. Likewise, magnetron-sputtered CaPO$_4$ deposits were found to be almost amorphous (ACP) and therefore completely dissolved after only 24 h exposure to PBS, while the dissolution rate of the same deposits after annealing (crystallization) was more restricted [967]. Furthermore, HA deposits were found to be less stable than those of FA and similar in stability to Mg-whitlockite (i.e., Mg-substituted β-TCP) [974–976].

On the other hand, there are also cases where the biodegradation rate of CaPO$_4$ deposits is not fully correlated with the solubility values of the individual compounds listed in Table 1 [908,977–980]. Namely, three types of CaPO$_4$ (ACP, β-TCP and HA) deposited by laser ablation techniques were immersed in SBF to investigate their behavior. The results showed that HA and ACP were insoluble, while β-TCP was slightly solubilized. Precipitation of the apatite phase was found to favor both HA and β-TCP, but not ACP [979]. Furthermore, the degradation rate of dental implants containing HA, FA and fluorohydroxyapatite (FHA) bonded at thicknesses of 50 and 100 microns was studied [908]. Implants were placed in the jaws of dogs and retrieved for histological analysis after 3, 6 and 12 months. The results revealed that HA and FA (even at 100 micron thickness) were mostly degraded during the implantation period, while FHA deposits showed no significant degradation over the same period [908]. FHA with 25% fluorine substitution appeared to degrade less in SBF than HA and FHA with 60% fluorine substitution [980].

A model for the development of CaPO$_4$ splats immersed in SBF and the influence of their structure on formation of bone-like deposits is presented in Figure 37 [956]. The molten part of the HA splat shows a higher dissolution rate compared to HA due to the complex mixture of different phases (α-TCP, β-TCP, high-temperature ACP, TTCP, OA, calcium pyrophosphate, calcium metaphosphate and CaO), while the unmelted splat centers are composed of the original raw powder of the crystalline HA, which has a slower

dissolution rate. Thus, the molten portion of the HA splat rapidly disappeared without forming bone-like apatite and, in contrast, the remaining core provided a surface for the biomimetic precipitation of ion-substituted CDHA. Thus, to provide a basis for bone-like apatite precipitation, the ideal $CaPO_4$ deposits should be a mixture of rapidly dissolving phases and crystalline structures, e.g., those evolving from the unmelted core [956]. A comprehensive scheme describing the various possibilities that occur in aqueous solutions containing plasma-sputtered $CaPO_4$ deposits is shown in Figure 38 [225].

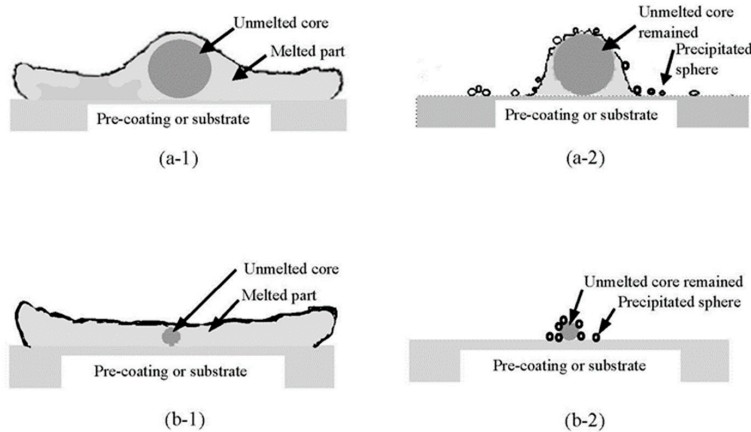

**Figure 37.** Schematic diagram showing the dissolution–precipitation behavior of individual HVOF sputtering HA splashes. It can be seen that the surrounding molten and subsequently solidified areas are almost completely dissolved, leaving only the unmelted HA core; (**-1**) belongs to the original splat and (**-2**) is the corresponding aged morphology. Compared to (**a**), in (**b**) the decrease in HA particle size leads to more conformability. Reproduced with permission from [956].

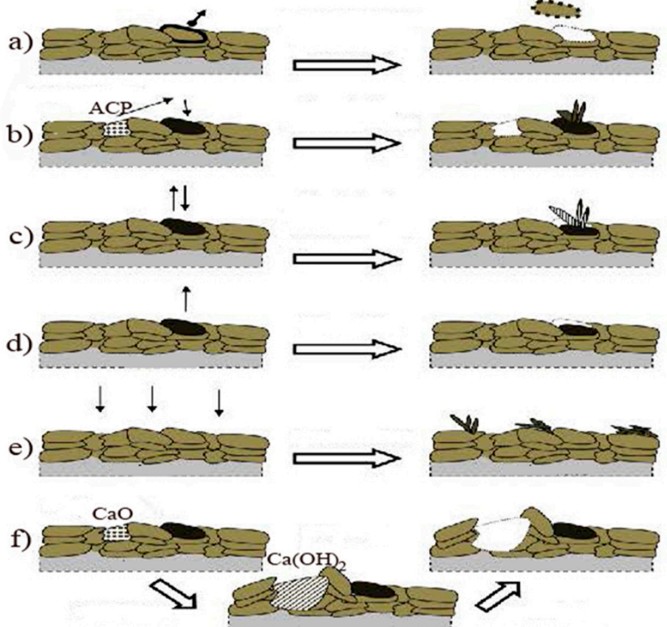

**Figure 38.** A schematic diagram of the processes occurring in plasma sprayed $CaPO_4$ deposits immersed in aqueous solutions: (**a**) disintegration (intergranular dissolution), (**b**) dissolution of the soluble phases (e.g., ACP) and precipitation of the less soluble phases (e.g., CDHA), (**c**) hydrolysis of the metastable phase (e.g., OA), (**d**) partial dissolution, (**e**) precipitation from supersaturated solutions (e.g., SBF), (**f**) hydrolysis of CaO and subsequent dissolution of $Ca(OH)^2$. Adapted with permission from [225].

### 7.8. Interaction with Cells and Tissue Responses

The interaction of $CaPO_4$ deposits with both cells in vitro and surrounding tissues in vivo has been widely studied [36,137,160,281,303,354–356,438,612,629,639,981–993]. As before, until $CaPO_4$ deposits become thin enough for cells and tissues to have direct access to the substrate surface, any interactions of cells are similar to those with $CaPO_4$ bulk bioceramics of the same structure, composition and surface properties [953,954]. In other words, in vitro studies with different cell lines have shown that in most cases $CaPO_4$ deposits promote cell adhesion, proliferation and differentiation, and in vivo studies have shown that they promote bone regeneration. Namely, the combination of surface topography and $CaPO_4$ deposits was found to be beneficial for implant–bone reactions during the healing phase [994]. $CaPO_4$ deposits immobilized with sodium bisphosphonate were found to be effective in promoting bone formation on the surface of dental implants [160]. Ti implants coated with $CaPO_4$ had a higher bone contact length at 3 and 12 weeks after implant placement compared to uncoated controls [995]. Similar results were obtained in other studies [996–998]. It is clear that the wettability of $CaPO_4$-coated substrates promotes adhesion and proliferation of various cell types compared to uncoated controls (Figure 39) [394]. Furthermore, the electrical polarization of the $CaPO_4$ deposits further enhanced the biomedical properties of the coated substrates [10,141,202].

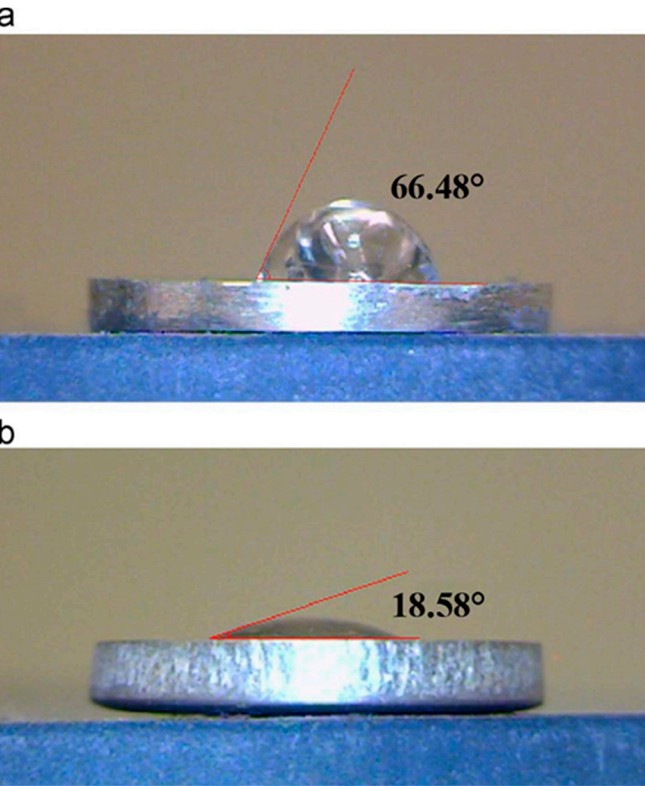

**Figure 39.** Water contact angle measurements of (**a**) uncoated and (**b**) HA-coated Ti samples. Reproduced with permission from [394].

In cell-based experiments, adhesion, diffusion, extracellular matrix formation and focal adhesion plaque formation of human gingival fibroblasts were investigated on commercially available Ti, HA-coated Ti and porous TCP/HA-coated Ti. On the surfaces of TCP/HA and HA deposits, the number of adherent cells and cell proliferation area were higher in TCP/HA deposits than in pure Ti and focal adhesion plaques formed faster than on uncoated substrates: TCP/HA deposits had higher adherent cell number and type I collagen formation than HA deposits [999]. OCP [291], CDHA [970,1000,1001] and HA [47,291,1002] deposits successfully proliferated osteoblasts and osteoblast-like cells on



their surfaces, indicating that all types of deposits promoted cell proliferation, metabolic activation and differentiation. Moreover, in vitro studies with MG63 osteoblast-like cells were performed on $CaPO_4$ deposited by sol–gel, plasma spray and sputtering methods. The studies demonstrated an ability of the cells to proliferate on the tested materials. Among others, sol–gel coatings were found to promote better cell growth, greater alkaline phosphatase activity and higher osteocalcin production when compared with the sputtered and plasma spray deposits [303]. In one more study, $CaPO_4$ deposition significantly induced higher levels of cell differentiation when compared with uncoated controls [1003].

Furthermore, the surface topography of the coated substrate was found to influence the nature and scale of the reaction. That is, $CaPO_4$ was deposited on substrates with different topographies. A layer of fibronectin from the solution was then deposited on each surface and the response of MG63 osteoblast-like cells was investigated. The cells on fibronectin-coated $CaPO_4$ with regular topography in the nanometer range were found to show statistically significant differences in osteocalcin expression, alkaline phosphatase activity and focal adhesion assembly when compared with $CaPO_4$ deposits without these topographies [988]. Thus, both the adsorbed bioorganic material and the surface topography of the substrate are able to influence the cell adhesion and differentiation.

Regarding in vivo studies, the osseointegration rate of Ti-6Al-4V implants with porous surfaces was compared between control (unmodified sintered coating) and implants with porous surfaces modified by deposition of $CaPO_4$ in the form of inorganic or organic root sol–gel processes. The grafts were implanted in the rabbit condyle of the distal femur and, after a 9-day healing period, the fixation strength of the implants was evaluated by a tensile test. Both types of $CaPO_4$ deposition significantly increased the initial bone growth rate and fixation, as evidenced by higher tensile strength and interface stiffness compared to the control [1004].

To finalize this section, it should be noted that, unfortunately, the positive clinical benefits of $CaPO_4$ deposits have not always been established [1005]. Namely, studies using both in vitro and in vivo tests have been conducted to evaluate the processes involved in the biological response of HA-coated Ti-6Al-7Nb alloys. The results of coated samples were found to be similar to those obtained with uncoated samples [1006–1010]. Similarly, there are studies in which, over periods of 8 years [1011], 8–12 years [1012,1013], 10 years [1014], 13 years [1015,1016], 15 years [1017], 15–16 years [1018], 15–18 years [1019], 22–27 years [1020] and undated [1021,1022], no influence of HA deposition was detected. For instance, the following conclusions were reached in 2017: "HA-coated cups have a similar risk of aseptic loosening as uncoated cups, thus the use of HA coating seems to not confer any added value in terms of implant stability. The risk of infection seemed higher in THA (total hip arthroplasty) with use of HA-coated cups, an observation that must be investigated further." ([1016], Abstract). That conclusion was confirmed in both 2021: "With the limitations of the present review, Ca-P-coated Ti surfaces have similar osseointegration performance to conventional etched surfaces." ([1023], Abstract) and 2022: "HA coating did not improve THA implant survival in our veteran population. Although HA-coated versions of hip implants tend to be more costly than their noncoated counterparts, these results do not support their general use." ([1013], Abstract); "In large animals, there does not seem to be much effect of TCP-coated or HA-coated implants over uncoated rough titanium implants in the short term." ([1010], Abstract).

Furthermore, one study found that implants coated with $CaPO_4$ were advantageous in the short term (4 weeks) but after 6 months there was no significant difference from uncoated specimens [1024]. Moreover, BoneMaster® (BIOMET Corp., Warsaw, IN, USA) surfaces had significantly higher osteocalcin production and alkaline phosphatase activity when compared to uncoated controls. Nevertheless, no differences were found between the uncoated and gold-coated BoneMaster® samples, indicating that the topography of BoneMaster® is the main factor and not the nature of $CaPO_4$ [1025]. Furthermore, cases of inflammatory tissue reactions have been identified [1026,1027]. More importantly, adverse events associated with these deposits have been reported [1017,1028–1030], which can lead

to fragmentation, migration and even increased polyethylene wear due to the third body abrasive wear. It is worth mentioning that a short-term inflammatory response to HA deposits on Ti was lower when compared with DCPD deposits; the differences observed between Ti-HA and Ti-DCPD implants were attributed to their dissolution properties: HA deposits had higher stability and thus reduced the inflammatory response [1025]. Also, HA deposits could be a risk factor for cup revision due to aseptic loosening [1031]. There- fore, measures to prevent contamination (aseptic technique) and infection (perioperative antibiotics) seem to be more important for $CaPO_4$-coated implants when compared with the uncoated controls [1032].

## 8. Biomedical Applications of $CaPO_4$ Deposits

The first patent for development of thermal sprayed HA deposits on metal implants was issued in 1979 [170]. The results of the first clinical trials were published in 1987 [1033]. Shortly thereafter two leading surgeons in the field of orthopedic surgery, Furlong and Osborn, began implanting plasma-spray-deposited HA stems in patients [1034]. Other clinicians followed their lead [1035,1036]. Since then, many scientific publications have been reported on the benefits of $CaPO_4$-coated implants. Summarizing the available in- formation on the biomechanical and biomedical properties of $CaPO_4$-deposited implants, the following data can be claimed. Compared to uncoated controls, deposited $CaPO_4$ im- proved bone-implant contact [639,964,965,1037–1043], initial stability [1044], implant fixa- tion [910,1045–1049] and nanomechanical properties of adjacent bone [1050], higher torque values [1038,1039,1049,1051] and extrusion strength [1052], protecting the interface from wear particles [1053], closing small gaps [1054,1055], reducing ion emissions from metal sub- strate [963,1056–1058], retarding metal degradation and corrosion [38,43,78,79,465,941,1059–1061], bone growth [1062–1064],remodeling [1065,1066], osteointegration [35,486,914,1067–1070], im- proving biocompatibility [1071], osteoconductivity [465,639,782,997,1004,1072–1075], os- teoinductivity [1076], bone immunomodulation [1077], osteogenesis [160,1042,1049,1078,1079], early bone [486,1048,1079–1081] and healing [1082] responses, prevention of fibrous tissue formation (Figure 40) [180,1083], ectopic bone formation [507], osteoblast density [1084] and osteoblast proliferation [759], and improvement of the clinical performance of or- thopedic hips. Furthermore, the antimicrobial properties of deposited $CaPO_4$ have been detected in several studies [38,1084]. Remarkably, to improve osteoinductive properties, biphasic formulations HA + β-TCP were coated with nanosized HA [1085,1086]. It should be emphasized that all those cases represent a range of positive effects of $CaPO_4$ deposi- tion by different techniques but comparative studies have revealed that these effects are highly dependent on the deposition technique. That is, compared to uncoated controls, electrochemically deposited $CaPO_4$ was found to contribute to bone-implant fixation, while biomimetic deposition had little effect on fixation [1087].

As briefly mentioned in Section 5 above, $CaPO_4$ can be deposited as various biocom- posites with numerous additives. Among them, drugs, amino acids, and other biolog- ically active compounds such as hormones, peptides, genes, growth factors and DNA are present [61,580,843,845,1088–1097]. Antibiotic-containing $CaPO_4$ deposits were found to show significant in vivo improvements in infection prevention when compared to just $CaPO_4$ deposits [1089,1094–1097]. Similar effects were also seen in Ag-doped deposits [824–835]. These bioactive molecule delivery methods extend the function of $CaPO_4$ deposits to promote new bone formation in orthopedic implants. However, there are still many open questions regarding the incorporation method and optimal release kinetics of antibiotics.

In the case of porous implants, the deposited $CaPO_4$ facilitate bone penetration within the pores [1098]. Furthermore, one study concluded that there was significantly less pin loosening in the $CaPO_4$-treated group [1099]. Thus, many clinical studies are optimistic about the in vivo performance of $CaPO_4$-stored implants. Nevertheless, for the sake of objectivity, it is also necessary to mention the studies in which no positive effects were found [1100,1101]. Furthermore, the presence or absence of the positive effects may depend on the deposition method [1087,1102,1103] and the coating supplier [996]. Moreover, the

application or non-application of post-deposition treatments also affects the biological response of $CaPO_4$ deposits [1104]. These uncertainties may be due to several reasons, including variations in chemical and phase composition, porosity and additives, as well as various surgeon and patient factors that often confound clinical trials.

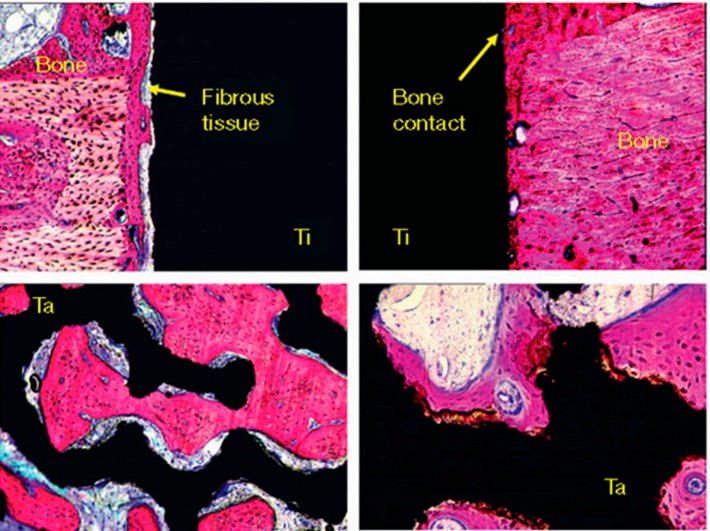

**Figure 40.** Osteointegration properties of uncoated (**left**) and $CaPO_4$-biomimetic coated metal implants (**right**) are compared after 6 weeks of implantation in goat femurs. Reproduced with permission from [180].

In biomedical applications, bone grafts are usually much thicker than the $CaPO_4$ deposits applied on them. Thus, the coated implants combine the surface biocompatibility and bioactivity of $CaPO_4$ with the core strength of a strong substrate (Figure 41) [1105]. Clinical results of coated implants reveal a much longer post-implantation lifetime than uncoated devices and are therefore particularly beneficial for younger patients [1106]. Their biomedical properties approach those of bioactive glass-coated implants [1107,1108].

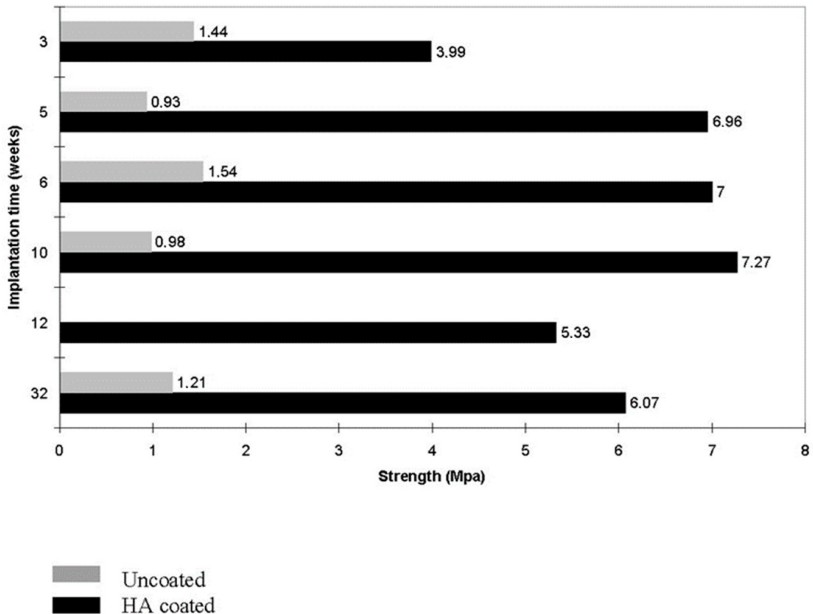

**Figure 41.** A demonstration of how plasma-sprayed HA coatings on porous titanium (**black rods**) increase interfacial bond strength compared to uncoated porous titanium (**grey rods**) depending on implantation time. Reproduced with permission from [1105].

Among the available $CaPO_4$ compounds, HA seems to be the most popular deposition material, so most of the clinical studies have been performed with HA. Namely, HA coatings as an in vivo fixation system for hip implants have been found to perform well in the short to medium term: 2 years [1109], 5 years [1110], 6 years [1111], 8 years [1112,1113], 9 to 12 years [1114], 10 years [1115–1117], 10 to 13 years [1117], 10 to 15.8 years [1118], 10 to 17 years [1119], 13 to 15 years [1120], 15 years [1121], 15 to 21 years [1122], 16 years [1123], 17 years [1124], 17 to 25 years [1125], 18 years [1126], 19 years [1127], 25–30 years [244] and 30–35 years [1128]. Similar data have been obtained for HA-coated [1129–1133] and biphasic HA + TCP coated [1134] dental implants. Longer-term clinical results are still awaited with great interest. Additional details on this topic can be found in the references [32,179,304,340,1135,1136].

At the end of this section, it should be emphasized that many in vivo studies on $CaPO_4$ deposits have shown stronger and faster fixations, more bone growth at the interface, etc., but not all types of $CaPO_4$ deposits give the same results [884,885]. Furthermore, negative results should always be kept in mind and the reasons for this should be carefully investigated and understood. Thus, the clinical applications of $CaPO_4$ deposits are still far from faultlessness. The main areas of concern are listed below [981–983]:

In vivo degradation and resorption of $CaPO_4$ deposits can lead to a loss of bond strength between the substrate and the coating, which can hinder implant fixation.

Delamination and delamination of deposits can induce the formation of particle debris.

$CaPO_4$ deposited on polymers may also alleviate osteolysis problems by causing increased polymer wear from the acetabular cup.

In addition, in vivo studies are still scarce in the literature. The limitations of such experiments can be attributed to the following reasons:

It is difficult to select an appropriate animal model to simulate the actual mechanical loading and unloading states to which an implant may be subjected in a human environment.

Normally, experiments require the sacrificing of many animals because of the statistical analysis needed to validate the results.

These experiments demand high costs and long clinical trial durations.

The lack of collaboration between materials scientists and biologists has led to a lack of understanding of this interdisciplinary topic.

The use of animals in experiments raises serious ethical issues because of the painful procedures and exposure to poisons that occur during experiments.

To complete this section, one should briefly mention some non-biomedical applications of $CaPO_4$ deposits. For example, β-TCP was deposited on Mg alloys and further tested as anodes in biocompatible and degradable batteries [1137]. In addition, alginate coatings charged by a complex of CDHA with lactoferrin and quercetin were found to enhance the shelf life of pork [1138].

## 9. Future Directions

A potential drawback of most deposition techniques is their relatively high cost for large-scale production. Therefore, to reduce the processing time and make their production commercially viable, it is desirable to process the finest $CaPO_4$ deposits that can significantly enhance the biological response [1139]. The need for and effectiveness of $CaPO_4$ deposits at anatomic sites, their robustness in withstanding physiologic loading without fragmentation and issues related to wear of the semicircular canals limit their wider use. Further research to answer these requests will improve the mechanical and biological behavior of $CaPO_4$ deposits, and optimize their safety and efficacy. More attention should be paid to biomimetic-like functionally graded $CaPO_4$ deposits with an unstructured top layer and a crystalline bottom layer. This will allow the resorption rate adjustment to the values at which new bone grows at the initial stages, which is most important for successful mineralization [1140]. In addition, the therapeutic potential of $CaPO_4$ deposits as patterns for in situ delivery of drugs and bone inducers (growth factors, peptides and hormones) at the required time should also be further elucidated. Those biomedical credentials could be further enhanced by the addition of growth factors and other molecules to provide

truly osteoinductive properties [1141]. Furthermore, the surface functionalization of some inorganic compounds, such as calcium carbonates, by $CaPO_4$ deposition seems to be very promising [1142].

In the future, the goal will be to create therapeutic deposits with dual beneficial effects by combining osteoconductivity with the ability to directly deliver therapeutic drugs, proteins and growth factors. Such new types of $CaPO_4$ deposition technologies could offer the ability to stimulate bone growth, fight infections and ultimately extend the life of implants. Furthermore, such deposits may be useful for non-viral transfection of stem cells [1143]. Finally, using $CaPO_4$ deposits as the templates for cells seems to be the most promising biomedical direction [1144].

## 10. Conclusions

As clearly written in the Abstract, the main aim of this article is to collect, summarize and systematize the literature on known deposition methods to prepare $CaPO_4$ coatings, films and/or layers on various types of substrates with the purpose of imparting or improving their biocompatibility and/or bioactivity. Careful search in the literature revealed the presence of more than 60 different deposition techniques (this number is doubled, if all known modifications are counted) and new techniques are constantly being introduced. Since no technique can provide the perfect deposition, each deposition method requires its own equipment, reagents and conditions resulting in specific advantages and disadvantages (Table 3); in addition, various types of defects such as secondary phases, pores, cracks, adhesion defects and residual stresses will always be present regardless of the chosen deposition procedure. All these defects reduce the durability of the $CaPO_4$ deposits and can lead to partial or complete breakdown, while contacting with body fluids after implantation. Therefore, standardized guidelines for the placement of $CaPO_4$ deposits on implant surfaces do not yet exist. Furthermore, the solubility requirements of $CaPO_4$ deposits remain debatable, because, on the one hand, a partial dissolution of $CaPO_4$ deposits improves osseointegration of the implant and is a fundamental requirement for bioactivity. Nonetheless, this dissolution reduces implant stability and increases the potential for loosening on the other hand. Thus, $CaPO_4$ deposits with lower solubilities and higher stabilities are desirable for the long-term performance of implants because they promote faster initial bone fixation, fill large gaps in the mismatch and degrade at a controlled rate. This results in the necessity of biomimetic-like functionally graded $CaPO_4$ deposits with an amorphous top layer and a crystalline bottom layer.

Many animal studies and in vitro investigations have reported the benefits of using implants coated with $CaPO_4$, but most did not consider the chemical and structural properties of the deposits. In this context, it is difficult to make meaningful comparisons between the various reports and studies. Therefore, much research is still needed to clarify the influence of certain parameters of $CaPO_4$ deposits (such as thickness, Ca/P ratio, phase composition, porosity, surface topography and roughness) that depend on the deposition approach used for bone formation. In addition, future research will require clinical trials to better understand bone response to coated implant surfaces and extensive studies on the binding of $CaPO_4$ deposits to drugs, growth factors and cells. Also, further development of ion-substituted, multilayered, graded and composite $CaPO_4$ deposits is required, including those reinforced with metal, ceramic and polymer particles, as well as carbon nanotubes and natural biopolymers, such as collagen and chitosan. For example, a recent study describes deposition of a double-layer DCPD sandwiched siloxane composite coating on the surface of Mg alloy via three successive techniques: chemical conversion and dip-coating, followed by biomimetic method. That DCPD/polymethyltrimethoxysilane/DCPD-chitosan biocomposite coating successfully achieved a surface functionalization of Mg alloy, which provided an application and promising potential for implantable medical devices and bone tissue engineering [1145].

While it has been generally accepted that $CaPO_4$ deposits increase bone strength and early osteointegration rates, the optimum properties required for maximal bone response

have not yet been reported. Therefore, additional cell culture experiments, animal and clinical studies with well-characterized $CaPO_4$ deposits should be performed and well documented to avoid controversial conclusions. Moreover, standardization regarding both the characterization of $CaPO_4$ deposits and in vivo models would be very helpful to facilitate appropriate comparisons between different preclinical studies [1005].

Finally, clinicians need to consider that $CaPO_4$-coated implants are more susceptible to bacteria compared to uncoated controls. Therefore, $CaPO_4$ deposits possessing antibacterial properties are demanded [1146,1147]. Also, the possibility of degradation of the $CaPO_4$ deposits due to interface disruption between them and the substrate should be considered. It is also important that clinical investigators become familiar with the material properties of the $CaPO_4$-deposited implants.

**Funding:** This research received no external funding.

**Data Availability Statement:** Not applicable.

**Conflicts of Interest:** The authors declare no conflict of interest.

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
