# Peer review of "There Are over 60 Ways to Produce Biocompatible Calcium Orthophosphate (CaPO4) Deposits on Various Substrates"

_jcs, doi:10.3390/jcs7070273_

Round 1
Reviewer 1 Report (Previous Reviewer 2)
Good revision for publication
Extensive editing of English language required
Reviewer 2 Report (New Reviewer)
The present overview is very interesting, but a difficult to read ....It is too long and describes a lot of production techniques of biocompatible calcium orthophosphate ( CaPO4) .
Reviewer 3 Report (New Reviewer)
The review describes in a comprehensive manner over 60 ways to produce biocompatible calcium orthophosphate deposits on various substrates. It must be noted the enormous work to write such a gigantic review. There are 179 pages in total, the manuscript having the size of a book. It contains 41 figures, 3 tables and 1147 bibliographic references.
The manuscript is relevant and the topic is original. The conclusions are consistent with the evidence and arguments presented.
The manuscript is well written and the text is clear and easy to read.
I agree with the publication in its actual form.
Reviewer 4 Report (New Reviewer)
The paper presents a stunning review on CaP coatings on different substrates and is ready to be published.
This manuscript is a resubmission of an earlier submission. The following is a list of the peer review reports and author responses from that submission.
Round 1
Reviewer 1 Report
The huge demand for medical devices made of implantable materials, resulting from the significant progress and development of various fields of surgery and prosthetics, made it necessary to describe the issues related to them and to search for solutions that meet the criteria of biotolerance and biocompatibility. The proposed calcium phosphate surfaces are important as matrices giving high osteoconductive activity and providing support to the newly forming bone, which allows the original shape and volume of the implant to be preserved. They are an important aspect of osteintegration processes and such comprehensive work is important from the point of view of application values.
The work submitted for review contains a list of possibilities and procedures for the preparation of Ca-PO4 layers. The author systematically presents a group of techniques related to deposited CaPO4 (Thermal spraying techniques - 2 techniques, Vapor deposition techniques - 6 techniques, Wet techniques - 10 techniques, Other - 46 techniques.
After reviewing the work, the following comments appear:
1. The work concerns the presentation of techniques for the synthesis of calcium-phosphate layers, taking into account a huge number of literature items. Despite this, their importance, and in particular the comparison with the hydroxyapatite material, is quite limited.
2. "Chapter 7 and 8 I would suggest moving at the beginning of the work before presenting the synthesis techniques.
3. 4.3.7. Hydrothermal deposition - space after subsection
4. The summary section is well-written and adequately summarizes the rather lengthy text of the manuscript.
Author Response
Dear unknown reviewer!
First, thank you very much for spending your valuable time, kind efforts and suggestions to make my manuscript even better. Second, please, find below the point-to-point responses.
Yours sincerely,
Sergey V. Dorozhkin
Reviewer 1.
The huge demand for medical devices made of implantable materials, resulting from the significant progress and development of various fields of surgery and prosthetics, made it necessary to describe the issues related to them and to search for solutions that meet the criteria of biotolerance and biocompatibility. The proposed calcium phosphate surfaces are important as matrices giving high osteoconductive activity and providing support to the newly forming bone, which allows the original shape and volume of the implant to be preserved. They are an important aspect of osteintegration processes and such comprehensive work is important from the point of view of application values.
The work submitted for review contains a list of possibilities and procedures for the preparation of Ca-PO4 layers. The author systematically presents a group of techniques related to deposited CaPO4 (Thermal spraying techniques - 2 techniques, Vapor deposition techniques - 6 techniques, Wet techniques - 10 techniques, Other - 46 techniques.
After reviewing the work, the following comments appear:
- The work concerns the presentation of techniques for the synthesis of calcium-phosphate layers, taking into account a huge number of literature items. Despite this, their importance, and in particular the comparison with the hydroxyapatite material, is quite limited. – I did not think on an importance, I just summarized the available data and knowledge on this specific topic. I believe that this review will be read mainly by those, who knows both a necessity and an importance of calcium phosphate deposition.
- "Chapter 7 and 8 I would suggest moving at the beginning of the work before presenting the synthesis techniques. – This is an interesting point of view; however, moving 2 last chapters to the beginning will require renumbering of over 800 references, which may create many mistakes.
- 4.3.7. Hydrothermal deposition - space after subsection – Thank you, corrected.
- The summary section is well-written and adequately summarizes the rather lengthy text of the manuscript. – Thank you very much.
Reviewer 2 Report
I recommend that the manuscript can be accepted with minor revisions. Please, check the errors/typos throughout the manuscript.
Author Response
Dear unknown reviewer!
First, thank you very much for spending your valuable time, kind efforts and suggestions to make my manuscript even better. Second, please, find below the point-to-point responses.
Yours sincerely,
Sergey V. Dorozhkin
Reviewer 2.
I recommend that the manuscript can be accepted with minor revisions. Please, check the errors/typos throughout the manuscript. – Thank you, a spell checking has been performed.
Round 2
Reviewer 1 Report
The work is ready for publication
Reviewer 2 Report
Please rewrite the manuscript to reduce the similarity index
check the manuscript for English grammar and error